# Fasting shapes chromatin architecture through an mTOR/RNA Pol I axis

Nada Al-Refaie[1,2], Francesco Padovani[1], Johanna Hornung[1], Lorenz Pudelko[1,2], Francesca Binando[1,3], Andrea del Carmen Fabregat[4,5], Qiuxia Zhao [6], Benjamin D. Towbin [7], Elif Sarinay Cenik[6], Nicholas Stroustrup [4,5], Jan Padeken[8], Kurt M. Schmoller [1] & Daphne S. Cabianca [1]✉

Chromatin architecture is a fundamental mediator of genome function. Fasting is a major environmental cue across the animal kingdom, yet how it impacts three-dimensional (3D) genome organization is unknown. Here we show that fasting induces an intestine-specific, reversible and large-scale spatial reorganization of chromatin in *Caenorhabditis elegans*. This fasting-induced 3D genome reorganization requires inhibition of the nutrient-sensing mTOR pathway, acting through the regulation of RNA Pol I, but not Pol II nor Pol III, and is accompanied by remodelling of the nucleolus. By uncoupling the 3D genome configuration from the animal's nutritional status, we find that the expression of metabolic and stress-related genes increases when the spatial reorganization of chromatin occurs, showing that the 3D genome might support the transcriptional response in fasted animals. Our work documents a large-scale chromatin reorganization triggered by fasting and reveals that mTOR and RNA Pol I shape genome architecture in response to nutrients.

The regulation of chromatin architecture at multiple scales, from local DNA looping to larger-scale genome distribution at defined nuclear subcompartments, is a fundamental modulator of genome function[1,2]. During cell differentiation, three-dimensional (3D) genome organization is modified in response to developmental cues[3]. However, it remains largely unexplored whether, and how, environmental signals affect 3D chromatin organization[4].

Nutrients are a major environmental factor since virtually all animals, including humans, are exposed to different diets and to feeding/fasting alternations. Perturbations of intracellular metabolism have been shown to regulate chromatin marks through changes in metabolite levels, potentially influencing gene expression and organismal health[5]. However, whether nutrient availability affects chromatin architecture at a larger scale in a multicellular organism is unknown. Here, we use *Caenorhabditis elegans* to investigate the impact of fasting on

the large-scale genome organization using live imaging within tissues of intact animals, at single-cell resolution.

## Results

### Fasting induces a 3D genome reorganization in the intestine

To investigate whether a complete lack of nutrients affects chromatin architecture, we exposed *C. elegans* larvae at the first larval stage (L1) to 12 h of fasting. As a proxy for 3D genome organization, we monitored global distribution of five different fluorescently labelled histones: all H3 protein variants present in *C. elegans* HIS-72/H3.3, HIS-71/H3.3 and HIS-6/H3.2 (refs. 6,7), as well as HIS-24/H1.1 and HIS-1/H4. Live confocal imaging of fed and fasted worms showed that histones underwent a drastic spatial reorganization forming two 'concentric rings', corresponding to concentric spheres in the 3D space, in all intestinal cell nuclei during fasting (Fig. 1a). To quantify chromatin distribution, we used the brightest

[1]Institute of Functional Epigenetics, Helmholtz Zentrum München, Neuherberg, Germany. [2]Faculty of Medicine, Ludwig-Maximilians Universität München, Munich, Germany. [3]Institute of Medical Sciences, University of Aberdeen, Aberdeen, UK. [4]Centre for Genomic Regulation, The Barcelona Institute of Science and Technology, Barcelona, Spain. [5]Universitat Pompeu Fabra, Barcelona, Spain. [6]Department of Molecular Biosciences, University of Texas Austin, Austin, TX, USA. [7]University of Bern, Bern, Switzerland. [8]Institute of Molecular Biology, Mainz, Germany. ✉e-mail: daphne.cabianca@helmholtz-munich.de

HIS-72/H3.3 signal and segmented nuclei both in 3D and two-dimensions (2D). For each nucleus, we averaged the fluorescence intensity profiles along the normalized centre of nucleolar mass, which is known to be centrally positioned in intestinal nuclei[8], across all possible angles. We found nearly identical results when we compared the 3D and 2D segmentation, with the radial chromatin intensity shifting from a single peak in fed larvae to two peaks during fasting (Extended Data Fig. 1a,b), allowing us to use 2D quantifications henceforth. Interestingly, we find that other tissues, such a hypoderm and muscle, did not reorganize the genome into two rings (Fig. 1a–c). As muscle nuclei are about 25% the size of intestinal nuclei and their radial distance might not always allow to resolve two distinct peaks of chromatin with our imaging system, we repeated the chromatin profile analysis in fasted muscle excluding radial profiles estimated to be not resolvable with our imaging setup (Methods) and confirmed that chromatin rings are not formed during fasting in muscle (Extended Data Fig. 1c,d). These results indicate that fasting induces this 3D chromatin configuration in a tissue-specific manner.

The fact that all histones tested reorganized similarly into two rings in intestine cells suggested that the genome at large, rather than specific subdomains, is spatially reorganized in response to fasting. Indeed, the same concentric rings are visible using live imaging of DNA in fasted worms (Fig. 1d,e and Extended Data Fig. 1e), and quantification of the radial chromatin fluorescence distributions revealed comparable large-scale reorganizations in DNA- and HIS-72/H3.3-labelled nuclei (Fig. 1c,e).

To confirm that this spatial reorganization of chromatin is determined by the nutritional status of the organism, we conducted a number of controls. We showed that similar chromatin reorganizations occurred independently of animals being fed and fasted on plates or in liquid (Fig. 1a–c and Extended Data Fig. 1f–h) and over a wide range of physiological osmolarities (Extended Data Fig. 1i–k). Moreover, we did not observe a fasting-like 3D genome organization in heat- or cold-shock conditions (Fig. 1f,g and Extended Data Fig. 1l), indicating that the observed spatial reorganization of chromatin into two rings is not a general response to stress but is specific to lack of nutrients. Remarkably, adult worms, in which intestinal cells no longer undergo DNA replication[8], also reorganized their genomes into concentric rings upon fasting (Fig. 1h,i and Extended Data Fig. 1m), albeit with a different relative amplitude of the two peaks in the radial chromatin profiles compared with L1s (Extended Data Fig. 1n). This result argues that the newly identified 3D reorganization of the genome can occur in absence of cell division and is not restricted to specific larval stages.

## Chromatin rings form independently of heterochromatic marks

The striking reorganization of the genome in the intestinal nuclei during fasting led us to hypothesize that the formation of concentric rings might reflect an accumulation of chromatin at specific nuclear structures. Indeed, by conducting in vivo imaging, we found that the outer ring aligns with the nuclear envelope, as revealed by monitoring both

chromatin and the nuclear envelope protein EMR-1/emerin (Fig. 2a), while the inner ring encircled the single, centrally located nucleolus of intestinal cells[8], visualized by labelling its core component FIB-1/fibrillarin (Fig. 2b). We note that chromatin was not enriched in direct proximity of the nucleolus in fed intestinal cells, in agreement with recent live imaging results in flies and human lymphocytes[9]. Indeed, by measuring the radial chromatin intensity from the nucleolus towards the nuclear periphery in the fed state, we find that the signal is low at the edge of the nucleolus. In contrast, in fasted cells the chromatin signal is highest near this compartment (Extended Data Fig. 2a), confirming that the inner chromatin ring locates around the nucleolus.

The 3D organization of chromatin reflects the functional compartmentalization of the genome[10], with the nuclear and nucleolar periphery being sites where heterochromatin, the silenced portion of the genome, accumulates[11,12]. Thus, we tested the role of well-established heterochromatic marks, such as histone H3 lysine 9 methylation (H3K9me) and H3K27me3, in chromatin ring formation during fasting. Interestingly, loss of these marks upon deletion of the histone methyltransferases SET-25/SUV39H1 and MET-2/SETDB1 for H3K9me[13] and MES-2/EZH2 for H3K27me3 (ref. 14) did not alter chromatin reorganization during fasting (Fig. 2c and Extended Data Fig. 2b,c), indicating that these heterochromatic marks are dispensable for ring formation.

We then asked how the reorganization of the genome into concentric chromatin rings affects gene positioning. Standard fixation procedures disrupt the fasting-induced organization of chromatin (Extended Data Fig. 2d), as shown for other nuclear substructures[15]. Thus, we chose to monitor a transcriptionally repressed, high-copy number transgene (heterochromatic) and a low-copy number reporter, actively transcribed in intestine (euchromatic), both integrated at a single site in the worm genome, using a LacO/LacI–GFP-based visualization strategy in living cells.

The cells of the *C. elegans* intestine can be divided into three subgroups that show distinct patterns of spatial gene positioning[16] and gene expression[8,17]. We focused on the most abundant of these three intestinal subgroups, namely, the cells that occupy the middle section of the digestive tract (Extended Data Fig. 2e). To quantify positioning of the heterochromatic allele (Extended Data Fig. 2f), we used a zoning assay[18] (Extended Data Fig. 2g) that scores the frequency with which a fluorescently tagged allele is found in one of three concentric zones of equal surface. As previously reported[16], this repressed, repetitive and H3K9me-marked heterochromatic reporter is strongly enriched at the nuclear periphery in fed intestinal cells. We find that its distribution is not altered during fasting (Extended Data Fig. 2h).

Next, we quantified the position of a euchromatic gene expressed under the *pha-4* promoter (Extended Data Fig. 2i), which was previously reported to be located at the nuclear interior in intestine[19], with respect not only to the nuclear periphery but also to the single, centrally located nucleolus. We found that during fasting, this euchromatic allele shifted towards the nucleolus (Extended Data Fig. 2j,k), consistent with a shift to the inner chromatin ring (Fig. 2b and Extended Data Fig. 2a).

---

**Fig. 1 | Fasting induces a large-scale genome reorganization in the *C. elegans* intestine. a**, Single focal planes of representative WT L1 larvae expressing the indicated fluorescently tagged histone, regularly fed or fasted. Insets: zoom of single nucleus of the indicated tissue. Hyp: hypoderm; int: intestine; mus: muscle. Inset scale bar, 2.5 μm. **b**, Heat maps showing the radial fluorescence intensity profiles of HIS-72/H3.3–GFP in intestinal, hypodermal and muscle nuclei of fed and fasted WT animals as a function of the relative distance from the nucleolus centre. Each row corresponds to a single nucleus, segmented in 2D at its central plane. Seventy-two intestinal nuclei were analysed in animals from three independent biological replicas. **c**, Line plots of the averaged single nuclei profiles shown in **b**. The shaded area represents the 95% confidence interval of the mean profile. Single-peak profiles were compared to estimate the statistical significance of differences, as described in the methods. *P* values are given in Supplementary Table 2. **d**, As in **a** but for Hoechst 33342 live-stained animals

to visualize DNA. Insets: zoom of single intestinal nucleus. **e**, Line plots as in **c** but of the averaged single nuclei profile of Hoechst 33342-stained DNA from 72 intestinal nuclei of fed and fasted animals from three independent biological replicas. **f**, As in **a** but of fed L1s expressing HIS-72/H3.3–GFP heat shocked at 34 °C or cold shocked at 4 °C for 6 h. Insets: zoom of single intestinal nucleus. **g**, Line plots as in **c** but of the averaged single nuclei profile of HIS-72/H3.3–GFP from 48 intestinal nuclei of L1s either heat shocked or cold shocked, from two independent biological replicas. **h**, Single focal planes of representative adults expressing HIS-72/H3.3–GFP, regularly fed or 12 h fasted. Insets: zoom of single intestinal nucleus, scale bar 5 μm. **i**, Line plots as in **c** but of the averaged single nuclei profile of HIS-72/H3.3–GFP from 60 intestinal nuclei of fed and fasted adults from three independent biological replicas. For **e**, **g** and **i**, heat maps of single nuclei profiles are provided in Extended Data Fig. 1e,l,m, respectively. Source numerical data are available in Source data.

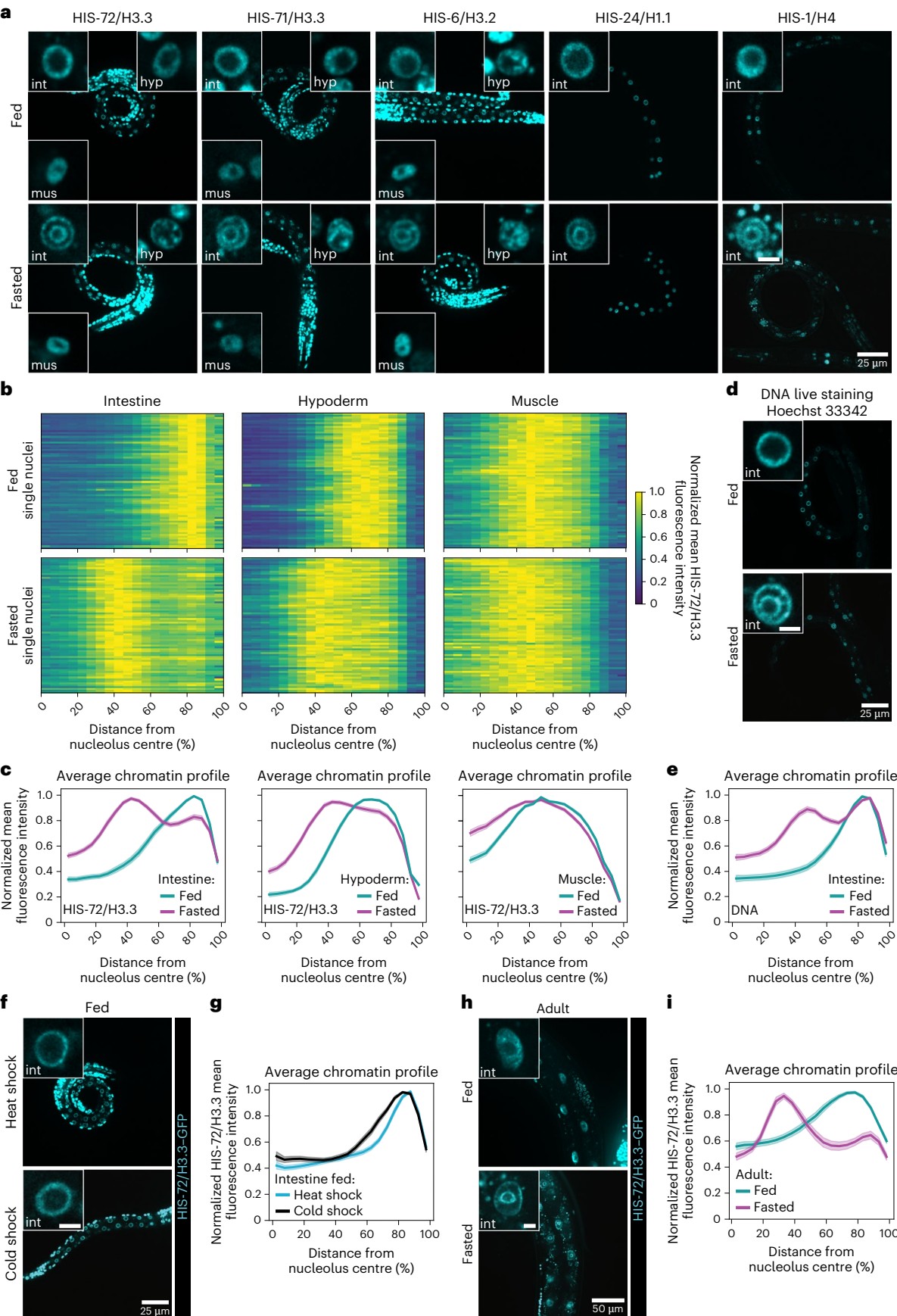

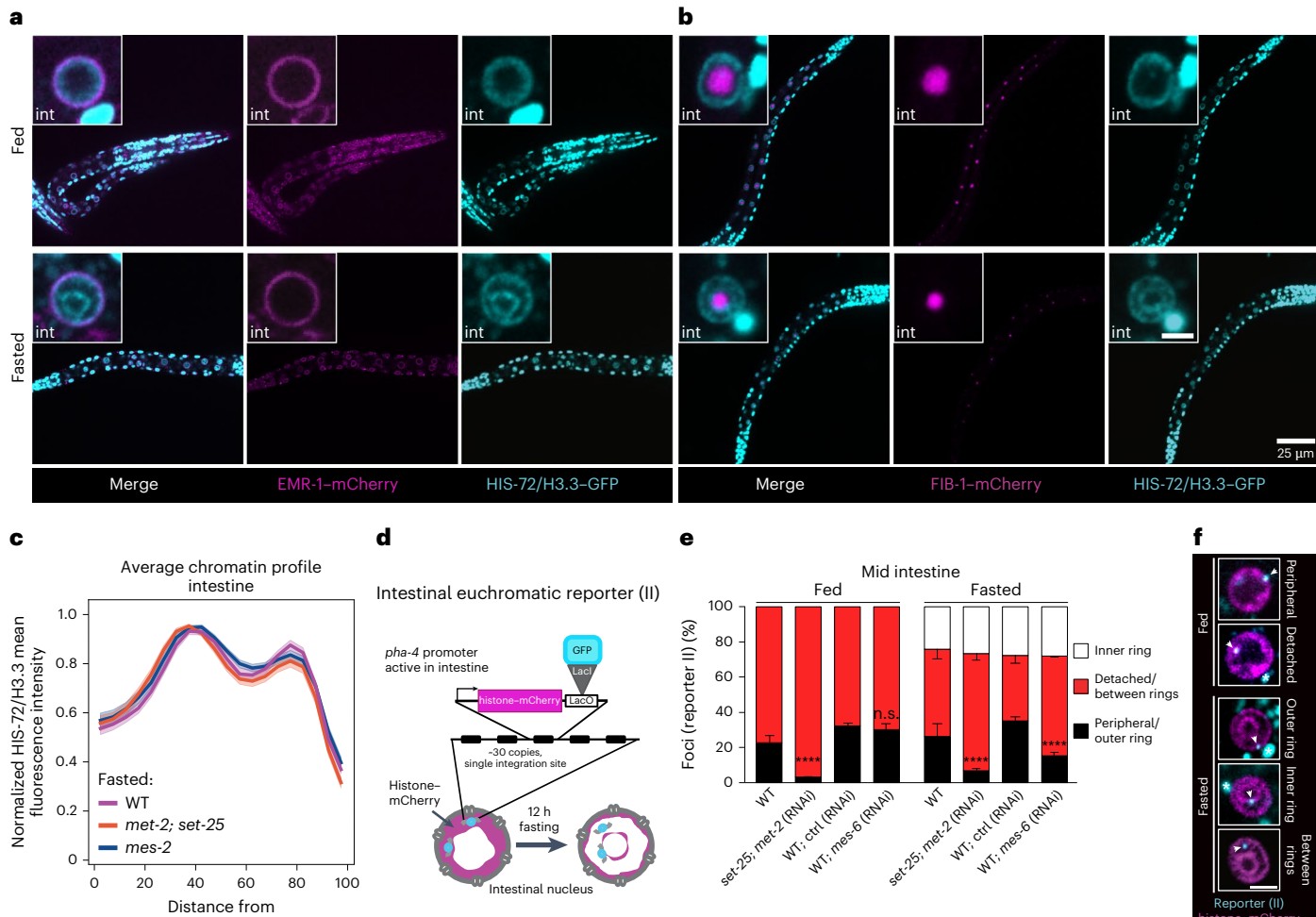

**Fig. 2 | The inner and outer chromatin rings form independently of heterochromatic marks. a**, Single focal planes of representative WT L1 larvae bearing HIS-72/H3.3–GFP and EMR-1–mCherry in fed and fasted conditions. **b**, As in **a**, but expressing FIB-1–mCherry. Insets: zoom of single intestinal (int) nucleus. Inset scale bar, 2.5 µm. **c**, Line plots of the averaged single nuclei profile of HIS-72/H3.3–GFP from 70 intestinal nuclei of fasted WT, *met-2; set-25* and *mes-2* mutants, from three independent biological replicas, showing the radial fluorescence intensity as a function of the relative distance from the nucleolus centre. The shaded area represents the 95% confidence interval of the mean profile. Heat maps displaying the single nuclei profiles are provided in Extended Data Fig. 2c. **d**, Schematic representation of the euchromatin reporter (II) used in **e** and **f**. **e**, Quantification of the subnuclear distribution of the active allele represented in **d** in the indicated locations in fed and fasted animals of the indicated genotype and/or RNAi treatment. The averages of five independent biological replicates are shown for WT, and three independent biological replicates for *set-25*; *met-2* (RNAi), WT; control (ctrl) (RNAi) and WT; *mes-6* (RNAi). The error bars show the s.d. Samples were compared pairwise to WT or WT ctrl RNAi by chi-squared tests. ****$P < 0.0001$. Exact $P$ values and $n$ are in Supplementary Table 2. **f**, Single focal planes of representative single WT intestinal nuclei in L1 larvae expressing the indicated fluorescently tagged markers, fed (top) or fasted (bottom) and with the indicated subnuclear location of the allele in **d**. Scale bar. 2.5 µm, asterisk denotes autofluorescence from the intestinal cytoplasm. Source numerical data are available in Source data.

To determine the position of a euchromatic reporter with respect to the two rings during fasting, we used a transcriptionally active locus carrying the same *pha-4* promoter as before but driving the expression of a histone–mCherry cassette (Fig. 2d). As expected, in wild-type (WT) fed animals the active reporter predominantly occupied an internal position (77.2 ± 4%; Fig. 2e,f). Animals lacking H3K9me (*set-25/SUV39H1* mutants with *met-2/SETDB1* RNA interference (RNAi)) have more loci internally located (96.9 ± 0.3%), in agreement with the role of this mark in perinuclear positioning of genes in worms[13,20] and mammals[21–24]. By contrast, reducing H3K27me3 levels by knocking down *mes-6/EED*[25] did not cause a major allele repositioning in fed larvae (Fig. 2e) (69.9 ± 3.4%). During fasting, in WT animals, 24.1% (±5.5%) of the reporter alleles are localized to the inner ring, 26% (±7.4%) are found at the outer ring (Fig. 2e,f) and the majority occupied the space between the rings (49.9% (±10.6%)). Interestingly, in both *set-25/SUV39H1;met-2/SETDB1* and *mes-6/EED* loss of function animals we did not observe alterations in the frequency with which the reporter localizes at the inner ring

(26.7% (±3.7%) and 28.2% (±0.4%) for reduced H3K9me and H3K27me3, respectively). However, both heterochromatin perturbations led to a reduced positioning at the outer ring (6.9% (±1.2%) and 15.1% (±2%) for impaired H3K9me and H3K27me3, respectively; Fig. 2e).

These results indicate that positioning at the inner ring is independent of the heterochromatic marks tested, while H3K9me and, to a lesser degree, H3K27me3 contribute to positioning of the *pha-4* allele at the outer ring.

## mTOR regulates chromatin architecture in response to nutrients

The regulation of the large-scale chromatin organization in the intestine by fasting suggests that nutrient-sensing signalling pathways might be implicated. We investigated the role of adenosine monophosphate (AMP)-activated kinase (AMPK), which is activated by a reduction in the ATP/AMP ratio[26] under poor nutrient conditions. Despite causing a decreased survival to prolonged starvation[27], mutating the two

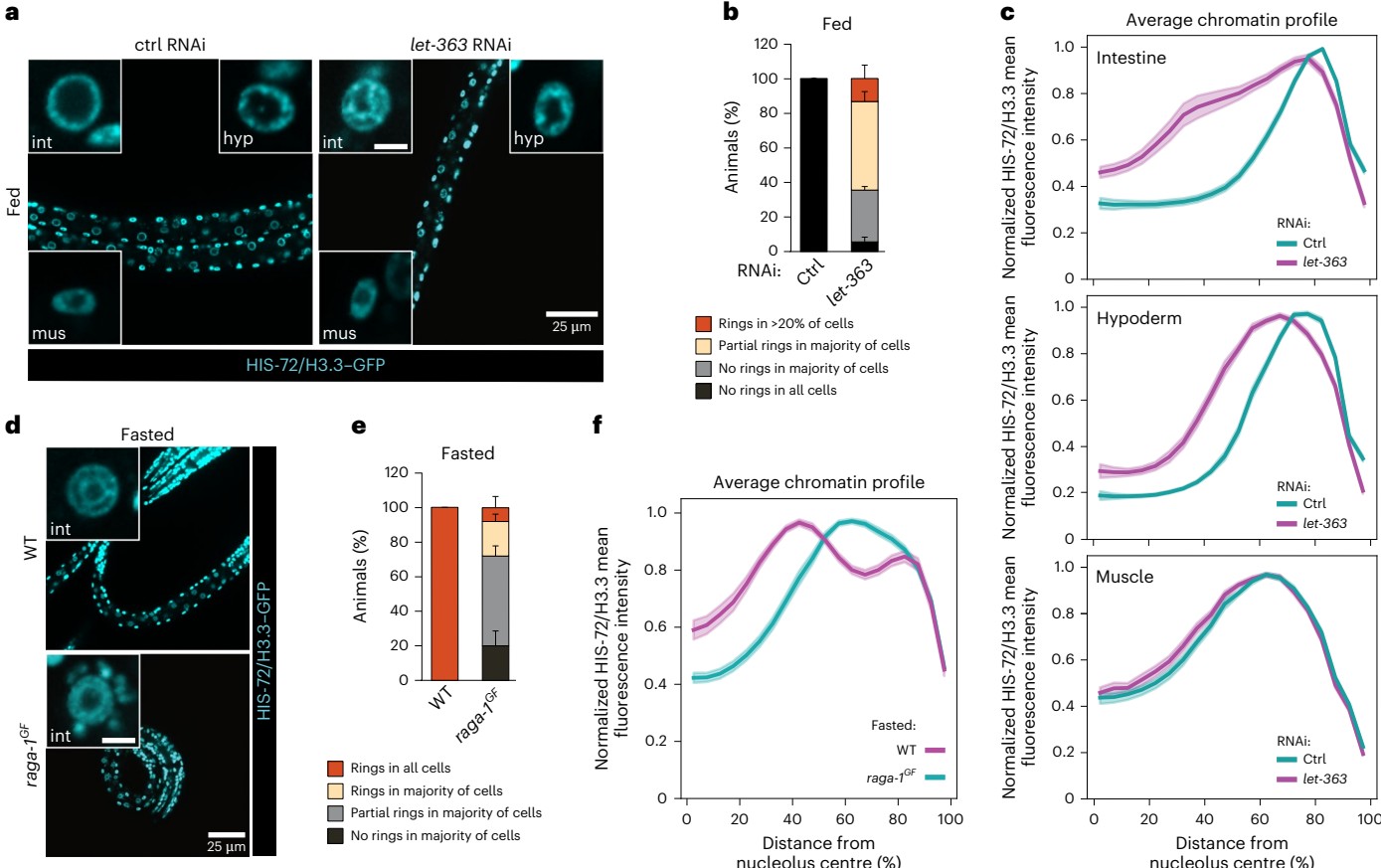

**Fig. 3 | mTORC1 signalling is necessary and sufficient to regulate 3D genome architecture in response to nutrients in intestine. a**, Single focal planes of representative fed L1 expressing HIS-72/H3.3–GFP upon control (ctrl) or *let-363* RNAi treatment. Insets: zoom of single nucleus of the indicated tissue. Hyp, hypoderm; int, intestine; mus, muscle. Inset scale bar, 2.5 µm. **b**, Quantification of the percentage of fed animals under control or *let-363* RNAi, within the indicated categories for 3D chromatin organization in intestine. Data are shown as mean ± s.e.m. of three independent biological replicas. **c**, Line plots of the averaged single nuclei profile of HIS-72/H3.3–GFP from 72 intestinal, hypodermal or muscle nuclei of fed animals upon control and *let-363* RNAi, showing the radial fluorescence intensity as a function of the relative distance from the nucleolus centre. Data are from three independent biological replicas. For *let-363* RNAi, larvae in proportions to their relative abundance within the chromatin organization categories as in **b**, were analysed for all tissues. The shaded area

represents the 95% confidence interval of the mean profile. Profiles with a single peak were compared to estimate the statistical significance of differences, as described in Methods. *P* values are given in Supplementary Table 2. **d**, As in **a** but showing fasted WT and *raga-1^GF^* animals expressing HIS-73/H3.3–GFP. Insets: zoom of single nucleus of the intestine. **e**, Quantification of the percentage of fasted WT and *raga-1^GF^* animals in the indicated categories for 3D chromatin organization in intestine. Data are shown as mean ± s.e.m. of three independent biological replicas. **f**, Line plots as in **c** but of the averaged single nuclei profile of HIS-72/H3.3–GFP from 72 intestinal nuclei of fasted WT and *raga-1^GF^* larvae. Data are from three and four independent biological replicas for *raga-1^GF^* and WT, respectively. For *raga-1^GF^*, larvae in proportions to their relative abundance within the chromatin organization categories as in **e**, were analysed. For **c** and **f**, heat maps of single nuclei profiles are provided in Extended Data Fig. 3h,p, respectively. Source numerical data are available in Source data.

*C. elegans* AMPK catalytic subunits *aak-1* and *aak-2* did not alter the 3D chromatin reorganization of intestinal cells during fasting (Extended Data Fig. 3a–c), suggesting that the AMPK signalling pathway is not involved. Similarly, knocking down MPK-1/ERK1/2, a component of the mitogen-activated protein kinase (MAPK) pathway, which is activated during lack of nutrients[28–30], did not perturb the formation of the chromatin rings during fasting (Extended Data Fig. 3d–f).

The mechanistic target of rapamycin (mTOR) signal-transduction pathway allows cells to adjust their protein biosynthetic capacity to nutrient availability[31]. In particular, mTOR is active in rich nutrient conditions and inactive in the absence of nutrients. We thus asked what is the effect of mTOR inhibition on the spatial organization of chromatin in fed animals. To this aim, we performed RNAi of the kinase LET-363/mTOR as done previously[32] and verified that, as expected, this leads to developmental arrest as L3s (Extended Data Fig. 3g). Before arresting, *let-363/mTOR* RNAi animals grow more slowly than controls. Therefore, we monitored chromatin organization in stage-matched L1s and found that mTOR inhibition is sufficient to induce a partial reorganization of

chromatin into concentric rings in the intestine of fed animals, but not in hypoderm or muscle (Fig. 3a–c and Extended Data Fig. 3h), even when analysing only the longest radial distances for muscle nuclei (Extended Data Fig. 3i,j). Raptor, termed DAF-15 in worms[33], is a critical effector of mTORC1, the main mTOR-containing complex responsible for sensing nutrients[34] and fundamental for larval development[35] (Extended Data Fig. 3k). Interestingly, depleting DAF-15/Raptor using the auxin-inducible degradation (AID) system also led to the partial reorganization of chromatin into two rings in the intestine of fed animals (Extended Data Fig. 3l–o). This reorganization resembled that observed for the impairment of LET-363/mTOR, suggesting that mTORC1 is involved in the regulation of the 3D chromatin architecture in response to nutrients.

In mammals, the inactivation of mTORC1 in absence of nutrients is counteracted by a Q66L substitution in RagA, a protein of the Rag family of GTPases that acts upstream of mTORC1. This mutation impairs GTP hydrolysis, rendering RagA constitutively active, thus keeping mTOR signalling active even when nutrients are lacking[36,37]. In *C. elegans*, a similar phenotype is observed when RAGA-1, a homologue

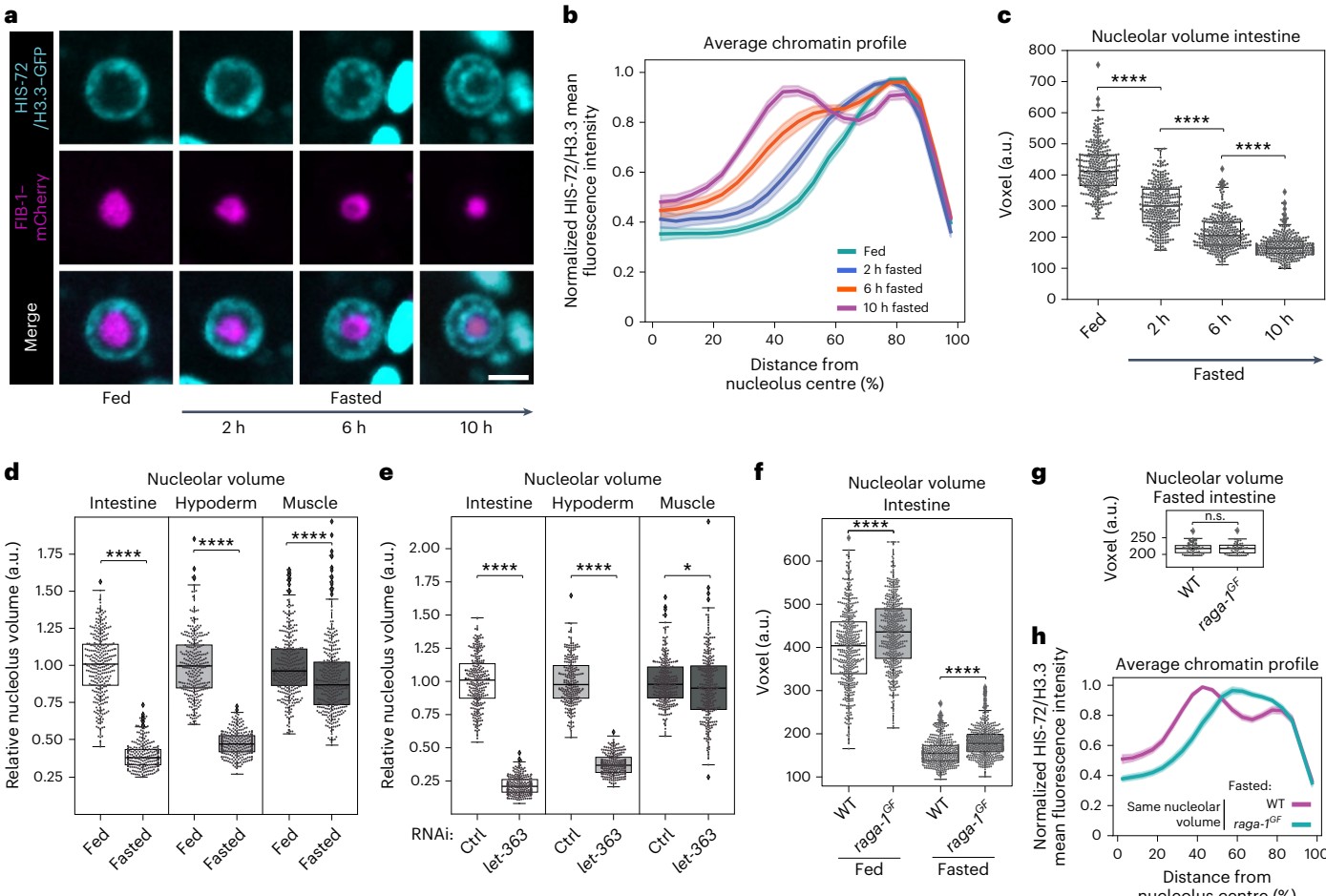

**Fig. 4 | Remodelling of the nucleolus correlates with chromatin ring formation. a**, Single focal planes of representative single intestinal nuclei expressing HIS-72/H3.3–GFP and FIB-1–mCherry in WT L1 larvae that were fed or fasted for the indicated time. Scale bar, 2.5 μm. **b**, Line plots of the averaged single nuclei profile of HIS-72/H3.3–GFP from 72 intestinal nuclei of WT animals under the indicated nutritional status, from three independent biological replicas, showing the radial fluorescence intensity as a function of the relative distance from the nucleolus centre. The shaded area represents the 95% confidence interval of the mean profile. Single-peak profiles were compared to estimate the statistical significance of differences, as described in Methods. *P* values are given in Supplementary Table 2. **c**, Box plot comparing the volume of the nucleolus, measured with FIB-1–mCherry, in the intestine of larvae that were fed or fasted for the indicated time. **d**, Box plot as in **c** but in the intestine, hypoderm and muscle of fasted WT animals relative to the corresponding tissue

in fed animals. **e**, Box plot as in **d** but for fed larvae under *let-363* RNAi relative to control RNAi in the corresponding tissue. **f**, Box plot as in **c** but in the intestine of fed and fasted WT and *raga-1GF* larvae. **g**, Box plot as in **f** but for 50 fasted WT and *raga-1GF* intestinal cells selected for having the same nucleolar volume. **h**, Line plots as in **b** but of the averaged single nuclei profile of HIS-72/H3.3–GFP from the intestinal nuclei shown in **g**. For **b** and **h**, heat maps displaying the single nuclei profiles are provided in Extended Data Fig. 4a,c, respectively. For **c**–**g**, box limits are 25th and 75th percentiles, whiskers denote 1.5 times the interquartile ranges, points outside the whiskers are outliers and the median is shown as a line. Probability values from two-sided Wilcoxon rank sum tests are shown: *\**P* < 0.05 and \*\*\*\**P* < 0.0001. Data are from three independent biological replicas. See *P* values and *n* in Supplementary Table 2. Source numerical data are available in Source data.

of RagA, carries a Q63L mutation[38]. We introduced the Q63L mutation in the endogenous *raga-1* gene, thus generating a constitutively active gain-of-function mutant, *raga-1GF*. Remarkably, keeping mTORC1 active during fasting by expressing RAGA-1GF antagonized the reorganization of chromatin into concentric rings in the intestine (Fig. 3d–f and Extended Data Fig. 3p), revealing that mTOR inactivation is required for the spatial reorganization of the genome induced by fasting.

## Nucleolar remodelling correlates with chromatin ring formation

To gain insights into the kinetics of chromatin reorganization during fasting, we monitored the radial distribution of chromatin in L1 larvae at earlier timepoints during fasting, namely, 2, 6 and 10 h. While after 2 h without nutrients the radial distribution of chromatin yields a single peak, we detected a partial reorganization of chromatin into two rings after 6 h and the two chromatin rings were fully formed after 10 h (Fig. 4a,b

and Extended Data Fig. 4a). In parallel, by using live imaging of animals expressing FIB-1/fibrillarin–mCherry, we quantified the volume of the nucleolus, which is a major contributor to chromatin organization[39–42] and has previously been shown to be affected by nutrients[43,44]. We found that the size of the nucleolus is strongly reduced after 2 h of fasting and further decreases at later timepoints (Fig. 4c), indicating that a reduction in nucleolar size precedes the formation of the two chromatin rings.

In agreement with previous observations[45], nucleoli of different tissues have remarkably different sizes, with intestinal cells having the largest nucleoli compared with muscle and hypoderm (Extended Data Fig. 4b). Upon 12 h of fasting, the volume of the nucleolus decreases in all tissues tested, albeit to a different degree (Fig. 4d).

mTOR is an evolutionary conserved regulator of nucleolar size[43,44,46]. Consistently, we found that inhibition of mTOR signalling through knockdown of LET-363/mTOR, led to a strong reduction in nucleolar volume in intestine and hypoderm and to a weaker change

in muscle (Fig. 4e), mirroring what observed during fasting. Notably, both for mTOR inhibition and fasting, the strongest relative decrease in nucleolar volume compared with the respective control is observed in the intestine (Fig. 4d,e). Interestingly, RAGA-1^GF- expressing animals, which fail to reorganize the genome into chromatin rings upon fasting (Fig. 3d–f), display larger nucleoli compared with WT (Fig. 4f), suggesting that reducing nucleolar volume to a critical size might be required for ring formation. To test whether differences in nucleolar volume could directly explain the occurrence of chromatin rings, we selected WT and *raga-1^GF* fasted intestinal cells with the same nucleolar volume (Fig. 4g). We found that the differences in chromatin organization between the strains persist: WT animals reorganize the genome into two rings while *raga-1^GF* mutants do not (Fig. 4h and Extended Data Fig. 4c). Importantly, we obtained analogous results when we selected intestinal cells with the same nucleolar and nuclear area (Extended Data Fig. 4d–f). These results suggest that, while a reduction in nucleolar volume might be necessary for 3D genome reorganization in the intestine, changes in nucleolar and nucleolar/nuclear size per se are not sufficient to induce the formation of the two chromatin rings.

## RNA Pol I is required to restore a fed-like genome architecture

As nutrient deprivation induces a reorganization of the 3D genome, we asked whether refeeding would reverse it. We therefore fasted L1 larvae for 12 h, put them back on food and monitored the intestinal chromatin distribution over time. While after 5 min of refeeding the genome is still organized into concentric rings (Fig. 5a,b and Extended Data Fig. 5g), 30 min on food was sufficient to restore the normal 'fed' 3D genome configuration (Fig. 5a,b and Extended Data Fig. 5g). This shows that the spatial architecture of the genome of intestinal cells is dynamic and reflects the nutritional status of the organism.

Upon refeeding, the nucleoli enlarge. Whereas 2 h are required for the nucleolus to regain the size of fed animals, refeeding for only 5 min was sufficient to detect an increase in nucleolar volume (Fig. 5c). This argues that nucleolar size is regulated very early in the response to refeeding and its increase might be part of the changes that contribute to the dispersion of the chromatin rings.

Typically, nucleolar size reflects the production of ribosomal RNA by RNA Pol I, with bigger nucleoli correlating with higher levels of transcription[47–49]. However, nucleolar structure and size have also been reported to be regulated by RNA Pol II[50,51]. As transcription is a major driver of genome organization[52–56], we tested whether RNA Pol I, Pol II or Pol III have a role in restoring the fed-like 3D chromatin organization upon refeeding. To this end, we introduced AID[57] and a fluorescent tag at the endogenous loci of the genes *rpoa-2*, *rpb-2* and *rpc-1* (corresponding to *POLR1B*, *POLR2B* and *POLR3A* in humans), which encode core subunits specific to RNA Pol I, Pol II and Pol III, respectively. The addition of auxin should lead to their acute degradation, selectively inhibiting the transcriptional activity of the targeted RNA polymerase.

One hour on auxin rendered the RNA Pol III- and II-specific core subunits RPC-1 and RPB-2 undetectable in intestine, while about 4 h were needed for the RNA Pol I-specific component RPOA-2 (Extended Data Fig. 5a–f).

We fasted larvae to induce the 3D genome reorganization into two rings, added auxin during the last 2 h of nutrient deprivation and subsequently placed the worms on food- and auxin-containing plates for 30 min (Fig. 5d). The 3D chromatin architecture reverted to the fed-like state in control animals (TIR1-expressing only), as well as in those depleted for RNA Pol II and Pol III subunits. However, the animals depleted for the RNA Pol I core subunit were unable to restore the fed-like chromatin configuration, retaining the concentric rings of fasted cells despite feeding (Fig. 5e–g and Extended Data Fig. 5h). As DNA staining can only be obtained if worms eat Hoechst 33342-containing bacteria, we can exclude that the retention of the chromatin rings in RNA Pol I-depleted animals stems from a failure to eat. From this, we conclude that RNA Pol I, but not Pol II nor Pol III, is necessary to re-establish the fed-like genome architecture upon refeeding.

## RNA Pol I inhibition induces chromatin rings in fed animals

When nutrients are scarce, cells reduce ribosome synthesis by down-regulating RNA Pol I transcription[31,58–60]. A fasting-induced decrease in RNA Pol I activity via mTOR inhibition[61–63] is compatible with the reduced nucleolar volume that we observed upon fasting (Fig. 4d) and *let-363/mTOR* knockdown (Fig. 4e). Consistently, pre-rRNA levels are reduced in intestine, hypoderm and muscle of fasted and LET-363/mTOR-depleted animals (Extended Data Fig. 6a,b). However, RNA Pol I is only one of many targets affected by fasting[63]. To determine whether inhibition of RNA Pol I transcription per se is sufficient to disrupt the normal 3D genome organization in fed animals, we monitored chromatin distribution in fed larvae upon degradation of RPOA-2. Strikingly, even though complete RPOA-2 degradation in fed worms required nearly 4 h on auxin (Extended Data Fig. 5a,b), 1 h on auxin was sufficient to induce strong changes in chromatin distribution (Fig. 6a,b). After 3 h on auxin, the concentric chromatin rings appeared in the majority of larvae (Fig. 6a,b) and a fasting-like radial spatial reorganization of chromatin could be observed when averaging over all cells at this timepoint (Fig. 6c and Extended Data Fig. 6c). After 5 h, 100% of these animals had converted the fed chromatin distribution into two concentric chromatin rings in more than 80% of the intestinal cells (Fig. 6a,b).

To test whether this effect is specific to RNA Pol I transcription, we blocked the activity of RNA Pol II and III in fed animals, using the RBP-2- and RPC-1-degron strains, respectively. Remarkably, after 1 h on auxin we detected no changes in the global distribution of chromatin in intestine (Fig. 6a,b), despite both core subunits being fully degraded at this timepoint (Extended Data Fig. 5c–f). While loss of RNA Pol III transcription did not affect chromatin organization even after 3 or 5 h of auxin exposure (Fig. 6a–c and Extended Data Fig. 6c), variable changes in chromatin distribution were observed upon RNA

---

**Fig. 5 | RNA Pol I is required to restore a fed-like genome architecture.**
**a**, Single focal planes of representative intestinal nuclei expressing HIS-72/H3.3–GFP and FIB-1–mCherry in WT L1 larvae fed, fasted or refed for the indicated time. Scale bar, 2.5 μm. **b**, Line plots of the averaged HIS-72/H3.3–GFP profiles from 72 intestinal nuclei of animals in the indicated nutritional status, from three independent biological replicas, showing the radial fluorescence intensity as a function of the relative distance from the nucleolus centre. The shaded area represents the 95% confidence interval of the mean profile. Single-peak profiles were compared to estimate the statistical significance of differences as described in Methods. Exact *P* values are in Supplementary Table 2. **c**, Box plot comparing nucleolar volume in intestine, measured with FIB-1–mCherry, in larvae of the indicated nutritional status. Box limits are 25th and 75th percentiles, whiskers denote 1.5 times the interquartile ranges, points outside the whiskers are outliers and the median is shown as a line. Probability values from two-sided Wilcoxon rank sum tests comparing to fed are shown: ****P < 0.0001. Data are from three independent biological replicas. See *P* values and *n* in Supplementary Table

2. **d**, A scheme of the experimental timeline in **e**–**g**. **e**, Single focal planes of representative L1s live stained with Hoechst 33342, expressing TIR1 only (control) or either RPOA-2-AID (RNA Pol I-AID), RPB-2-AID (RNA Pol II-AID), RPC-1-AID (RNA Pol III-AID), fasted or 30 min refed in the presence of auxin, as outlined in **e**. Insets: zoom of single intestinal nucleus. Scale bar, 2.5 μm. **f**, Percentage of TIR1 only and AID-tagged animals within the indicated 3D chromatin organization categories in intestine after 30 min refeeding. The mean ± s.e.m. of three independent biological replicas is shown. **g**, Line plots as in **b** but of the averaged Hoechst 33342-stained DNA profiles in intestinal nuclei of animals expressing the indicated AID tag in presence of auxin, fasted or 30 min refed, from three independent biological replicas. Seventy-two single intestinal nuclei were analysed, except for fasted RNA Pol II-AID and 30 min refed TIR1 only, where 69 and 71 nuclei were analysed, respectively. For **b** and **g**, heat maps displaying the single nuclei profiles are provided in Extended Data Fig. 5g,h, respectively. Source numerical data are available in Source data.

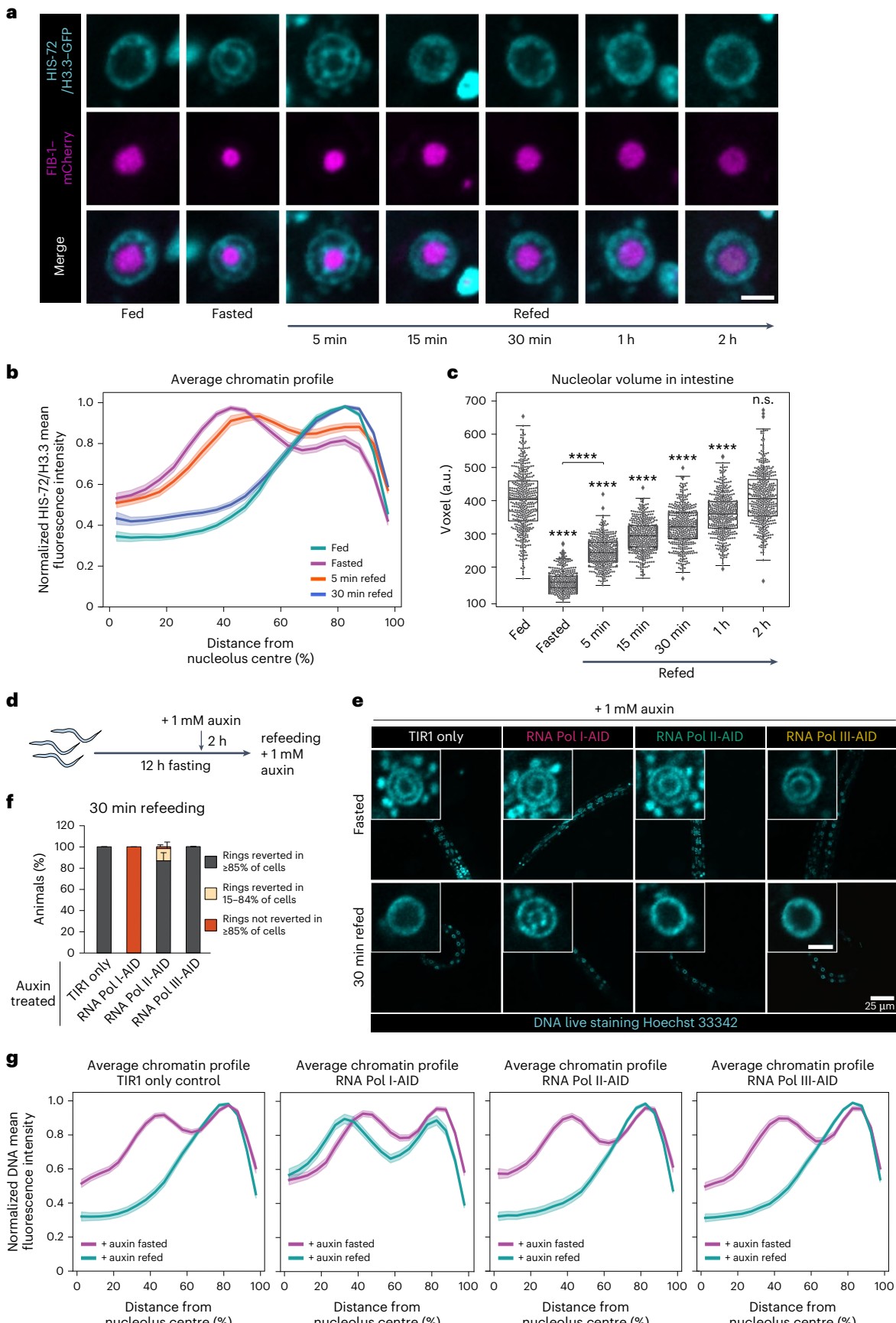

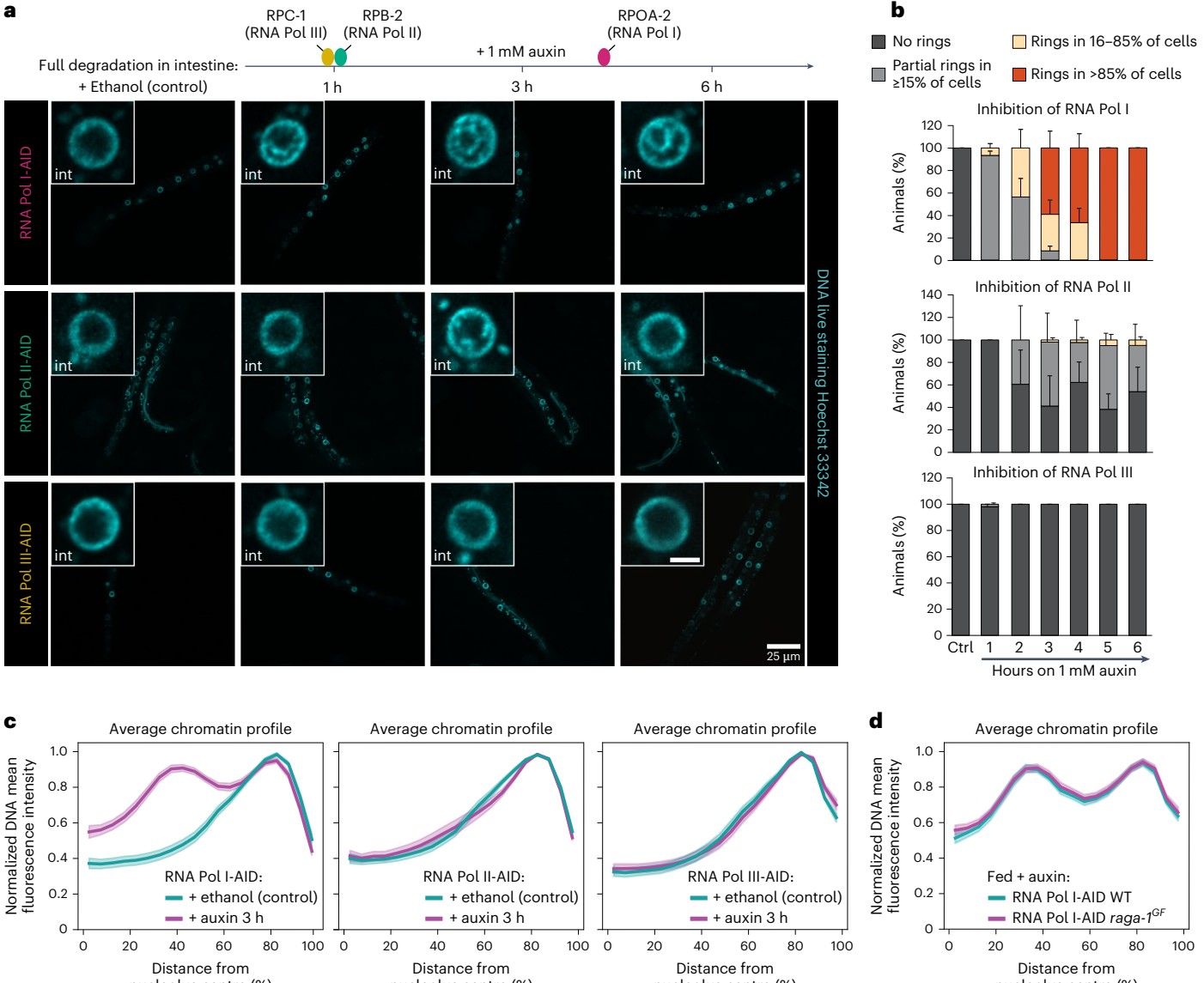

**Fig. 6 | Inhibition of RNA Pol I activity is sufficient to induce a fasting-like chromatin architecture in the intestine of fed animals. a**, Top: schematic representation of the degradation kinetics of the core subunits RPC-1 (RNA Pol III), RPB-2 (RNA Pol II) and RPOA-2 (RNA Pol I) in intestine, as quantified in Extended Data Fig. 5a–f. Bottom: single focal planes of representative fed L1 larvae live stained with Hoechst 33342 to monitor DNA, ubiquitously expressing TIR1 and the indicated core subunit endogenously tagged with degron (AID), shown during ethanol exposure as a control, and after 1, 3 and 6 h of 1 mM auxin exposure. Insets: zoom of single nucleus of the intestine. Scale bar, 2.5 μm. **b**, Quantification of the percentage of the indicated AID-tagged animals (RNA Pol I, Pol II and Pol III) within the indicated categories for 3D chromatin organization in intestine and timepoint of auxin exposure. Data are shown as mean ± s.e.m. of three independent biological replicas. **c**, Line plots of the averaged single

nuclei profile of Hoechst 33342-stained DNA from 72 intestinal nuclei of the indicated fed, degron-tagged animals in presence of ethanol as control or 3 h on 1 mM auxin, from three independent biological replicas, showing the radial fluorescence intensity as a function of the relative distance from the nucleolus centre. The shaded area represents the 95% confidence interval of the mean profile. For RNA Pol I- and Pol II-AID at 3 h of auxin exposure, animals in proportions to their relative abundance within the 3D chromatin organization categories as in **b** were analysed. **d**, Line plot as in **c** but from 70 intestinal cells of fed WT and *raga-1^GF* animals expressing RNA Pol I degron-tagged (RNA Pol I-AID) treated with 1 mM auxin for 5 h. Data are from three independent biological replicas. For **c** and **d**, heat maps displaying the single nuclei profiles are provided in Extended Data Figs. 6c,n, respectively. Source numerical data are available in Source data.

Pol II inhibition, with concentric rings of chromatin being detected in about 2–5% of animals, depending on the timepoint (Fig. 6b). Still, the average chromatin organization was largely unaffected by RNA Pol II inhibition (Fig. 6c and Extended Data Fig. 6c).

RNA Pol II transcription is reduced when RPB-2 is depleted (Extended Data Fig. 6d,e). Nonetheless, we decided to confirm the results by specifically inhibiting this polymerase in fed animals using α-amanitin and found that chromatin distribution remained fed-like (Extended Data Fig. 6f–h). In contrast, exposure to actinomycin D, a

broad transcriptional inhibitor with a higher affinity for blocking rRNA synthesis by RNA Pol I (refs. 64,65), induced a fasting-like reorganization of chromatin in intestine of fed animals (Extended Data Fig. 6i–k).

We conclude that transcription by RNA Pol I is necessary to maintain the chromatin architecture typical of the fed state in intestinal cells as its inhibition, and not that of RNA Pol II nor III, is sufficient to induce a fasted-like chromatin reorganization in the intestine of fed animals.

The *raga-1^GF* mutants fail to counteract the formation of the concentric rings induced by Pol I degradation in fed animals (Fig. 6d and

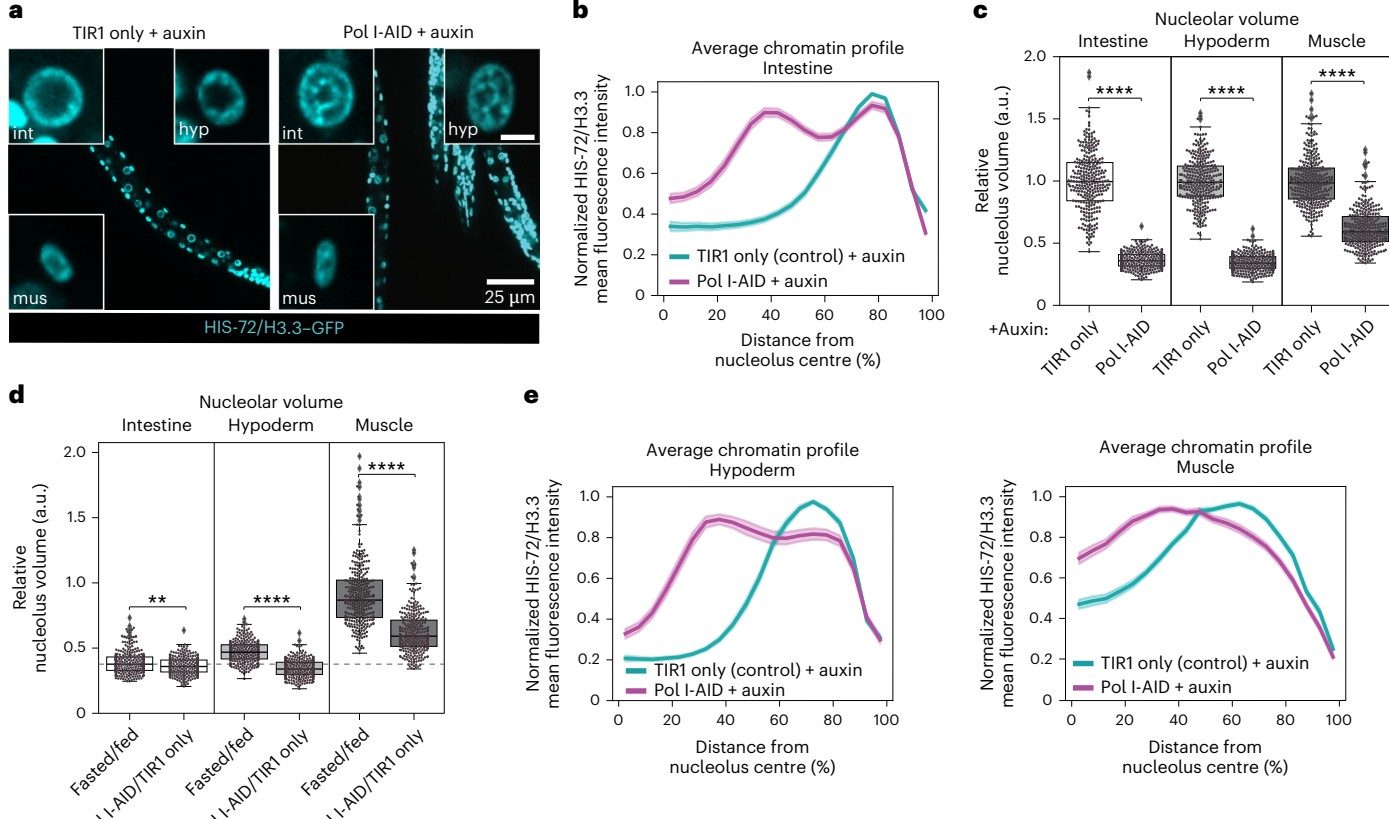

**Fig. 7 | RNA Pol I depletion is sufficient to trigger the formation of chromatin rings in hypoderm of fed animals. a**, Single focal planes of representative fed L1 larvae expressing HIS-72/H3.3–GFP, ubiquitous TIR1 alone (TIR1 only) or with RPOA-2-AID (Pol I-AID), treated with 1 mM auxin for 8 h. Insets: zoom of single nucleus of the indicated tissue. Hyp: hypoderm; int: intestine; mus: muscle. Inset scale bar, 2.5 μm. **b**, Line plot of the averaged single nuclei profile of HIS-72/H3.3–GFP from 72 intestinal nuclei of fed, Pol I-AID or TIR1 only as control, in the presence of 1 mM auxin, showing the radial fluorescence intensity as a function of the relative distance from the nucleolus centre. The shaded area represents the 95% confidence interval of the mean profile. Data are from three independent biological replicas. **c**, Box plot showing relative changes in nucleolar volume, measured using FIB-1–mCherry, in intestine, hypoderm and muscle cells of Pol I-AID animals, compared with control TIR1 only, in the corresponding tissue upon treatment with 1 mM auxin for 8 h. **d**, Box plot as in **c** but comparing fasted over fed animals and Pol I-AID over TIR1 only animals (upon 1 mM auxin exposure for 8 h), in the indicated tissues. The dotted grey line represents the median reduction in nucleolar volume in fasted intestine over fed. **e**, Line plots as in **b** but in 72 hypodermal and muscle nuclei from three independent biological replicas. Single-peak profiles were compared to estimate the statistical significance of differences, as described in Methods. *P* values are given in Supplementary Table 2. For **b** and **e**, heat maps displaying the single nuclei profiles are provided in Extended Data Fig. 7d. For **c** and **d**, box limits are 25th and 75th percentiles, whiskers denote 1.5 times the interquartile ranges, points outside the whiskers are outliers and the median is shown as a line. Probability values from two-sided Wilcoxon rank sum tests are shown: ***P* < 0.01 and *****P* < 0.0001. Data are from three independent biological replicas. See *P* values and *n* in Supplementary Table 2. Source numerical data are available in Source data.

Extended Data Fig. 6m,n). As RAGA-1^GF expression is unchanged after few hours of transcriptional inhibition (Extended Data Fig. 6l), this shows that RNA Pol I transcription acts downstream of mTORC1 in the regulation of the 3D chromatin architecture by nutrients.

During fasting, the nucleolus shrinks more in the intestine compared with hypoderm and muscle (Fig. 4d) and chromatin rings form only in the intestine (Fig. 1a–c and Extended Data Fig. 1c,d). This suggests that the fasting-induced decrease in RNA Pol I activity in these tissues may not be sufficient for chromatin to reorganize into concentric rings. This led us to investigate how a direct acute inhibition of RNA Pol I would affect nucleolar size and chromatin organization in hypoderm and muscle cells. To this aim, we used RPOA-2-AID worms and switched to monitoring HIS-72/H3.3–GFP, as the live staining of DNA labels only intestinal cells. To be able to simultaneously image the nucleolus in red and histones in green, we used a different *tir1* allele. In this experimental setting, a complete depletion of RPOA-2 from intestinal cells and whole larvae occurred after 8 h on auxin (Extended Data Fig. 7a,b), along with a reduction in pre-rRNA levels for all tissues tested, confirming RNA Pol I inhibition (Extended Data Fig. 7c). As expected, in intestinal cells of fed

animals exposed to auxin for 8 h, chromatin rings are formed (Fig. 7a,b and Extended Data Fig. 7d) and nucleolar size is strongly reduced (Fig. 7c). Next, we monitored the size of the nucleolus in hypoderm and muscle and found that for both it decreases more than during fasting (Fig. 7d), with the reduction in nucleolar volume in hypoderm, but not in muscle, now aligning to that observed in intestine both during fasting and Pol I inhibition (Fig. 7c,d). Strikingly, we observed the formation of chromatin rings in hypoderm but not in muscle (Fig. 7a,e and Extended Data Fig. 7d), even when analysing only the longest radial distances (Extended Data Fig. 7e,f). This shows that the ability to rearrange the 3D genome into chromatin rings is not unique to intestinal cells. However, during fasting, this chromatin reconfiguration appears to be an intestine-specific phenomenon, probably because of an insufficient inhibition of RNA Pol I in other tissues, such as hypoderm.

## Specific genes are upregulated when chromatin rings form

The 3D genome architecture is closely linked to gene expression regulation[11]. Thus, to begin exploring whether the reorganization induced by fasting influences gene expression, we performed RNA

sequencing (RNA-seq) in dissected intestines from fed and 12 h fasted adults (Extended Data Fig. 8a,b). We found that ~16% of all genes are differentially expressed, with 2,982 upregulated and 351 downregulated (Fig. 8a). This is consistent with the notion that lack of food induces a complex response involving multiple pathways[66], many of which are likely to regulate gene expression independently of the observed 3D chromatin reorganization. Thus, a first step to identify genes that might be sensitive to the reorganization of the intestinal chromatin into two rings is to uncouple the configuration of the 3D genome from the nutritional status of the animal and quantify gene expression changes (Fig. 8b). To this aim, we performed RNA-seq in intestinal cells of fed adults where chromatin rings are induced by AID-mediated RNA Pol I inhibition (Extended Data Fig. 8c–e). In particular, we monitored gene expression in the intestine of TIR only as a control, and Pol I-AID animals at three different timepoints of auxin exposure: 0, 12 and 24 h (Extended Data Fig. 8f,g). As expected, without auxin exposure (0 h), Pol I-AID and TIR1-only expressing animals show virtually no difference in gene expression. In contrast, after 12 and 24 h on auxin, loss of Pol I induces gene expression changes, with a stronger effect after 24 h (316 genes upregulated and 285 downregulated; Fig. 8c). Genes that are changed by Pol I depletion after 24 h on auxin have the tendency to be already altered in the same direction (up- or downregulated) after 12 h, albeit to a lesser degree (Fig. 8d–f), revealing that the changes induced by RNA Pol I inhibition are progressive. Next, we compared the Pol I degradation-induced gene expression changes with those occurring in fasted animals. Strikingly, the majority of genes overexpressed in response to Pol I inhibition in fed animals were also upregulated in intestinal cells during fasting (Fig. 8d,f and Extended Data Fig. 9a). In contrast, most genes that are downregulated by Pol I depletion after 24 h on auxin are not repressed during fasting (Fig. 8e,f and Extended Data Fig. 9a). Interestingly, the reverse analysis yielded similar results: genes that are upregulated during fasting (Fig. 8a) tended to be upregulated also in Pol I-depleted animals (Fig. 8g), while genes with reduced expression during fasting were overall not downregulated in absence of Pol I activity (Fig. 8g). Thus, in the two different conditions where the intestinal genome is reorganized into chromatin rings, we observed the upregulation of an overlapping set of genes. Whether their expression is regulated by RNA Pol I activity alone or through the reorganization of the 3D genome remains to be determined.

Both fasting (Extended Data Fig. 6a) and Pol I inhibition (Extended Data Fig. 7c) affect pre-rRNA levels, potentially influencing the expression of genes involved in ribosome biogenesis, nucleolar structure and translation. To test whether these genes were the drivers of the overlap detected between the two conditions, we selected them using four Gene Ontology (GO) categories (ribosomal, rRNA processing, nucleolus and translation-related genes) and examined their expression in the intestine of both Pol I-depleted and fasted animals. Interestingly, all categories are upregulated in fasted intestinal cells, which aligns

with prior studies[67], but none is overexpressed when Pol I is inhibited (Fig. 8h and Extended Data Fig. 9b), indicating that, overall, these genes are sensitive to lack of food but not to the formation of chromatin rings. Intriguingly, we found that, instead, genes that are upregulated when rings are formed either in fed animals lacking RNA Pol I or during fasting are enriched in GO categories related to metabolism and stress response (Fig. 8i).

## Discussion

Connections between cellular metabolism and chromatin architecture have been studied primarily in yeast[68] or in isolated cells or tissues[69]. In this study, we describe how the nutritional state alters the large-scale spatial organization of the genome of a living multicellular organism.

By using live confocal microscopy in *C. elegans*, we discovered that fasting triggers a reversible (Fig. 5a,b) and large-scale 3D genome reorganization specifically in intestine (Fig. 1a–c).

On the basis of our results, we propose that the 3D genome architecture of intestinal cells is modulated by mTOR signalling in response to nutrients acting through the regulation of RNA Pol I, which transcribes the ribosomal DNA in the nucleolus (Fig. 8j). We find that RNA Pol I has a critical and unique role in shaping 3D genome architecture: transcription by RNA Pol I, but not Pol II or Pol III, is essential to reverse the chromatin rings in the intestine upon refeeding (Fig. 5e–g). Moreover, its inhibition, but not that of Pol II or III, in fed animals is sufficient to mimic fasting and induce the two chromatin rings organization not only in intestine (Fig. 6a,c), but also in hypoderm (Fig. 7a,e), a tissue where this 3D chromatin configuration is not observed during fasting.

In agreement with a critical role of RNA Pol I activity in 3D chromatin reorganization, the nucleolus shrinks during fasting, before rings are formed in the intestine (Fig. 4c), and it enlarges rapidly upon refeeding before a fed-like 3D genome architecture is re-established (Fig. 5c). Furthermore, in fasted animals the volume of the nucleolus is more strongly reduced in intestine compared with hypoderm and muscle, which do not reorganize their chromatin into rings (Fig. 4d). However, during fasting, pre-rRNA levels are reduced to a similar degree in the tissues analysed (Extended Data Fig. 6a). This suggests that rather than RNA Pol I transcription itself, it is the remodelling of nucleolar structure, particularly a drastic drop in size following RNA Pol I inhibition, that might be critical for 3D genome reorganization in response to nutrient deprivation. Accordingly, deleting rDNA repeats in *Drosophila melanogaster* leaves the steady-state concentration of rRNA unaltered but remodels the nucleolus and compromises heterochromatic silencing in other sites of the genome[70].

During fasting (Fig. 4d), mTOR inhibition (Fig. 4e) and upon RNA Pol I inhibition (Fig. 7c), the volume of the nucleolus in muscle cells is only weakly decreased, suggesting that nucleolar remodelling can be regulated tissue specifically in response to the same stimulus, potentially explaining the inability of muscle cells to form chromatin

**Fig. 8 | Specific genes are upregulated when chromatin rings form.**
**a**, Representative scatter plot comparing the relative gene expression FCs in intestinal cells of WT fasted animals compared to fed in two different biological replicas (rep). In red and in blue are the upregulated (UP) and downregulated (DOWN) genes, respectively (FDR <0.01 and 1.5 FC, $N = 4$). **b**, A scheme of the rational of the RNA-seq experiment in **a** and **c**–**i**. **c**, Scatter plots as in **a** but in intestines of animals expressing RPOA-2-AID (Pol I-AID) compared with control (TIR1 only), treated with 1 mM auxin for the indicated time. Significant up- and downregulated transcripts are highlighted in red and blue, respectively (FDR <0.05 and 2.0 FC, $N = 4$). **d**, Heat map of the expression changes of the genes that are significantly upregulated in the intestine of Pol I-AID after 24 h on auxin (encircled with a black square) in the intestine of Pol I-AID at 0 and 12 h on auxin and of fasted animals. **e**, Heat map as in **d** but of genes that are downregulated in the intestine of Pol I-AID after 24 h on auxin. **f**, Line plots of z-score values to measure the relative expression changes of genes that are upregulated (top) or downregulated (bottom) in Pol I-AID (24 h on auxin), in Pol I-AID at 0 and 12 h

on auxin and in the intestine of fasted animals. Data are shown as mean ± s.d. of four biological replicas. **g**, As in **f** but for genes that are upregulated (top) or downregulated (bottom) upon fasting. **h**, Comparison of the relative expression of genes belonging to the GO category of 'ribosomal' or 'rRNA processing' in the intestine of Pol I-AID, treated with 1 mM auxin for the indicated time or in WT fasted animals. Data are shown as mean ± s.d. of four biological replicas. **i**, GO term enrichment for genes upregulated both in Pol I-depleted (24 h on auxin) and fasted animals. ncRNA, non-coding RNA; tRNA, transfer RNA. Significance was calculated using the hypergeometric test, adjusted for multiple testing with the default g:SCS (set counts and sizes) method integrated into the gprofiler2 R package. **j**, A model showing that in fed animals, mTORC1 promotes transcription by RNA Pol I. During fasting, mTORC1 inactivation represses RNA Pol I transcription leading to a drastic reduction in nucleolar size in intestinal cells, thus promoting a 3D reorganization of the genome, which becomes enriched at the nuclear and nucleolar periphery. This 3D chromatin configuration might support the upregulation of a set of metabolic and stress-related genes.

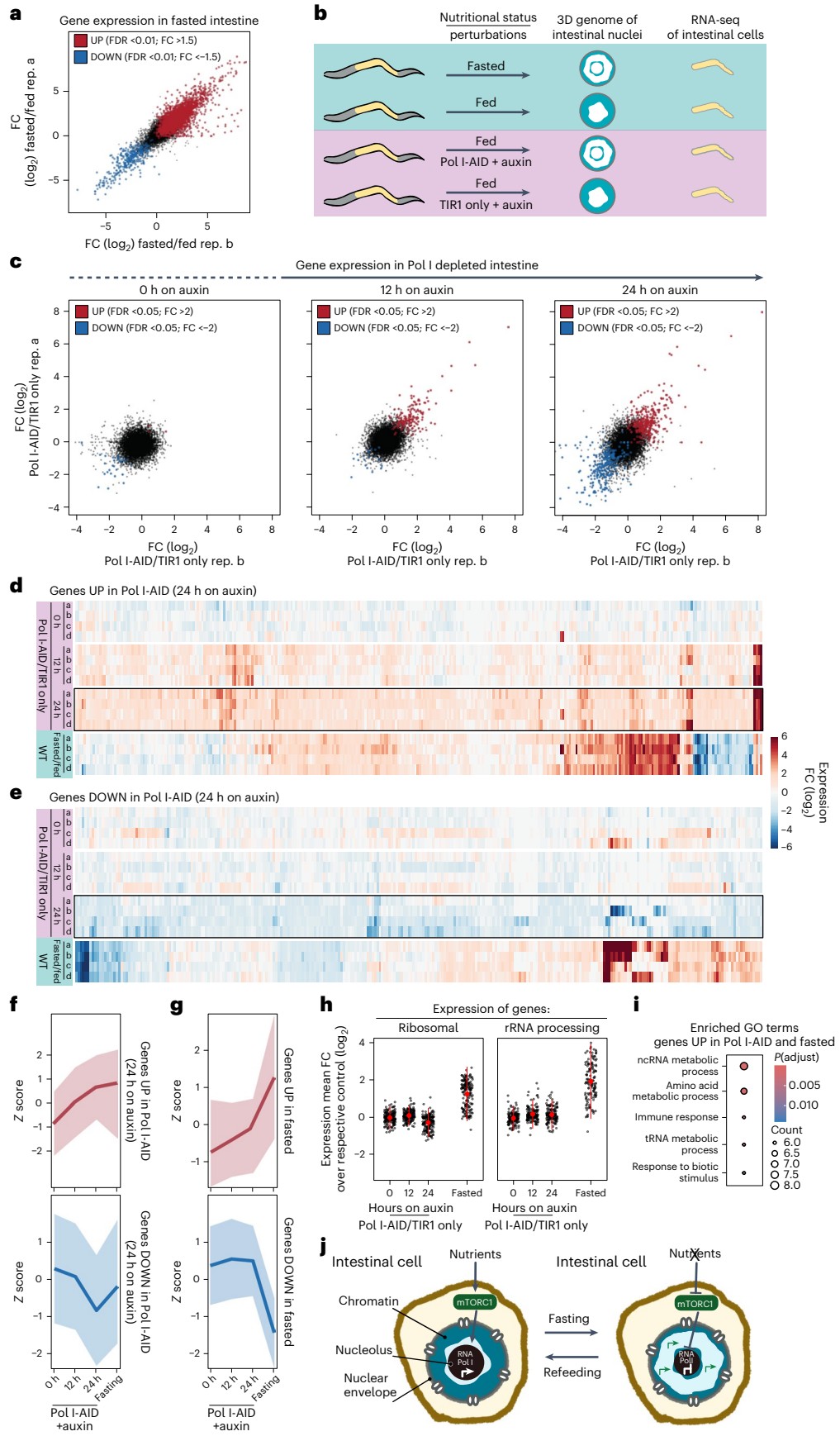

rings. Alternatively, the relatively small size of muscle nuclei (~4 times smaller than intestinal nuclei), might counteract ring formation. Further confirming that nucleolar structure is regulated tissue specifically, vacuole-containing nucleoli are prominent in intestine but not in hypodermal cells at the L3–L4 stage in *C. elegans* and their formation is promoted by an alternative rRNA processing pathway[71]. This suggests that, like nucleolar size, the processing of rRNA varies across cell types and might be implicated in 3D genome organization in response to nutrients.

With live microscopy experiments, we showed that chromatin rings form at the nuclear and nucleolar periphery (Fig. 2a,b), well-known heterochromatic compartments[11]. Yet, the depletion of canonical heterochromatic marks, namely, H3K9me and H3K27me3, leaves the fasting-induced 3D genome reconfiguration unchanged (Fig. 2c), suggesting that other mechanisms are involved. Rapamycin-induced inhibition of the mTOR pathway increases core histone expression in the intestine of *D. melanogaster*[72]. Instead, in *C. elegans*, prolonged starvation leads to a global degradation of histone H2Bs[73]. The tissue-specific dynamics of histone abundance in the early stages of fasting is not known, and its investigation might help to shed light on the mechanism of chromatin ring formation.

RNA pol I depletion has been recently reported to increase, more than decrease, chromatin accessibility[74] and, although the effect on absolute transcriptional output remains to be determined, our RNA-seq data revealed that the formation of chromatin rings during fasting and upon Pol I depletion coincides with specific genes being upregulated rather than repressed (Fig. 8d–g). Whether the observed changes in gene expression are driven by RNA Pol I inhibition, the 3D genome reconfiguration or both remains to be determined. Nonetheless, our data provide a basis to further explore the regulation of RNA Pol II targets by a 3D chromatin configuration that is modulated by RNA Pol I activity.

There is growing evidence that environmental cues in the form of diet and lifestyle contribute to the onset of metabolic diseases in humans via epigenetic mechanisms[75]. While a role for chromatin architecture in this process remains undetermined, our work unveils that nutritional stimuli from the environment alter the spatial organization of the genome, adding an additional layer of complexity to the regulation of genome architecture, which may contribute to health and diseased states.

## Online content

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

## Methods

### *C. elegans* maintenance and strains

Nematodes were grown with *Escherichia coli* OP50 bacteria on nematode growth medium (NGM) agar plates at 20 °C except where otherwise stated. All strains used in this study are listed in Supplementary Table 1.

### Constructs and strains

Endogenously tagged *rpc-1* at the C-terminus with STSGGSGGTGGS-mNeonGreen-GSAGSA-degron was obtained by CRISPR–Cas9 from the company SunyBiotech.

For the HIS-1–GFP fusion construct, the *his-1* gene was amplified from N2 worm genomic DNA and fused by PCR to GFP, which contained introns. The *ges-1* promoter and *unc-54* 3′ untranslated region were amplified from N2 genomic DNA. The final plasmid construct was generated by MultiSite Gateway cloning (Invitrogen). The strain expressing HIS-1/H4–GFP was made using the Mos1-mediated single-copy insertion technique[76]. The transgene was inserted into ttTi5605 on chromosome II.

The *degron–GFP* tagged *rpoa-2* allele was constructed as described[77] using Cas9 protein driven by *eft-3* promoter in pDD162 and genomic RNA targeting a genomic sequence in the N-terminus of *rpoa-2* in pRR13, a derivative of pRB1017, an empty vector for gRNA cloning. The *degron–GFP-c1^sec^3xflag* repair template was constructed for generating the knock-in into the N-terminus of the *rpoa-2* gene. The 5′ and 3′ homology arms 751 bp upstream of the *rpoa-2* start codon and 566 bp downstream of the start codon were used to replace the *ccdB* cassette in the *degron–GFP-c1^sec^3xflag* repair template. Each knock-in was isolated via hygromycin selection and the self-excising cassette (SEC) was then excised by heat shock to produce *degron::GFP::rpoa-2* strain.

Information on the hetero- and euchromatic reporters used in Fig. 2 and Extended Data Fig. 2. In *C. elegans*, integrated transgene arrays acquire different chromatin marks and subnuclear locations, based on their size. Especially in embryos, large arrays that are composed of 300–500 plasmid copies are 'heterochromatinized' by the deposition of H3K9me3 and H3K27me3, peripherally located and silenced, while the same sequence, if present in less than 50 copies, is not[78]. The heterochromatin reporter used in Extended Data Fig. 2f–h, is a large array composed of about 300 copies. As cells differentiate, arrays carrying tissue-specific promoters shift inwards from the nuclear periphery exclusively in the tissue in which they are active, regardless of their copy number[19]. The euchromatic reporters used in Fig. 2d–f and Extended Data Fig. 2i–k are small transgene of about 30 copies[19] driven by the *pha-4* promoter, which is actively transcribed in intestine. Accordingly, these reporters are internally positioned in fed intestinal cells (ref. 19 and data in this study). The GW429 strain[19] was created by ballistic transformation, generating a rare integration event of about 30 copies of the *pha-4::mCherry* plasmid. The 256 copies in the genotype refers to the copies of *LacO* repeats carried by the cointegrated plasmid to enable visualization of the allele by GFP–LacI.

### Feeding, fasting and refeeding

Worms were maintained well fed on OP50 at 20 °C for at least two generations.

Fed L1 larvae (L1s) were obtained by washing plates of mixed-stage animals twice with M9 buffer to remove adults and larvae. The washes were performed with gentle swirling to avoid removing the bacteria. To obtain synchronized L1s, the embryos remaining on the plate were allowed to hatch for 2 h.

For L1s fasting, embryos were isolated by standard hypochlorite treatment and maintained in M9 buffer on a roller at 20 °C. L1s hatch approximately 12 h after hypochlorite treatment (ref. 79 and our own observation), and hence, this timepoint was considered 0 h of fasting. Consequently, 12 h of fasting corresponds to 24 h after hypochlorite treatment.

Refeeding was performed by placing 12 h fasted L1s on OP50-seeded NGM plates for the indicated time, after which they were collected for imaging.

To study the kinetics of chromatin reorganization during fasting, synchronized L1s were obtained as described above for fed L1s. They were then washed three times with M9 buffer for 10 min each and left in M9 buffer for the indicated time on a roller at 20 °C.

For fasting of adults, synchronized L1s by hypochlorite treatment were grown on OP50-seeded plates until day 1 of adulthood. Next, adults were collected with M9 buffer. A fraction of worms was immediately imaged for the fed state or dissected for intestine-specific RNA-seq, the rest were washed three times with M9 buffer for 10 min and fasted in M9 buffer for 12 h on a roller at 20 °C prior to imaging or intestine dissection.

For feeding in liquid, to obtain fed and synchronized L1s that were maintained in liquid for the same duration as the fasted animals, embryos obtained from hypochlorite treatment were kept in M9 buffer for 21 h. Next, L1s were pelleted, M9 buffer was removed and S-basal complete medium supplemented with 6 mg ml⁻¹ *E. coli* OP50 was added. L1s were subsequently fed in liquid culture for 3 h before imaging, reaching a total of 24 h in liquid.

For fasting on plates, isolated embryos obtained by hypochlorite treatment were placed onto M9 agarose plates without any bacteria for 24 h so that L1s were fasted for an average of 12 h as described above for fasting in liquid.

### Auxin stock and plates

Auxin 3-indoleacetic acid (Sigma-Aldrich) was dissolved in ethanol to prepare a 57 mM stock solution and stored at 4 °C. Auxin was added to NGM plates to a final concentration of 1 mM. Control plates contained an equivalent amount of ethanol (1.75%). Plates were then seeded with OP50 bacteria.

### Auxin treatment

For all degron-tagged strains, the *tir1* allele *ieSi57* was used except for experiments shown in Fig. 7a–e and Extended Data Fig. 7a–f, where the *tir1* allele *wrdSi23* was used.

**Degradation of polymerase subunits.** Degradation of polymerase subunits in strains expressing RPB-2-AID, RPC-1-AID and RPOA-2-AID was obtained as follows: fed L1s were obtained as described above and added onto auxin or ethanol plates containing Hoechst 33342-stained OP50 bacteria. L1s were maintained on auxin plates for the indicated amount of time, and on ethanol plates for 3 h, before imaging.

For RPOA-2 degradation in animals expressing HIS-72/H3.3–GFP, fed L1s were collected as described above and added onto auxins plates containing unstained OP50 bacteria.

To quantify the effect of RPB-2 depletion on the transcriptome (Extended Data Fig. 6e), animals were grown on OP50 at a density of approximately 40 animals per plate. Animals were maintained at 25 °C, which rendered the population sterile, and exposed to 375 µM of auxin (α-naphthaleneacetic acid) starting at age 2 of adulthood. They were collected into lysis buffer for sequencing[80] on day 6, 96 h post-auxin treatment.

For the refeeding experiments in absence of the polymerases core subunits, embryos were isolated using hypochlorite treatment and left in M9 buffer to allow L1s to hatch in absence of food. After a fasting period of 10 h (22 h post-bleaching), an aliquot of L1s was DNA live stained to verify the formation of the chromatin rings in intestine by microscopy, as follows: 10 h fasted L1s were incubated in M9 supplemented with 1 mM auxin and 40 µM Hoechst 33342 for 2 h in the dark, then L1s were imaged. To the remaining 10 h fasted L1s, auxin was added at a final concentration of 1 mM in M9 buffer. After 2 h on auxin (12 h of fasting), L1s were transferred to auxin plates seeded with OP50 bacteria stained with Hoechst 33342. Next, L1s were fed for 30 min and imaged.

For RPOA-2 degradation in adults, synchronized L1s by hypochlorite treatment were grown on OP50-seeded NGM plates for 72 h until day 1 of adulthood. Next, they were picked from the plates and transferred onto auxin plates seeded with OP50 and maintained for the indicated time.

**Degradation of DAF-15.** Embryos isolated by hypochlorite treatment were plated onto auxin, or ethanol plates as control, seeded with OP50 bacteria for 21 h. To stain the DNA, hatched L1s were moved onto auxin (or ethanol plates) seeded with Hoechst 33342-stained OP50 for 3 h. The absorption of auxin is not efficient in embryos due to their eggshell[81]. Thus, because L1s hatch approximately 12 h after hypochlorite treatment[79], we considered our total 24 h of auxin treatment as embryos and L1s to correspond to 12 h of auxin exposure as L1s. Degradation of DAF-15-AID upon auxin treatment was previously performed and found to require 1 h on 1 mM auxin[35].

### Heat and cold stress
Fed L1s were placed onto standard NGM plates with OP50 and were subjected to either a 6 h heat shock at 34 °C or a 6 h cold shock at 6 °C in an air incubator.

### Osmolarity
The final osmolarity of a solution was calculated by summing the osmolarity contributed by each solute present in the solution. To determine the osmolarity of each solute, the molarity of the solute was multiplied by the number of osmoles it produces when dissolved in water. M9 buffer was calculated to be 360 mOsm. To achieve different osmolarities, M9 buffer was diluted in MilliQ water to obtain buffers with osmolarities of 180, 150 and 120 mOsm. To create a solution of 540 mOsm, NaCl (90 mM) was added to M9.

### Actinomycin D and α-amanitin treatments
A 25 mg ml$^{-1}$ stock solution of actinomycin D (Bioaustralis) was prepared in dimethylsulfoxide. For the treatment plates, actinomycin D was diluted into NGM medium to achieve a final concentration of 100 µg ml$^{-1}$. Control plates were prepared with an equivalent amount of dimethylsulfoxide (0.4%). The plates were seeded with OP50 bacteria and the bacteria were subsequently stained with Hoechst 33342 dye. Fed L1s were placed onto the plates and imaged after 6 h.

A 2 mg ml$^{-1}$ stock solution of α-amanitin (Th. Geyer) was prepared in water. For the treatment plates, α-amanitin was diluted into NGM medium to achieve a final concentration of 25 µg ml$^{-1}$. Control plates were prepared with an equivalent amount of water (1.25%). The plates were seeded with OP50 bacteria. Fed L1s were placed onto the plates and imaged after 6 h.

### RNAi experiments
RNAi experiments were performed at 20 °C on NGM agar plates supplemented with 100 µg ml$^{-1}$ carbenicillin and 400 µM isopropyl-β-D-thiogalactoside. Plates were seeded with double-stranded RNA-producing bacteria, grown overnight at 37 °C. As negative control, the RNAi clone containing the empty L4440 vector (Fire vector library) was used. In every experiment, *let-607* RNAi was performed in parallel to confirm the efficiency of the RNAi plates, as a phenotype of very delayed growth/lethality is expected compared with worms grown on L4440 RNAi. RNAi clones were obtained from Ahringer library, Source BioScience Ltd. All RNAi clones used were sequenced to confirm target specificity.

For *let-363* knockdown, synchronized L1s were seeded on RNAi plates and grown for 72 h. They were then transferred to freshly made RNAi-seeded plates. The following day, worms were washed off and discarded, leaving only laid embryos on the RNAi plates. After 2 h, hatched L1s were collected, washed three times, filtered with a 5 µm syringe strainer (PluriSelect) and seeded on fresh RNAi plates containing the respective RNAi bacteria. After 12 h, L1s of *let-363* were collected for live imaging/RNA fluorescence in situ hybridization (FISH). To obtain stage-matched larvae, L1s seeded on the L4440 control were maintained on plates for 5 h before collection.

For *mpk-1* knockdown, embryos isolated by standard hypochlorite treatment were placed and grown on RNAi plates until adulthood (P0). Next, the progeny of this RNAi-fed population was obtained by hypochlorite treatment of the adults and fasted for 12 h (24 h post-bleaching), before imaging.

For *mes-6* and *met-2*, fasted animals were obtained as described for *mpk-1* but fed L1s were also imaged in parallel by collecting the larvae that hatched on the RNAi plates.

For *mes-6*, RNAi efficiency was verified by incubating L1s on *mes-6* RNAi plates for two generations, which produced 50–60% F1 adult sterility, as expected given the role of this gene in the germline[82]. For *mpk-1*, RNAi efficiency was verified by monitoring the amount of progeny of the RNAi-fed P0 and observing about 75% reduction, as expected given the role of this gene in the germline[83].

### Live staining of DNA
To prepare the staining solution, Hoechst 33342 (ChemCruz, 20 mM stock) was diluted 1:1,000 in M9 buffer, resulting in a final concentration of 20 µM. Next, the staining solution was added on OP50 bacteria, previously seeded and dried on NGM plates, making sure that all bacteria are covered. Next, the plates were left to dry in the dark.

To perform DNA live staining in fed larvae, L1s were obtained by washing plates of mixed stages and waiting 2 h for embryos to hatch. Hatched L1s were collected and added to these plates directly on the food and allowed to feed for 3 h on Hoechst 33342-stained bacteria.

Staining of the intestinal DNA in fasted worms was achieved by first feeding L1s with Hoechst 33342-stained bacteria for 3 h, as described above. Next, L1s were collected using M9, washed three times in M9 for 10 min to remove the bacteria and subsequently fasted in M9 buffer for 6–10 h.

### Staining of DNA in fixed larvae
In Extended Data Fig. 2d, left, fasted L1s were fixed with 4% paraformaldehyde in PBS for 5 min and washed with PBS containing 30 mM glycine (PBS-G) for 10 min at room temperature. Next, worms were permeabilized with acetone at −20 °C for 1 min and subsequently washed with PBS-G. For staining, a solution was prepared by adding two drops of EasyProbe-Hoechst 33342 Live Cell Stain (GeneCopoeia) into 1 ml of M9 buffer. Next, the fixed worms were incubated in this staining solution for 15 min. After staining, a final 5 min wash with PBS-G was performed before preparing the samples for imaging on an agarose pad as described above but without sodium azide. In Extended Data Fig. 2d, right, fasted L1s were spotted on poly-L-lysine covered slides and allowed to settle for few minutes. Next, coverslips were applied and slides were snap frozen on dry ice for 25 min. To permeabilize worms, samples were then freeze-cracked by flicking the coverslips and immersed immediately in 100% ice-cold methanol for 10 s. Next, the slides were transferred to a fixing solution (0.08 M HEPES pH 6.9, 1.6 mM MgSO$_4$, 0.8 mM EGTA and 3.7% formaldehyde, in PBS) for 10 min. After fixation, slides were washed three times with TBS-T (TBS with 0.1% Tween-20). To visualize DNA, samples were incubated with 10 µM Hoechst 33342 (ChemCruz) for 2 h, washed once with TBS-T, mounted with 80% glycerol in PBS and imaged.

### rRNA FISH
To detect pre-rRNAs, we ordered six FISH probes targeting the internal transcribed spacer region 1 (*its-1* in worms) labelled with fluorescent ATTO 488 at the 5′ end from Integrated DNA Technologies. Probe sequences are listed in Supplementary Table 3. To make the stock solution, each probe was dissolved in RNase-free H$_2$O to a concentration of 1 µg µl$^{-1}$. Next, 5 µl of each probe was pooled together and RNase-free

H$_2$O was added to reach a dilution of 1:12 for each probe (30 µl probes mix + 30 µl H$_2$O). This stock solution was then diluted 1:500 in a hybridization buffer containing 2× SSC, 10% formamide, 5% dextran sulfate and 2 mM Ribonucleoside Vanadyl Complexes, to obtain the hybridization mix. Right before use, the hybridization mix was denatured for 5 min at 80 °C and incubated of at least 20 min at 37 °C.

Worms were collected and washed twice with PBS containing 0.01% Triton X-100 for 5 min each on a roller. They were then transferred to 1.5 ml tubes and pelleted at 1,500g for 3 min. The supernatant was aspirated, and 1 ml of fixation buffer (3.7% formaldehyde solution, 1× PBS and 0.01% Triton X-100) was added. The samples were incubated with fixation buffer for 45 min at room temperature. Afterwards, the samples were pelleted again, the fixation buffer was removed and the samples were washed twice with 1 ml of PBS containing 0.01% Triton X-100. Next, 1 ml of 70% ethanol was added and the samples were incubated overnight at 2–8 °C.

The next day, ethanol was removed after pelleting the samples and the fixed worms were rehydrated with PBS containing 0.01% Triton X-100 for 5 min at room temperature. The PBS was then removed, and 1 ml of permeabilization buffer (PBS with 0.5% Triton X-100) was added and incubated for 5 min at room temperature. After removing the permeabilization buffer, 1 ml of equilibration buffer (2× SSC, 10% formamide and 0.01% Triton X-100) was added and incubated for 15 min at room temperature. The equilibration buffer was then removed and the worms were incubated in the dark with hybridization buffer containing the final probe mix solution for at least 16 h at 42 °C while shaking at 1,200 r.p.m. in a thermal shaker.

The following day, after two brief washes with 2× SSC, the worms were incubated with wash buffer (1× SSC, 10% formamide and 0.01% Triton X-100) twice for 5 min each at 42 °C. This was followed by an incubation with Hoechst (1:2,000 in 1× PBS) for 30 min at room temperature. The worms were then washed again with wash buffer for 5 min at 42 °C and transferred to a poly-L-lysine-coated slide. To ensure proper adhesion to the surface, they were left to dry for a few minutes. Finally, the samples were mounted on microscope slides with Vectashield Plus Antifade Mounting Medium (Biozol), coverslips were applied and the edges were sealed with nail polish.

## Microscopy

Microscopy was carried out using a live-cell imaging system (Confocal Spinning Disk Microscope) from Visitron Systems GmbH, equipped as follows: Nikon Eclipse Ti2 microscope with a Plan apo λ 100×/1.45 oil objective, Plan apo λ 60×/1.40 oil objective, Yokogawa CSU-W1 confocal scanner unit, VS-Homogenizer, Electron Multiplying CCD camera (Andor iXon Series) and VisiView software for acquisition.

Live microscopy was carried out on 2% agarose pads supplemented with 0.15% sodium azide (Interchim) to paralyse the worms, as previously described[19]. All Images were acquired with the Plan apo λ 100×/1.45 oil objective except for adults which were acquired with the Plan apo λ 60×/1.40 oil immersion objective. For each image, a range of stacks, from 50 to 120 stacks depending on the stage, were captured with a z-spacing of 200 nm.

## Image analyses

**Fluorescence intensities.** Fluorescence intensities of GFP and mNeon-Green were measured using Fiji/ImageJ[84,85].

For the polymerase subunits degradation experiments shown in Extended Data Figs. 5a–f and 7a, for each strain, images of the different treatments were acquired with the same settings. Intestinal nuclei were selected as region of interest based on Hoechst 33342 signal. Within each nucleus, quantitation of GFP or mNeonGreen mean signal intensity on focal stack images was done by selecting the brightest plane and subtracting the average background of the corresponding image.

For the analysis of RPOA-2-AID-GFP in whole larvae (Extended Data Fig. 7b), z-stack images were sum projected. Worms were selected as the region of interest using the freehand tool and the mean GFP signal intensity was measured. Background intensity was measured in animals expressing only TIR1 and subtracted from the GFP mean signal intensity.

For the RAGA-1$^{GF}$–GFP experiment (Extended Data Fig. 6l), z-stack images were sum projected. Quantitation of the GFP mean signal intensity on the projected images was done by selecting the worms as the region of interest using the freehand tool and subtracting the average background intensity from the corresponding image.

**Worm size measurements.** For DAF-15 degradation, the experiment was conducted as described in 'Auxin treatment' section. The first images were acquired 24 h after plating the embryos on auxin and ethanol plates, which was considered day 1. For *let-363* knockdown, the experiment details are described in 'RNAi experiments' section. Briefly, the first images were acquired immediately after transferring L1 larvae of the F1 generation onto fresh RNAi plates containing the RNAi bacteria. This was considered day 1. Images were captured with a Leica M165 FC fluorescent stereo microscope connected to a Leica K5 camera, using the LAS X software. Worm size was quantified using ImageJ. The freehand tool was used to draw around each worm and measure the area.

**Chromatin profiling.** Nuclei of mid-intestine, hypoderm and muscle were manually segmented in 2D or 3D using Cell-ACDC[86] based on HIS-72–GFP or Hoechst 33342 signal (the latter works only for intestinal cells). To extract the intensity profiles shown in Figs. 1c,e,g,i, 2c, 3,c,f, 4b,h, 5b,g, 6c,d and 7b,e and Extended Data Figs. 1b,d,h,k, 3c,f,j,o, 4f, 6h,k, 7f and 8e, we implemented a custom routine written in Python. The analysis steps were as follows: (1) manual annotation of the centre of the inner dark area of the nuclei using Cell-ACDC. This is assumed to be the centre of the nucleolus. (2) Determination of the nuclei contours from the segmented objects (using the function from OpenCV package called 'findContours'). (3) Extraction of the intensity profiles from the centre determined in the previous step and all the points on the contour. (4) Normalization of each profile using the distance from the centre to the contour point (0% centre, 100% contour). (5) Binning the intensities into 5% width bins (for example, intensities at any distance in the ranges 0–5%, 5–10%, 10–15% and so on were considered being at 2.5%, 7.5%, 12.5% (bin centre) and so on distance from the centre). (6) Averaging of the binned normalized profiles along the distance to obtain the single-nuclei average profiles shown as single rows on the heat maps (Fig. 1b and Extended Data Figs. 1a,c,e,g,j l,m, 2c, 3b,e,h,i,n,p, 4a,c,e, 5g,h, 6c,g,j,n, 7d,e and 8d). (7) Normalization of each single-nucleus profile by its max intensity value. (8) Averaging and standard error calculation of the single-nuclei profiles within the same experimental conditions to obtain single-condition average profiles and its associated standard errors. Since 3D segmentation is quite time consuming, we set out to determine whether 2D segmentation would yield similar results. Therefore, we segmented the same nuclei in 3D and in 2D, where 3D segmentation allows for automatic determination of the centre z-slice (using the z coordinate of the 3D object's centroid). We then compared the intensity profiles between 3D and 2D segmentation and found minimal differences. Thanks to this validation, we could proceed to segment all the other experiments in 2D.

To estimate the significance of the difference between the mean at each distance percentage 5% bin between chromatin profiles having a single peak, we performed permutation tests. To calculate a final P value, we adjusted the multiple P values for multiple tests using Bonferroni correction and combined the adjusted P values at three different regions of the plot (0–30%, 35–65% and 70–100%) using Pearson´s combined probability test.

To obtain the intensity profiles shown in Extended Data Fig. 2a, we applied the profile analysis described above but we included the nucleoli segmentation masks (see 'Nucleolar volume analysis' section). The centre of the inner dark area was therefore replaced with

the centroid of the segmented nucleoli. Finally, the distances were not normalized as in (4), but they were plotted as absolute differences from the nucleolus edge (in μm).

**Nucleolar volume analysis.** Nuclei of mid-intestine, hypoderm and muscle were manually segmented in 2D using Cell-ACDC[86] based on HIS-72–GFP or Hoechst 33342 signal and stacked into 3D 'cylindrical' objects. To detect and quantify the volume of nucleoli in 3D, we adapted the spot detection routine developed in ref. [87] and ref. [88]. The analysis steps were as follows: (1) application of a 3D Gaussian filter with a small sigma (0.75 voxel) of the FIB-1::mCherry signal. (2) Instance segmentation of the spots' signal using the best-suited automatic thresholding algorithm (either the threshold yen or Li algorithms from the Python library scikit-image[89]). (3) 3D connected component labelling of the thresholded mask obtained in (3) to separate and label the objects. (4) Manual inspection and removal of nucleoli segmented from other tissues. This step is required because nuclei were segmented in 2D and stacked into 3D, raising the possibility of nucleoli from other tissues to be detected. (5) The final volume of the nucleolus was calculated by fitting a 3D Gaussian function to the intensities of the pixels in the segmented mask. The 3D Gaussian function is

$$g(x, y, z) = g(x)g(y)g(z) + B,$$

where $B$ is a fitting parameter and $g(x)$, $g(y)$ and $g(z)$ are one-dimensional Gaussian functions given by

$$g(x) = A \exp\left(-\frac{(x - x_c)^2}{2\sigma_x^2}\right),$$

where $x_c$, $\sigma_x$ and $A$, are fitting parameters and they are the centre coordinate, the width (sigma) and the amplitude of the Gaussian peak. The $\sigma_x$, $\sigma_y$ and $\sigma_x$ parameters are then used to calculate the volume of the nucleolus with the following formula (ellipsoid's volume):

$$V_{nucleolus} = \frac{4}{3}\pi\sigma_z\sigma_y\sigma_x,$$

**Nucleolar/nuclear area.** Mid-intestine nuclei were manually segmented in 2D in their central plane, and the area was calculated using Cell-ACDC[86] based on HIS-72–GFP. Nucleoli were segmented in 3D as described in 'Nucleolar volume analysis' section. The segmentation mask was then projected to obtain a 2D representation and the area was calculated using ACDC.

**rRNA FISH quantification.** Nuclei of mid-intestine, hypoderm and muscle were manually segmented in 2D at their central plane using Cell-ACDC[86] based on Hoechst 33342 signal. The segmentation mask was extended ten slices (of 200 nm each) above and below the central plane to cover the whole nucleus. To detect and quantify the FISH signal, we adapted the spot detection routine developed in ref. [87] and ref. [88]. The analysis steps were as follows: (1) application of a 3D Gaussian filter with a small sigma (0.75 voxel) of the ATTO 488-labelled FISH probes. (2) Instance segmentation of the spots' signal using the best-suited automatic thresholding algorithm (Li algorithms from the Python library scikit-image[89]). (3) 3D connected component labelling of the thresholded mask obtained in (3) to separate and label the objects. (4) The background-corrected sum of all the voxel values belonging to the segmented objects was calculated. Background correction was performed by subtracting the median of the pixels' intensities outside of the segmented object and inside the nucleus at the central z-slice.

**Chromatin rings analysis in adult and L1 larvae.** The ratio between the outer and inner peak amplitudes of Extended Data Fig. 1n was calculated from each average single-nucleus chromatin profile (SNCP).

To calculate the amplitudes, we fitted the following model to the SNCPs (sum of two Gaussian peaks)

$$B + A_i \exp\left(-\frac{(x - x_i)^2}{2\sigma_i^2}\right) + A_o \exp\left(-\frac{(x - x_o)^2}{2\sigma_o^2}\right),$$

where $A_i$, $x_i$, $\sigma_i$, $A_o$, $x_o$, $\sigma_o$ and $B$ are fitting parameters. The subscripts i and o refer to inner and outer peaks, while A, $x$ and $\sigma$ are the amplitude, centre location and standard deviation (a measure of the width) of the specific peak, respectively. The parameter $B$ is the background level. The outer-to-inner ratios are therefore calculated as $A_o/A_i$.

For the fitting procedure we used the Python function 'scipy. optimize.curve_fit' from the 'scipy' library[90] with bounds on the fitting parameters. The amplitude and the background levels had a lower bound of 0 and an upper bound equal to the maximum of the SNCP; the centre location was bounded at ±5% (distance from the nucleolus centre of mass), and the standard deviation was bounded in the range (0%, 100%).

**Resolution limit analysis.** The procedure described in 'Chromatin rings analysis in adult and L1 larvae' section was also used to estimate the resolution limit of our imaging setup. The Gaussian function can in fact be used to approximate the Airy pattern[91]. Therefore, the average between $\sigma_i$ and $\sigma_o$ was used to compute the radius of the Airy disk as

$$d \cong 1.45\sigma_{io} + 2\Delta x_e,$$

where $d$ is the minimum resolvable distance, $\sigma_{io}$ is the average between $\sigma_i$ and $\sigma_o$, and $\Delta x_e$ is the average distance between the centre of the peaks, $x_i$ and $x_o$, and the edges of the nucleolus and the nucleus, respectively. The nucleolus was segmented as explained in 'Nucleolar volume analysis' section.

Since muscle cells have the smallest space between nucleolus and nucleus, we wanted to compare single-nucleus profiles where non-resolvable radial profiles were not included. We estimated the minimum resolvable distance in muscle cells to be approximately 8.7 pixels (4.7 + 2 × 2 in the equation above); therefore, we calculated the single-nucleus profiles by discarding from the average those radial profiles whose nucleolar edge–nuclear periphery length was shorter than 9 pixels.

**Positioning of heterochromatic and euchromatic reporters.** Quantitation of heterochromatic (Extended Data Fig. 2h) and euchromatic reporter (Extended Data Fig. 2j) distributions on focal stacks of images was done with the ImageJ plugin PointPicker (http://bigwww.epfl.ch/thevenaz/pointpicker/) as previously described[18]. Briefly, for Extended Data Fig. 2h, the disc of the nucleus in which the spot is the brightest is divided into three zones of equal surface, each containing 33% of the area and the frequency of the spot in these three zones is quantified for fed and fasted nuclei. For Extended Data Fig. 2j, the relative nucleolus edge–nuclear periphery distance is divided into 10 bins, with 0–1 closest to the nucleolus edge and 9–10 closest to the nuclear periphery, and the frequency of the spot in each bin is scored for fed and fasted nuclei.

### RNA extraction from L1 larvae and quantitative PCR with reverse transcription

To monitor the expression of pre-messenger RNAs in Extended Data Fig. 6d, the same number of RPB-2-degraded (3 h on 1 mM auxin) L1 larvae and TIR1-only controls was collected in Trizol (15596026, Thermo Fisher) and snap frozen in liquid nitrogen. After freeze cracking samples by five subsequent transfers from liquid nitrogen to 42 °C, samples were vigorously shaken for 2.5 min with 5× 30 s on/off cycles at room temperature. Next, total RNA was extracted following the manufacturer's instructions of the Rneasy Mini kit (74104, Qiagen), including an on-column DNase digestion (79254, Qiagen).

Complementary DNA was obtained by using the Maxima H Minus cDNA Synthesis Master Mix (M1661, Thermo Scientific).

For gene expression analysis, real-time PCR with the PowerUp SYBR Green Master Mix (A25742, Life Technologies) was used. Since the number of larvae was quantified and constant between the two samples, we performed a relative quantification, normalizing against unit mass by calculating the ΔCt against the control strain followed by the ratio (2ΔCt). All experiments were repeated at least three times. The primers are listed in Supplementary Table 3.

## Intestine dissection

For fed adults, animals were transferred into a M9 bath in a 6 cm plate and washed for at least 10 min before intestine dissection, while fasted adults were directly used.

Adults were placed into a drop (~15 μl) of dissection buffer (M9 buffer with 0.1% Tween and 12 mM levamisole) on a glass slide. The intestine was dissected by decapitating the worm using Gr.20 syringes (Sterican), as previously described[92]. The extruding part of the intestine was cut off the worm carcass, then transferred into pre-cooled RL lysis buffer (Norgen Biotek Corp.) on ice. A total of 30 intestines was collected per condition.

## RNA extraction for RNA-seq

RNA from dissected adult intestines was extracted using the Single Cell RNA Purification kit (Norgen Biotek Corp.) following the manufacturer's instructions. RNA quality was assessed in an RNA Pico Chip on a 2100 Bioanalyzer (Agilent Technologies) and quantified using the Qubit RNA HS Assay kit in a Qubit 3 Fluorometer (Life Technologies). cDNA synthesis, amplification and purification were performed using the SMART-Seq mRNA kit (Takara Bio) following the manufacturer's instructions. In brief, 10 ng RNA was used as input for cDNA synthesis, 9× PCR cycles were performed for cDNA amplification and cDNA was purified using AMPure XP beads (Beckman Coulter) in a ratio of 0.6:1 to cDNA. cDNA quality was assessed on a High Sensitivity DNA Chip on a 2100 Bioanalyzer (Agilent Technologies) and quantified using the Qubit dsDNA HS Assay kit in a Qubit 3 Fluorometer (Life Technologies). Dual-indexed libraries were prepared using the NexteraXT kit (Illumina) and sequencing was performed on a NextSeq2000 (Illumina) sequencer.

To quantify absolute transcript levels (Extended Data Fig. 6e), 30 nematodes were picked per replicate for a total of four technical replicates per sample and added to a lysis buffer, which was prepared together with a mix of External RNA Controls Consortium spike-ins[93]. cDNA libraries were prepared as previously described[94].

## RNA-seq analysis

For Fig. 8, reads were analysed as described previously[95]. In summary, adaptors were trimmed using Trimmomatic v0.39. Reads were aligned to the *C. elegans* genome (ce10) with Bowtie2 v2.4.5 and the R package QuasR v1.42.1,(www.bioconductor.org/packages/2.12/bioc/html/QuasR.html). The command 'proj <-qAlign('samples.txt','BSgenome.Celegans.UCSC.ce10', splicedAlignment=TRUE)' instructs hisat296 to align using default parameters, considering unique reads for genes and genome-wide distribution. Count tables of reads mapping within annotated exons in WormBase (WS220) were constructed using the qCount function of the QuasR package to quantify the number of reads in each window (qCount(proj, GRange_object, orientation = 'same')) and normalized by division by the total number of reads in each library and multiplied by the average library size. Transformation into $\log_2$ space was performed after the addition of a pseudocount of 8 to minimize large changes in abundance fold change (FC) caused by low count numbers. The EdgeR package v4.0.14 was applied to select genes with differential transcript abundances between indicated conditions (contrasts) based on false discovery rates (FDR) and $\log_2$ FC. The intestinal specificity of all samples (Extended Data Fig. 8b,g) was tested by measuring the expression enrichment within available datasets of different worm tissues[96]. Pearson coefficients describing the correlation of replicas are shown in Extended Data Fig. 8a,f. GO terms were extracted from

wormbase.org and the GO term and Kyoto Encyclopedia of Genes and Genomes pathway enrichment analysis was performed using the gprofiler2 package (v 0.2.2) as an interface to g: Profiler[97].

For Extended Data Fig. 6e, differences in absolute transcript abundance between samples were calculated via differential expression analysis using DESeq2 (ref. 98), with DESeq2 size-factors calculated exclusively from reads mapping to ERCC spike-ins.

## Statistics and reproducibility

No statistical method was used to predetermine sample size. Sample sizes were chosen based on previous literature and what is common practice in the field. No data were excluded from the analyses. The experiments were not randomized. Investigators were not blinded to group allocation of samples (genotype/treatment) during most data collection and analysis. However, key findings (chromatin reorganization in fasting and RNA Pol I-AID) were reproduced by at least an additional independent investigator who was blinded to group identity.

All experiments were reproduced with similar results. The images in Figs. 1a,d,h, 2a,b, 3a, 4a, 5a,e, 6a and 7a and Extended Data Figs. 1f,i, 2b,k, 3d,l, 6f,i,m and 8c represent data from three independent biological experiments (*n* > 30 animals). The experiment in Extended Data Fig. 3a was repeated independently four times (*n* > 50 animals). The images in Figs. 2f and 3d are representative of five independent biological experiments (*n* > 60 animals). Experiments in Fig. 1f and Extended Data Fig. 5a,c,e were repeated independently twice (*n* > 30 animals).

Statistics analysis and data visualization were performed in Python (3.10), Excel (Version 2308), R (4.3.1), GraphPad Prism (8.0.1) or the gprofiler2 package (v 0.2.2). All statistical tests, exact *P* values, *n* and number of biological replicas are listed in Supplementary Table 2.

## Reporting summary

Further information on research design is available in the Nature Portfolio Reporting Summary linked to this article.

## Data availability

The RNA-seq datasets comparing fed and fasted worms have been deposited at the Gene Expression Omnibus and can be accessed under GSE268926. The RNA-seq datasets measuring the effect of RNA Pol I depletion have been deposited at the Gene Expression Omnibus and can be accessed under GSE268974. The RNA-seq datasets analysing the transcriptome following RPB-2 degradation have been deposited at bioproject and can be accessed under PRJNA1140129. The publicly available dataset used in this study, BSgenome.Celegans.UCSC.ce10, is available at https://bioconductor.org/packages/BSgenome.Celegans.UCSC.ce10/ (ref. 99). All other data supporting the findings of this study are available from the corresponding author on reasonable request. Source data are provided with this paper.

## Code availability

All code used to perform chromatin profiling is available on GitHub at https://github.com/SchmollerLab/SeelMito (ref. 100) and https://github.com/ElpadoCan/ChromRings (ref. 101).

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

## Acknowledgements

We thank R. Schneider, M.-E. Torres-Padilla and S.M. Gasser for critical reading of the manuscript; the labs of H. Grosshans and F. Steiner; R. Gleason in X. Chen´s lab; the Caenorhabditis Genetics Center, funded by the National Institutes of Health Office of Research Infrastructure Programs (grant no. P40 OD010440CGC); and the *C. elegans* Gene Knockout Project at the Oklahoma Medical Research Foundation for sharing strains. We thank SunyBiotech for support in generating the *rpc-1-mNeonGreen-AID* allele. D.S.C. thanks Helmholtz Munich for support. Funding was provided by the German Research Foundation (Deutsche Forschungsgemeinschaft) Sonderforschungsbereiche (SFB) 1064, collaborative research center in chromatin dynamics (D.S.C.). The German Research Foundation (Deutsche Forschungsgemeinschaft) Schwerpunktprogramm 2202 Priority Programme 'Spatial Genome Architecture in Development and Disease' (D.S.C.). National Institutes of Health grant 5R35GM138340-03 (E.S.C.). The Welch Foundation F-2133-20230405 (E.S.C.). Human Frontier Science Program (career development award) (K.M.S.). The Swiss National Science Foundation in the form of an Eccellenza Professorial Fellowship (PCEFP3_181204) (B.D.T.). European Research Council under the European Union's Horizon 2020 Research and Innovation Programme (grant agreement no. 852201) (N.S.). The Spanish Ministry of Economy, Industry and Competitiveness to the European Molecular Biology Laboratory partnership, the Centro de Excelencia Severo Ochoa (CEX2020-001049-S, MCIN/AEI /10.13039/501100011033), the Centres de Recerca de Catalunya Programme/Generalitat de Catalunya (N.S.). The Spanish Ministry of Economy, Industry and Competitiveness Excelencia award PID2020-115189GB-I00 (N.S.)

## Author contributions

Conceptualization: N.A.-R. and D.S.C. Methodology: F.P., K.M.S., D.S.C. and J.P. Software: F.P. Investigation: N.A.-R., J.H., F.B., D.S.C., L.P., J.P. and A.d.C.F. Visualization: F.P., N.A.-R., D.S.C., K.M.S., J.P. and A.d.C.F. Validation: N.A.-R., J.H. and F.B. Resources: Q.Z., E.S.C., B.D.T. and N.S. Funding acquisition: D.S.C., K.M.S., E.S.C., N.S. and B.D.T. Supervision: D.S.C. Writing—original draft: N.A.-R. and D.S.C. Writing—review and editing: D.S.C., N.A.-R., K.M.S., B.D.T., J.P. and L.P.

## Funding

## Competing interests

The authors declare no competing interests.

## Additional information

**Extended data** is available for this paper at https://doi.org/10.1038/s41556-024-01512-w.

**Correspondence and requests for materials** should be addressed to Daphne S. Cabianca.

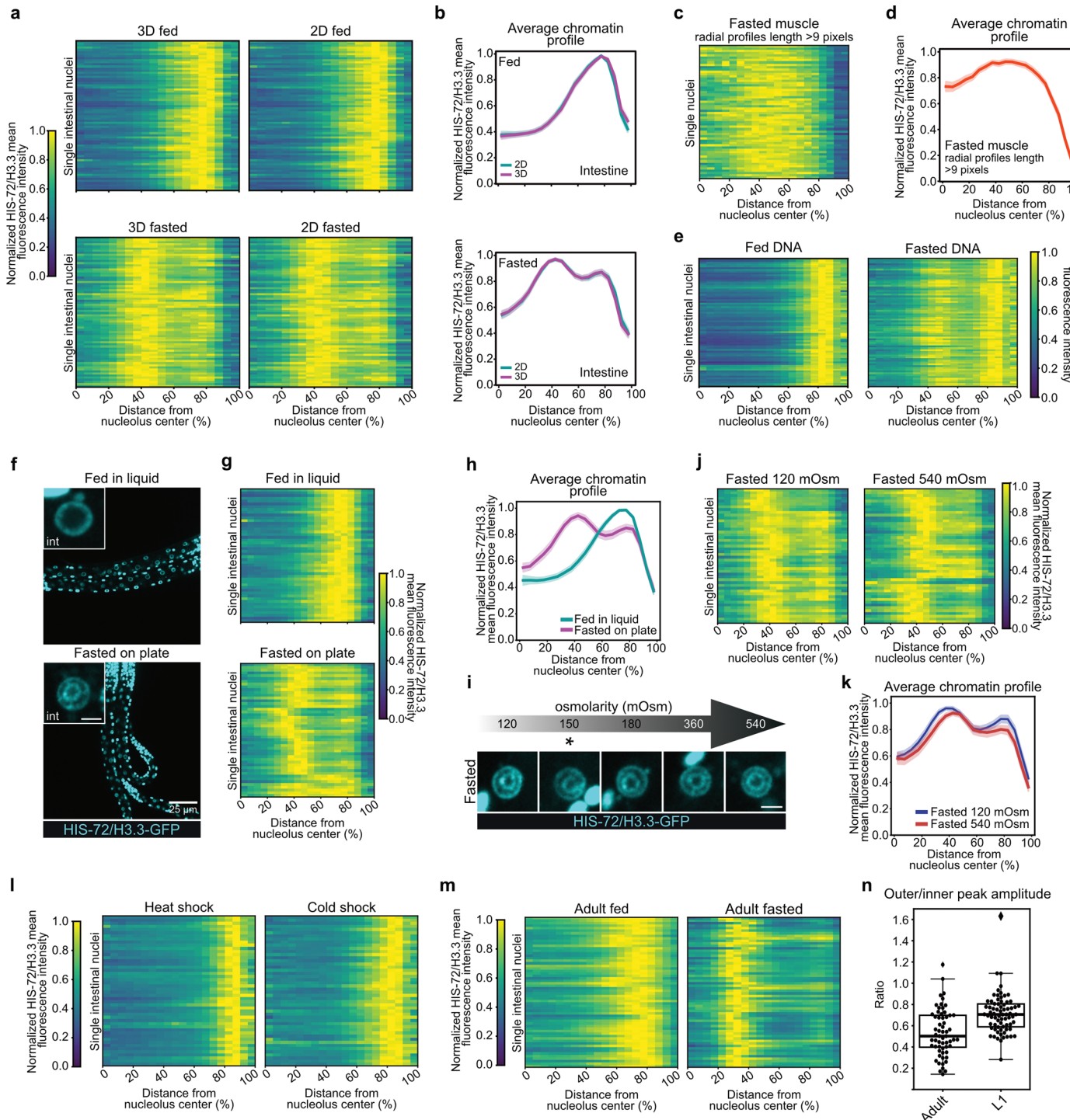

**Extended Data Fig. 1 | See next page for caption.**

**Extended Data Fig. 1 | related to Fig. 1. a**, Heatmaps showing the radial fluorescence intensity profiles of HIS-72/H3.3-GFP in intestinal nuclei of wt L1s of the indicated nutritional status as a function of the relative distance from the nucleolus center. Each row corresponds to a single nucleus, segmented in 3D (left) or in 2D at its central plane (right). The same 72 intestinal nuclei in animals from 3 independent biological replicas were analysed for the 3D and 2D segmentation. **b**, Line plots of the averaged single nuclei profile shown in (**a**). The shaded area represents the 95% confidence interval of the mean profile. **c**, Heatmap as in (**a**, right) but for muscle nuclei of fasted L1s where individual radial profiles with nucleolar edge-nuclear periphery length lower than 9 pixels were excluded. 59 muscle nuclei were analysed in animals from 3 independent biological replicas. **d**, Line plots as in (**b**) but of the averaged single nuclei profile shown in (**c**). **e**, Heatmaps as in (**a**, right) but of Hoechst 33342-stained DNA in intestinal nuclei of fed and fasted animals. 72 intestinal nuclei were analysed in animals from 3 independent biological replicas. **f**, Single focal planes of representative wt L1 larvae expressing HIS-72/H3.3-GFP, fed in liquid or fasted on plate. Insets: zoom of single intestinal nucleus. Inset scale bar, 2.5 μm. **g**, Heatmaps as in (**a**, right) but for 48 intestinal nuclei of wt animals fed in liquid or fasted on plate. Data are from 3 independent biological replicas. **h**, Line plots as in (**b**) but of the averaged single nuclei profile shown in (**g**). **i**, Single focal planes of representative, wt intestinal nuclei expressing HIS-72/ H3.3-GFP in L1 larvae that were fasted at the indicated osmolarity. * indicates the standard osmolarity of worm plates (150 mOsm). Scale bar indicates 2.5 μm. **j**, Heatmaps as in (**a**, right) but for 48 intestinal nuclei of animals fasted at 120 mOsm or at 540 mOsm. Data are from 3 independent biological replicas. **k**, Line plots as in (**b**) but of the averaged single nuclei profile shown in (**j**). **l**) Heatmaps as in (**a**, right) but for 48 intestinal nuclei of L1 larvae heat shocked at 34 °C or cold shocked at 4 °C for 6 hours. Data are from 2 independent biological replicas. **m**) Heatmaps as in (**a**, right) but for 60 intestinal nuclei of wt adults fed or fasted. Data are from 3 independent biological replicas. **n**, Boxplot showing the ratio of the amplitude of the outer and inner peak HIS-72/H3.3-GFP signal in fasted adults and L1 larvae. 60 and 72 intestinal nuclei were analysed for adults and L1 larvae, respectively, from 3 independent biological replicates. Box limits are 25th and 75th percentiles, whiskers denote 1.5 times the interquartile ranges, points outside the whiskers are outliers, and the median is shown as a line. Source numerical data are available in source data.

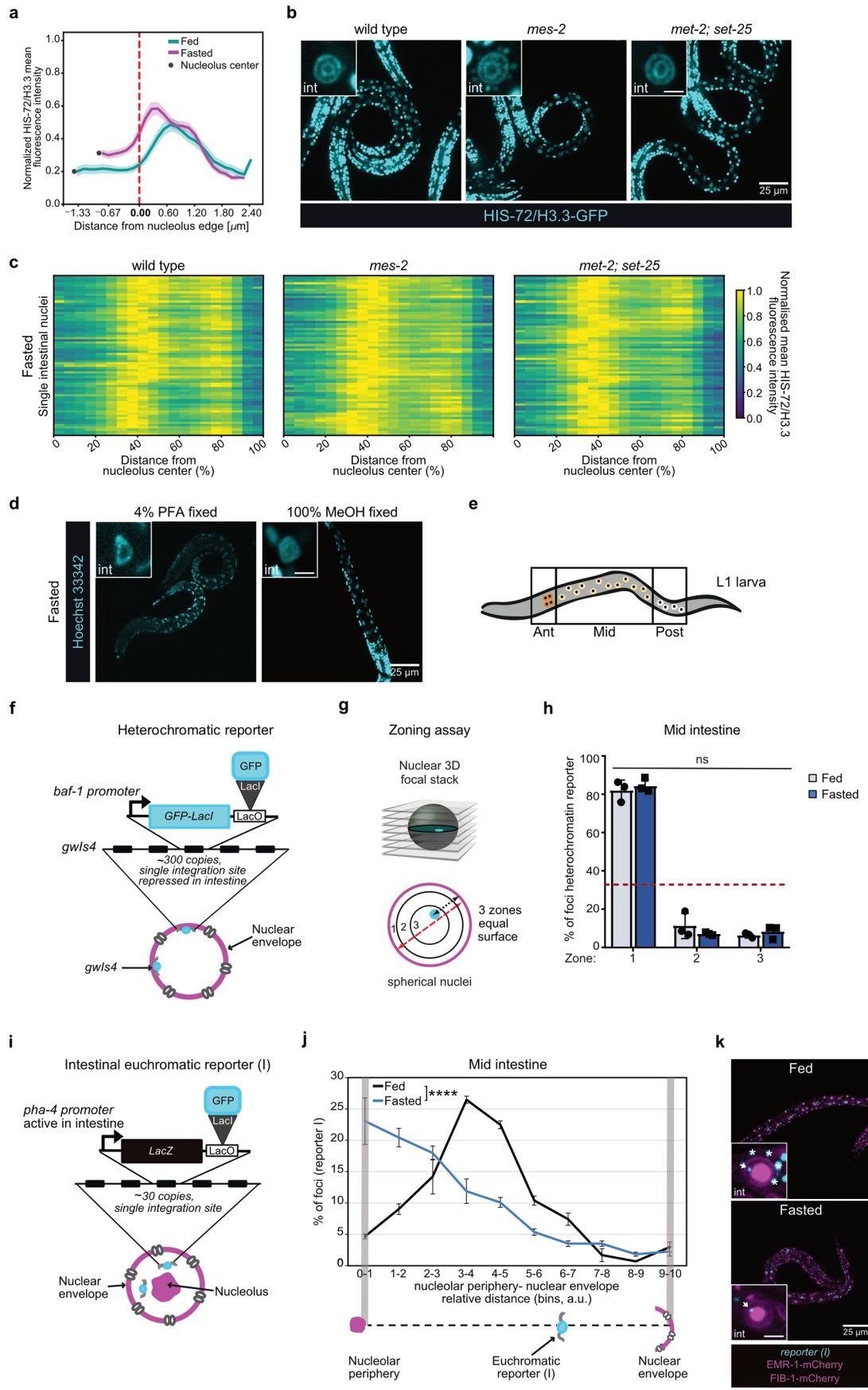

**Extended Data Fig. 2 | See next page for caption.**

**Extended Data Fig. 2 | related to Fig. 2. a**, Line plot of the average distance of the HIS-72/H3.3-GFP signal from the nucleolus edge, identified using FIB-1-mCherry, in single intestinal nuclei of fed and fasted L1 larvae. The shaded area represents the 95% confidence interval of the mean profile. 72 intestinal nuclei were analysed in animals from 3 independent biological replicas. **b**, Single focal planes of representative fasted L1 larvae of the indicated genotype, expressing fluorescently tagged HIS-72/H3.3. Insets: zoom of single intestinal nucleus. Inset scale bar, 2.5 µm. **c**, Heatmaps showing the radial fluorescence intensity profiles of HIS-72/H3.3-GFP in intestinal nuclei of fasted L1 larvae of the indicated genotype as a function of the relative distance from the nucleolus center. Each row corresponds to a single nucleus, segmented in 2D at its central plane. 70 intestinal nuclei were analysed in animals from 3 independent biological replicas. **d**, As in (**b**) but of representative wt L1 larvae fasted, fixed with the indicated chemical and stained with Hoechst 33342 to visualize DNA. **e**, Schematic representation of intestinal nuclei in L1 larvae. **f**, Schematic representation of the heterochromatic reporter used in (**h**). **g**, Zoning assay for GFP-LacI marked heterochromatic reporter distribution. Radial position is determined relative to the fluorescently-tagged nuclear membrane, and values are binned into three concentric zones of equal surface. Zone 1 is the most peripheral. **h**, Heterochromatic reporter distribution quantitation, as described in (**g**), in mid intestine cells of wt fed and fasted L1 larvae. Red dashed line indicates random distribution of 33%. Animals from 3 independent biological replicas, with at least 70 GFP-LacI spots per replicate, were analysed. Data are shown as mean ± SEM. By χ2 test, fed and fasted samples are not significantly (ns) different from each other. p value and exact n are in Supplementary Table 2. **i**, Schematic representation of the euchromatin reporter (**l**) used in (**j**) and (**k**). **j**, Line plots of the percentage of alleles represented in (**i**) found at the indicated relative position between the nucleolus and the nuclear envelope in fed and fasted animals. Data are shown as mean ± SEM of 3 independent biological replicas. p value by χ2 is shown; **** indicates a p value < 0.0001. See n and p values in Supplementary Table 2. **k**, As in (**b**), but for wt L1 larvae expressing the indicated fluorescently-tagged markers, fed or fasted. * mark autofluorescence from the intestinal cytoplasm. Source numerical data are available in source data.

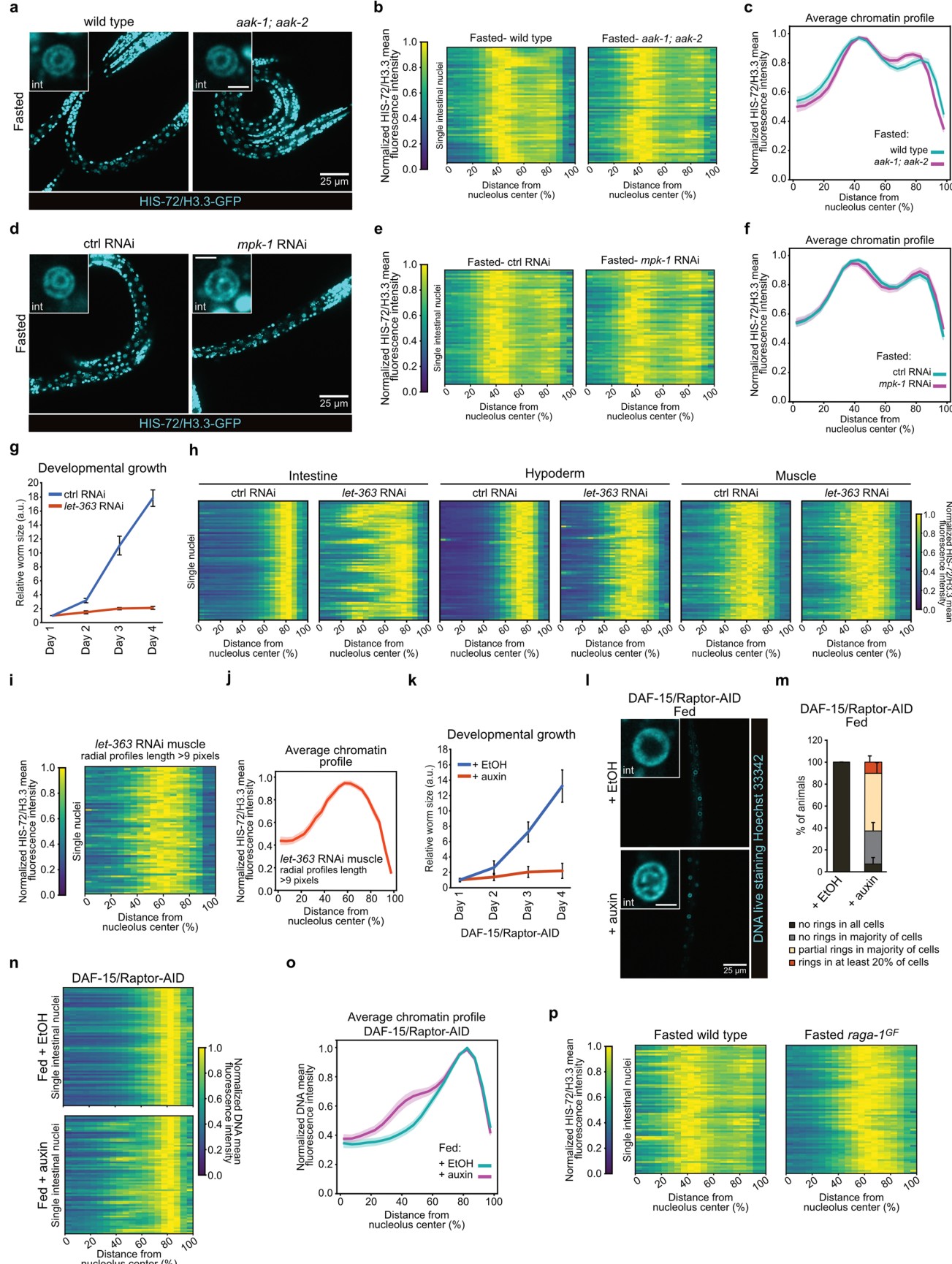

**Extended Data Fig. 3 | See next page for caption.**

**Extended Data Fig. 3 | related to Fig. 3. a**, Single focal planes of representative fasted L1 larvae of the indicated genotype, expressing fluorescently tagged HIS-72/H3.3. Insets: zoom of single intestine nucleus. Inset scale bar, 2.5 μm. **b**, Heatmaps showing the radial fluorescence intensity profiles of HIS-72/H3.3-GFP in 60 intestinal nuclei of fasted wild type or *aak-1; aak-2* mutant as a relative distance from the nucleolus center. Each row corresponds to a single nucleus, segmented in 2D at its central plane. Data are from 4 independent biological replicas. **c**, Line plot of the averaged single nuclei profile shown in (**b**). The shaded area represents the 95% confidence interval of the mean profile. **d**, As in (**a**) but of fasted wt L1 larvae under control or *mpk-1* RNAi. **e**, Heatmaps as in (**b**) but of 60 intestinal nuclei of wt L1 larvae under control or *mpk-1* RNAi from 3 independent biological replicas. **f**, Line plot as in (**c**) but of the averaged single nuclei profile shown in (**e**). **g**, Line plot comparing the differences in developmental growth between control and *let-363* RNAi animals, as measured by body size. Day 1 represents the first day after hatching. Data are shown as mean ± SEM of 4 independent biological replicas. **h**, Heatmaps as in (**b**) but of 72 intestinal nuclei of the indicated tissue of fed L1 larvae treated with control or *let-363* RNAi from 3 independent biological replicas. **i**, Heatmap as in (**b**) but of 74 muscle nuclei of fed L1s treated with *let-363* RNAi where individual radial profiles with nucleolar edge-nuclear periphery length lower than 9 pixels were excluded. Data are from 3 independent biological replicas. **j**, Line plots as in (**c**) but of the averaged single nuclei profile shown in (**i**). **k**, Line plot as in (**g**) but of fed animals, expressing an endogenously degron-tagged (AID) DAF-15/Raptor treated with 1 mM auxin or EtOH (control). Data are shown as mean ±SEM of 3 independent biological replicas. **l**, As in (**a**) but of fed L1 larvae expressing an endogenously tagged DAF-15/Raptor-AID treated with 1 mM auxin, or EtOH (control), and where the DNA has been stained with Hoechst 33342. **m**, Quantification of the percentage of fed DAF-15/Raptor-AID animals treated with 1 mM auxin or EtOH within the indicated categories for 3D chromatin organization in intestine. Data are shown as mean ±SEM of 3 independent biological replicas. **n**, Heatmaps as in (**b**) but of Hoechst 33342-stained DNA in intestinal nuclei of fed L1 expressing DAF-15/Raptor-AID, treated with 1 mM auxin or EtOH. 72 and 71 intestinal nuclei were analysed for 1 mM auxin and EtOH, respectively, from 3 independent biological replicas. For auxin-treated individuals, animals in proportions to their relative abundance within the chromatin organization categories as in (**m**) were analysed. **o**, Line plots as in (**c**) but of the averaged single nuclei profile shown in (**n**). **p**, Heatmaps as in (**b**) but for 72 intestinal nuclei of fasted wt and *raga-1*^GF animals. Data are from 3 and 4 independent biological replicas for *raga-1*^GF and wt, respectively. Source numerical data are available in source data.

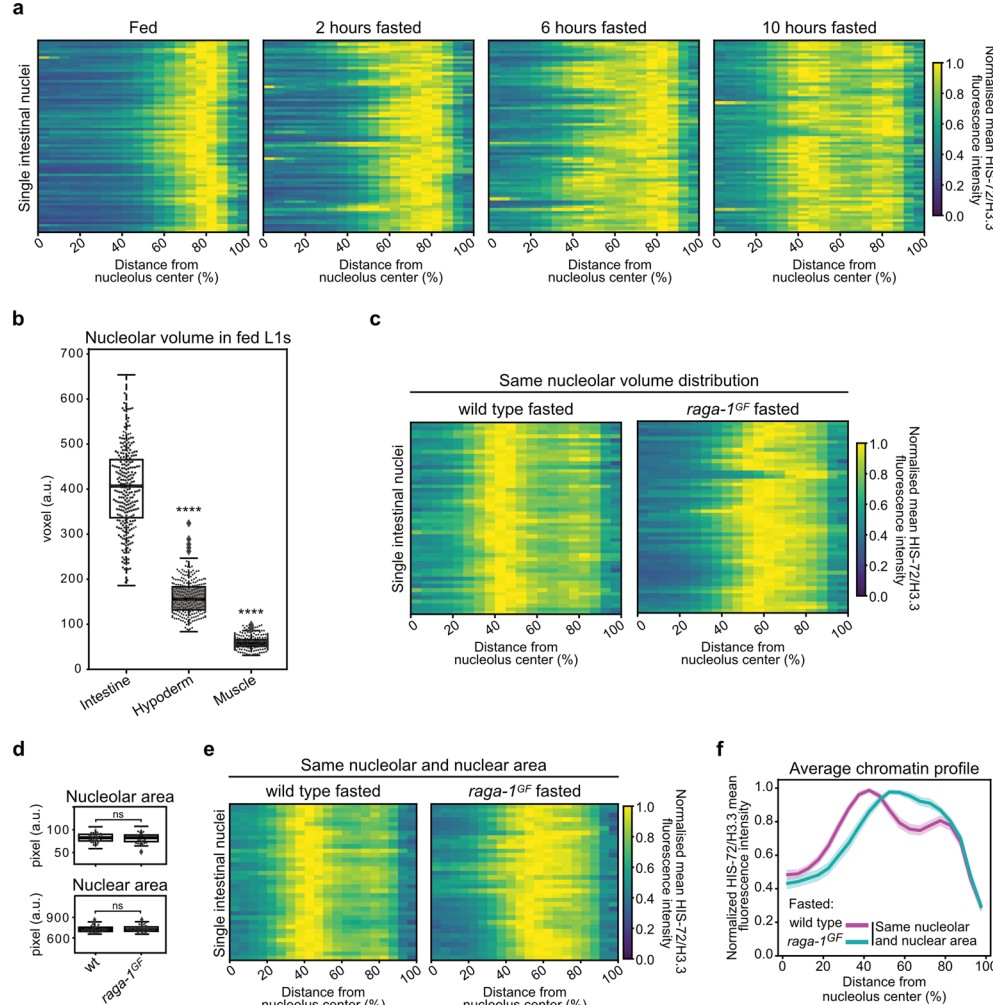

**Extended Data Fig. 4 | related to Fig. 4. a**, Heatmaps showing the radial fluorescence intensity profiles of HIS-72/H3.3-GFP in intestinal nuclei of wt animals of the indicated nutritional status as a function of the relative distance from the nucleolus center. Each row corresponds to a single nucleus, segmented in 2D at its central plane. 72 intestinal nuclei were analysed in animals from 3 independent biological replicas. **b**, Boxplots comparing the volume of the nucleolus, measured with FIB-1-mCherry, in the intestine, hypoderm and muscle of fed wt L1 larvae. Data are from 3 independent biological replicas. **c**, Heatmaps as in (**a**) but for single intestinal nuclei of fasted wt and *raga-1^GF* animals with the same nucleolar volume shown in (Fig. 4g). 50 intestinal nuclei were analysed. **d**, Boxplots showing 35 intestinal nuclei with the same nucleolar (up) and

nuclear (bottom) area, measured using FIB-1-mCherry and HIS-72/H3.3-GFP, respectively, in fasted wt and *raga-1^GF* animals. **e**, Heatmaps as in (**a**) but for single intestinal nuclei shown in (**d**). **f**, Line plot of the averaged single nuclei profile shown in (**e**). The shaded area represents the 95% confidence interval of the mean profile. For b and d, box limits are 25th and 75th percentiles, whiskers denote 1.5 times the interquartile ranges, points outside the whiskers are outliers, and the median is shown as a line. Probability values from two-sided Wilcoxon rank sum tests comparing to fed intestine (**b**) or to wt (**d**) are shown: **** indicates p value < 0.0001. ns= not significant. See p values and n in Supplementary Table 2. Source numerical data are available in source data.

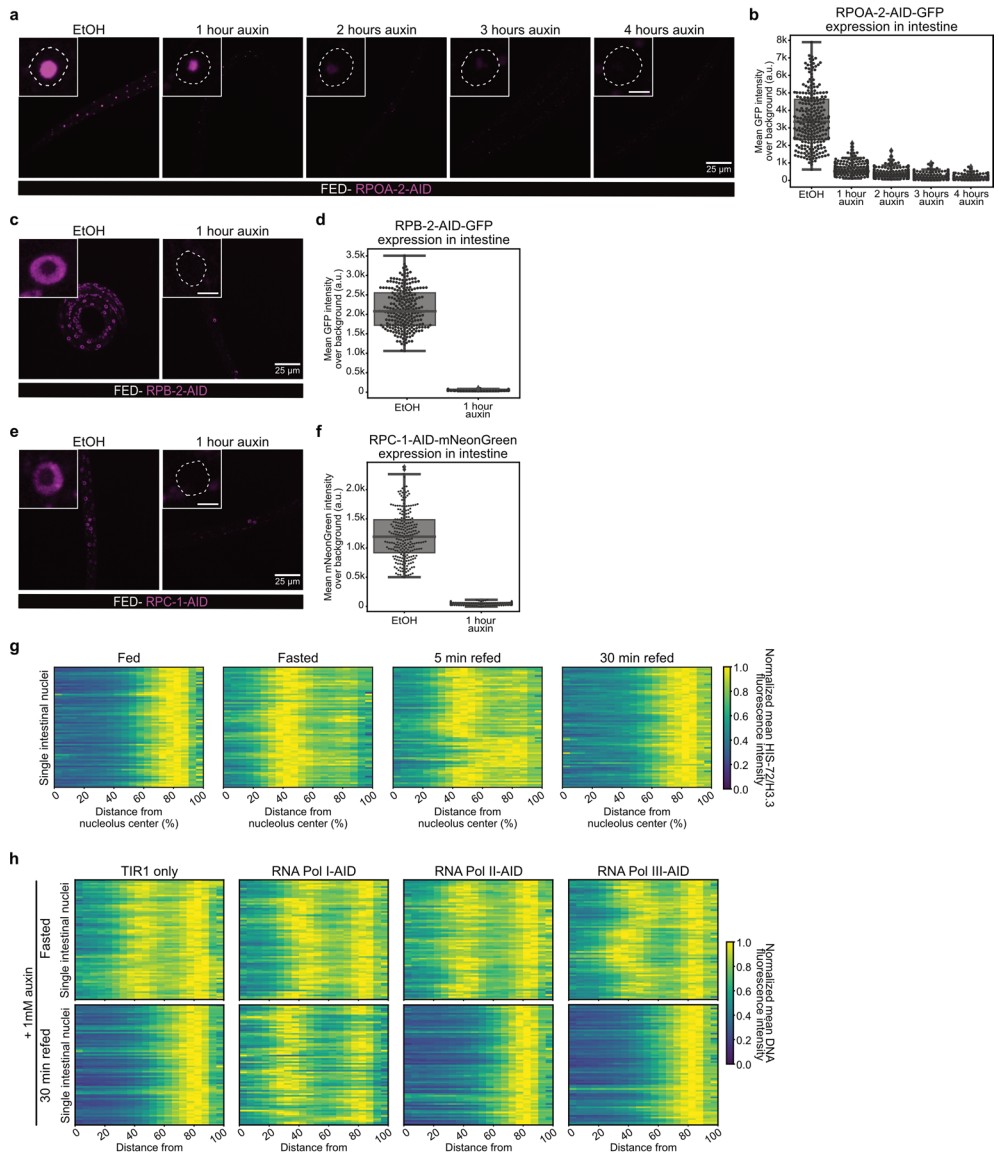

**Extended Data Fig. 5 | related to Fig. 5. a**, Single focal planes of representative fed L1 larvae expressing RPOA-2-GFP-AID from its endogenous locus, and TIR1, control treated (EtOH), or treated with 1 mM auxin for the indicated amount of time. Insets: zoom of single intestinal nucleus. Inset scale bar, 2.5 μm. **b**, Boxplot comparing the mean fluorescence intensity of RPOA-2-GFP-AID in intestinal nuclei in fed L1s expressing RPOA-2-GFP-AID and TIR1, control treated (EtOH), or treated with 1 mM auxin for the indicated amount of time. **c**, As in (**a**), but for animals expressing RPB-2-AID-GFP. **d**, As in (**b**), but for animals expressing RPB-2-AID-GFP. **e**, As in (**a**), but for animals expressing RPC-1-AID-mNeonGreen. **f**, As in (**b**), but for animals expressing RPC-1-AID-mNeonGreen. For **b**, **d** and **f**, data are from 2 independent biological replicas. See n in Supplementary Table 2. Box limits are 25th and 75th percentiles, whiskers denote 1.5 times the interquartile ranges, points outside the whiskers are outliers, and the median is shown as a line. **g**) Heatmaps showing the radial fluorescence intensity profiles of HIS-72/H3.3-GFP in intestinal nuclei of wt animals of the indicated nutritional status as a function of the relative distance from the nucleolus center. Each row corresponds to a single nucleus, segmented in 2D at its central plane. 72 intestinal nuclei were analyzed in animals from 3 independent biological replicas. **h**, Heatmaps as in (**g**) but of Hoechst 33342-stained DNA in intestinal nuclei of animals expressing the indicated AID tag or TIR1 only, as control, that were fasted or 30 minutes refed in presence of 1 mM auxin. 72 intestinal nuclei were analyzed, except for fasted RNA Pol II-AID and 30 minutes refed TIR1 only, where 69 and 71 nuclei were analyzed, respectively. Data are from 3 independent biological replicas. For RNA Pol II-AID 30 minutes refed, animals in proportions to their relative abundance within the 3D chromatin organization categories as in (Fig. 5f) were analyzed. Source numerical data are available in source data.

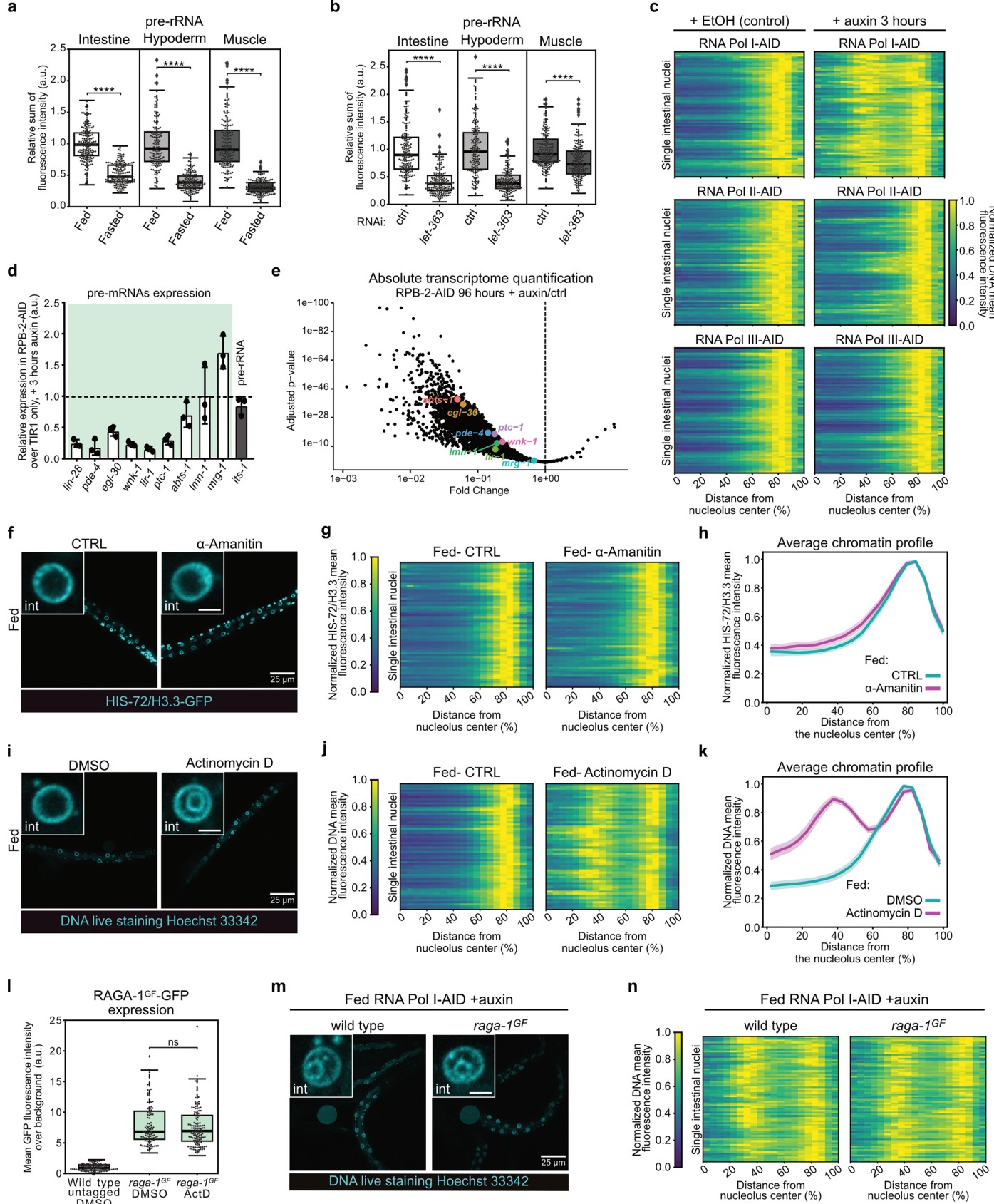

**Extended Data Fig. 6 | See next page for caption.**

**Extended Data Fig. 6 | related to Fig. 6. a**, Boxplot showing the relative changes in pre-rRNA levels, measured by FISH in the intestine, hypoderm and muscle of fasted wt animals compared to the corresponding tissues in fed animals. The results from 3 independent biological replicas are shown. **b**, same as (**a**) but for *let-363 RNAi* relative to control RNAi. Data are from 3 independent biological replicas. **c**, Heatmaps showing the radial fluorescence intensity profiles of DNA in 72 intestinal nuclei of the indicated fed, degron-tagged animals in presence of EtOH (control) or 3 hours on 1 mM auxin as a function of the relative distance from the nucleolus center. Each row corresponds to a single nucleus, segmented in 2D at its central plane. Data are from 3 independent biological replicas. **d**, RT-qPCR quantification of the expression level of 9 randomly selected genes with an intron >150 bp in RPB-2-AID larvae compared to controls (TIR1 only) exposed to 1 mM auxin for 3 hours. Data are shown as mean ±SD of 3 independent replicas. **e**, Scatter plot showing global reduction of the transcriptome in adults expressing RPB-2-AID and exposed to auxin for 96 hours compared to those not exposed to auxin. Statistical significance was assessed using the two-sided Wald test in DESeq2[98], with multiple hypothesis testing corrected by the Benjamini-Hochberg procedure. Marked genes correspond to those quantified in panel (**d**). **f**, Single focal planes of representative fed wt L1 larvae expressing HIS-72/H3.3, treated with 25 µg/mL α-amanitin or H2O (CTRL). Insets: zoom of single intestine nucleus. Inset scale bar, 2.5 µm. **g**, Heatmaps as in (**c**) but of HIS-72/H3.3 in 60 intestinal nuclei of fed wt animals treated with α-amanitin or control (H2O)

from 3 independent biological replicas. **h**, Line plots of the averaged single nuclei profile shown in (**g**). The shaded area represents the 95% confidence interval of the mean profile. Profiles were compared to estimate the statistical significance of differences, as described in the methods. p values are in Supplementary Table 2. **i**, Same as (**f**) but of fed wt L1 larvae, live-stained with Hoechst 33342 and treated with 100 µg/mL Actinomycin D or DMSO (control). **j**, Heatmaps as in (**c**) but of Hoechst 33342-stained DNA in 60 intestinal nuclei of fed wt animals treated with Actinomycin D or DMSO from 3 independent biological replicas. **k**, line plots as in (**h**) but of the averaged single nuclei profile shown in (**j**). **l**, Boxplot comparing the intensity of GFP-RAGA-1^GF in larvae treated with 100 µg/mL Actinomycin D or DMSO, as control, for 6 hours. Data are from 3 independent biological replicas. **m**, As in (**f**) but of fed wt and *raga-1^GF* L1 larvae expressing RNA Pol I degron-tagged (RNA Pol I-AID) treated with 1 mM auxin for 5 hours and live-stained with Hoechst 33342 to monitor DNA. **n**, As in (**c**) but for Hoechst 33342-stained intestinal nuclei of fed wt and *raga-1^GF* animals expressing RNA Pol I degron-tagged (RNA Pol I-AID) that were treated with 1 mM auxin for 5 hours. 70 intestinal nuclei were analysed in animals from 3 independent biological replicas. For a, b and l, box limits are 25th and 75th percentiles, whiskers denote 1.5 times the interquartile ranges, points outside the whiskers are outliers, and the median is shown as a line. Probability values from two-sided Wilcoxon rank sum tests are shown: p values < 0.0001 are indicated by ****. ns= not significant. See p values and n in Supplementary Table 2. Source numerical data are available in source data.

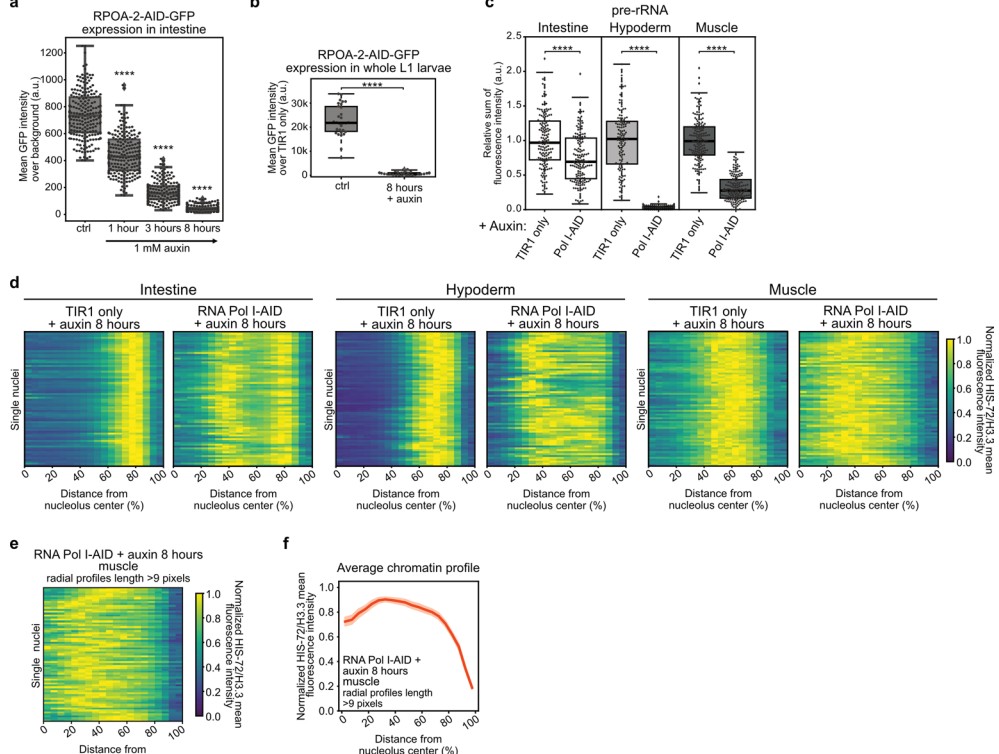

**Extended Data Fig. 7 | related to Fig. 7. a**, Boxplot comparing the expression of RPOA-2-AID-GFP in the intestine of fed L1 larvae expressing ubiquitous TIR1-BFP and treated with 1 mM auxin for the indicated time. Data are from 3 independent biological replicas. **b**, same as (**a**) but in whole larvae that were treated with 1 mM auxin for 8 hours. 30 animals from 2 independent biological replicas were analysed. **c**, Boxplot showing the relative changes in pre-rRNA levels, measured by FISH, in intestine, hypoderm and muscle cells of Pol I-AID animals, compared to the corresponding tissue in TIR1 only animals, upon treatment with 1 mM auxin for 8 hours. The results from 3 independent biological replicas are shown. For (**a-c**), box limits are 25th and 75th percentiles, whiskers denote 1.5 times the interquartile ranges, points outside the whiskers are outliers, and the median is shown as a line. Probability values from two-sided Wilcoxon rank sum tests comparing to control are shown: **** indicate p value < 0,0001. See p values and n

in Supplementary Table 2. **d**, Heatmaps showing the radial fluorescence intensity profiles of HIS-72/H3.3-GFP in the indicated tissues of fed animals expressing ubiquitous TIR1 alone (TIR1 only) or with RPOA-2-AID (Pol I-AID) in presence of 1 mM auxin, as a function of the relative distance from the nucleolus center. Each row corresponds to a single nucleus, segmented in 2D at its central plane. 72 intestinal nuclei were analysed in animals from 3 independent biological replicas. **e**, Heatmap as in (**d**) but for muscle nuclei of Pol I-AID animals where individual radial profiles with nucleolar edge-nuclear periphery length lower than 9 pixels were excluded. 69 muscle nuclei were analysed in animals from 3 independent biological replicas. **f**, Line plots of the averaged single nuclei profile shown in (**e**). The shaded area represents the 95% confidence interval of the mean profile. Source numerical data are available in source data.

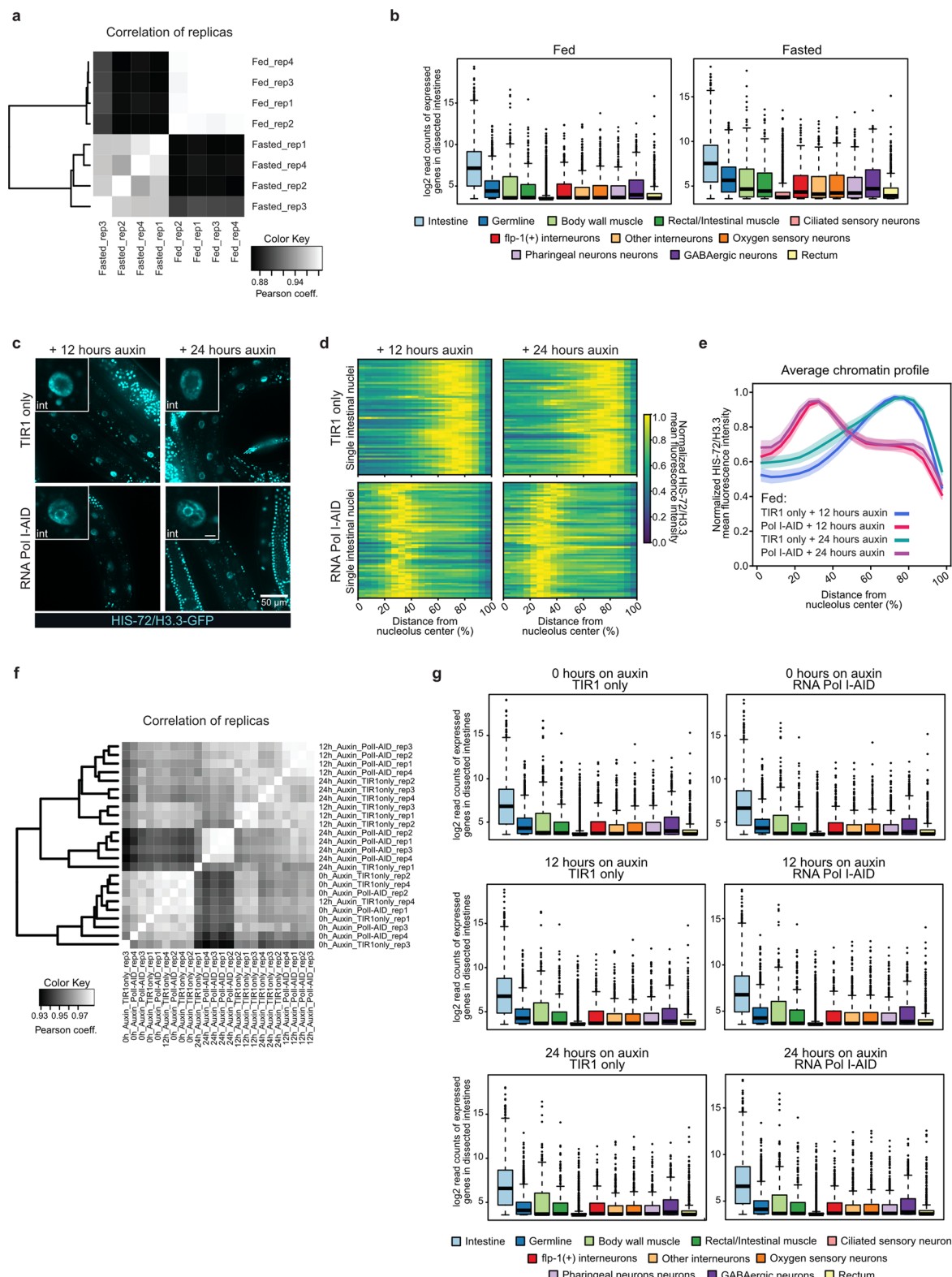

**Extended Data Fig. 8 | See next page for caption.**

**Extended Data Fig. 8 | related to Fig. 8. a**, Pearson correlation analysis between the biological replicates of intestinal RNA samples of fasted and fed adults. **b**, Boxplots showing the enrichment of the genes expressed in the dissected intestines of fed and fasted animals within expression datasets from different tissues indicated with different colors. The samples are enriched in intestinally-expressed genes (labelled in light blue). **c**, Single focal planes of representative fed adults expressing HIS-72/H3.3-GFP, ubiquitous TIR1 alone (TIR1 only) or with RPOA-2-AID (Pol I-AID), treated with 1 mM auxin for the indicated time. Insets: zoom of single intestinal nucleus. Inset scale bar, 5 μm. **d**, Heatmaps showing the radial fluorescence intensity profiles of HIS-72/H3.3-GFP in intestinal nuclei of adults, expressing ubiquitous TIR1 alone (TIR1 only) or with RPOA-2-AID (Pol I-AID), treated with 1 mM auxin for the indicated time, as a function of the relative distance from the nucleolus center. Each row corresponds to a single nucleus, segmented in 2D at its central plane. 60 intestinal nuclei were analysed in animals from 3 independent biological replicas. **e**, Line plots of the averaged single nuclei profile shown in (**d**). The shaded area represents the 95% confidence interval of the mean profile. **f**, As in (**a**) but for adults expressing TIR1 alone (TIR1 only) or with RPOA-2-AID (Pol I-AID), treated with 1 mM auxin for the indicated time. **g**, As in (**b**) but for adults expressing TIR1 alone (TIR1 only) or with RPOA-2-AID (Pol I-AID), treated with 1 mM auxin for the indicated time. For b and g, the average of 4 biological replicas is shown. The boxes represent 50%, and whiskers 90% of the group. The median is shown as a line and outliers are indicated as dots. Source numerical data are available in source data.

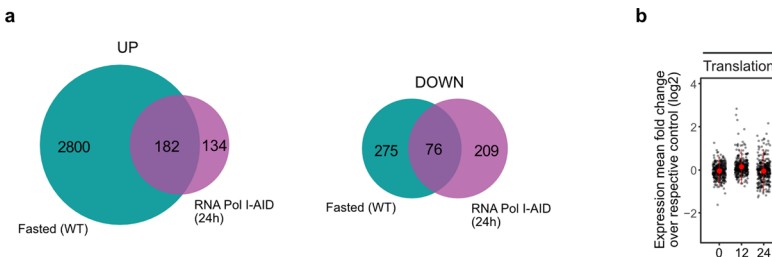

**Extended Data Fig. 9 | related to Fig. 8. a**, Venn diagrams showing the overlap of upregulated (left) and downregulated (right) genes between fasted and RNA Pol I-AID animals exposed to auxin for 24 hours. See Fig. 8a, c for scatterplots where the differentially expressed genes are shown color-coded in both conditions. **b**, Comparison of the relative expression of genes belonging to the GO category of 'Translation' or 'Nucleolus' in the intestine of RPOA-2-AID (Pol I-AID) versus TIR-1 only animals, treated with 1 mM auxin for the indicated time, or in wt fasted versus fed animals. Data are shown as mean ± SD of 4 biological replicas.

# Reporting Summary

## Statistics

For all statistical analyses, confirm that the following items are present in the figure legend, table legend, main text, or Methods section.

| n/a | Confirmed | |
|---|---|---|
| ☐ | ☒ | The exact sample size (*n*) for each experimental group/condition, given as a discrete number and unit of measurement |
| ☐ | ☒ | A statement on whether measurements were taken from distinct samples or whether the same sample was measured repeatedly |
| ☐ | ☒ | The statistical test(s) used AND whether they are one- or two-sided *Only common tests should be described solely by name; describe more complex techniques in the Methods section.* |
| ☒ | ☐ | A description of all covariates tested |
| ☐ | ☒ | A description of any assumptions or corrections, such as tests of normality and adjustment for multiple comparisons |
| ☐ | ☒ | A full description of the statistical parameters including central tendency (e.g. means) or other basic estimates (e.g. regression coefficient) AND variation (e.g. standard deviation) or associated estimates of uncertainty (e.g. confidence intervals) |
| ☐ | ☒ | For null hypothesis testing, the test statistic (e.g. *F*, *t*, *r*) with confidence intervals, effect sizes, degrees of freedom and *P* value noted *Give P values as exact values whenever suitable.* |
| ☒ | ☐ | For Bayesian analysis, information on the choice of priors and Markov chain Monte Carlo settings |
| ☒ | ☐ | For hierarchical and complex designs, identification of the appropriate level for tests and full reporting of outcomes |
| ☐ | ☒ | Estimates of effect sizes (e.g. Cohen's *d*, Pearson's *r*), indicating how they were calculated |

*Our web collection on statistics for biologists contains articles on many of the points above.*

## Software and code

Policy information about availability of computer code

| Data collection | Images were collected with a Confocal Spinning Disk Microscopefrom Visitron Systems GmbH, equipped as follows: Nikon Eclipse Ti2 microscope + Plan Apo λ 100X/1.45 oil objective, Plan apo λ 60X/1.40 oil objective, Yokogawa CSU-W1 confocal scanner unit, VS-Homogenizer, EMCCD camera [Andor - iXon Series], and VisiView software (version 5.0.0.17) for acquisition. |
|---|---|
| | Images were also captured with a Leica M165 FC fluorescent stereo microscope connected to a Leica K5 camera, using the LAS X software. |
| | RNA-seq data were collected using a NextSeq2000 (Illumina) sequencer. |
| | qPCR data were collected on a LightCycler 96 (Roche). |

| Data analysis | Fiji/ImageJ 1.53c |
|---|---|
| | ImageJ plugin PointPicker 578 (http://bigwww.epfl.ch/ thevenaz/pointpicker/) |
| | Cell-ACDC (Padovani et al. BMC Biol. 2022) |
| | Custom algorithms, code are available on GitHub at the following links: |
| | https://github.com/SchmollerLab/SeelMito |
| | https://github.com/ElpadoCan/ChromRings |
| | |
| | RNAseq data were analyzed using the R packages QuasR v1.42.1, The EdgeR package v4.0.14, gprofiler2 package v0.2.2, bowtie2 v2.4.5 and Trimmomatic v0.39 and BSgenome. Celegans.UCSC.ce10 |
| | Statistics analysis (other than RNAseq data) were calculated in R (4.3.1) or Excel (Version 2308) |
| | Plots were made in Python (3.10), Excel (Version 2308), R (4.3.1), GraphPad Prism (8.0.1) or gprofiler2 package v0.2.2. |

For manuscripts utilizing custom algorithms or software that are central to the research but not yet described in published literature, software must be made available to editors and reviewers. We strongly encourage code deposition in a community repository (e.g. GitHub). See the Nature Portfolio guidelines for submitting code & software for further information.

## Data

Policy information about <u>availability of data</u>

All manuscripts must include a <u>data availability statement</u>. This statement should provide the following information, where applicable:

- Accession codes, unique identifiers, or web links for publicly available datasets
- A description of any restrictions on data availability
- For clinical datasets or third party data, please ensure that the statement adheres to our <u>policy</u>

The RNA-seq datasets comparing fed and fasted worms have been deposited at GEO and can be accessed under GSE268926. The RNA-seq datasets measuring the effect of RNA-Pol I depletion have been deposited at GEO and can be accessed under GSE268974. The RNA-seq datasets analyzing the transcriptome following RPB-2 degradation have been deposited at bioproject and can be accessed under PRJNA1140129. Publicly available dataset used in this study: BSgenome. Celegans.UCSC.ce10 "https://bioconductor.org/packages/BSgenome.Celegans.UCSC.ce10/"
Source data have been provided in Source Data. All other data supporting the findings of this study are available from the corresponding author on reasonable request.

## Research involving human participants, their data, or biological material

Policy information about studies with <u>human participants or human data</u>. See also policy information about <u>sex, gender (identity/presentation), and sexual orientation</u> and <u>race, ethnicity and racism</u>.

| | |
|---|---|
| Reporting on sex and gender | N/A |
| Reporting on race, ethnicity, or other socially relevant groupings | N/A |
| Population characteristics | N/A |
| Recruitment | N/A |
| Ethics oversight | N/A |

Note that full information on the approval of the study protocol must also be provided in the manuscript.

# Field-specific reporting

Please select the one below that is the best fit for your research. If you are not sure, read the appropriate sections before making your selection.

☒ Life sciences ☐ Behavioural & social sciences ☐ Ecological, evolutionary & environmental sciences

For a reference copy of the document with all sections, see nature.com/documents/nr-reporting-summary-flat.pdf

# Life sciences study design

All studies must disclose on these points even when the disclosure is negative.

| | |
|---|---|
| Sample size | No statistical methods were used to predetermine sample size. Every experimental repetition included parallel imaging of multiple worms, resulting in samples sizes of mostly in the range of tens to hundreds foci, nuclei or nucleolus scored, as commonly done in the field (all n and N values are in Supplementary Table 2). Therefore, sample sizes were chosen based on previous literature & what is common practice in the field : <br> - Meister P, Towbin BD, Pike BL, Ponti A, Gasser SM. The spatial dynamics of tissue-specific promoters during C. elegans development. Genes Dev. 2010 Apr 15;24(8):766-82. doi: 10.1101/gad.559610. PMID: 20395364; PMCID: PMC2854392 <br> - Cabianca DS, Muñoz-Jiménez C, Kalck V, Gaidatzis D, Padeken J, Seeber A, Askjaer P, Gasser SM. Active chromatin marks drive spatial sequestration of heterochromatin in C. elegans nuclei. Nature. 2019 May;569(7758):734-739. doi: 10.1038/s41586-019-1243-y. Epub 2019 May 22. PMID: 31118512 |
| Data exclusions | No data was excluded |
| Replication | Every experiment was reliably reproduced with 2 to 5 biological replicates, as specified for each experiment in the figure legends or methods |
| Randomization | No method of randomization was used as this is not relevant to the field of study. All samples were allocated to a group according to their genotype or treatment. However, individual worms were selected randomly for each experiment. |
| Blinding | Investigators were not blinded to group allocation of samples (genotype/treatment) during most data collection and analysis. However, key |

| Blinding | findings (chromatin reorganization in fasting, RNA Pol I-AID) were reproduced by at least an additional independent investigator who was blinded to group identity. For qPCR, the investigator was blinded to group identity. For RNA-seq, correlation and clustering was performed for all samples and replicas at the same time, regardless of group allocation. After that, the investigator grouped the samples according to genotype/treatment to be able to average the results across 4 independent biological replicas. |

# Reporting for specific materials, systems and methods

We require information from authors about some types of materials, experimental systems and methods used in many studies. Here, indicate whether each material, system or method listed is relevant to your study. If you are not sure if a list item applies to your research, read the appropriate section before selecting a response.

## Materials & experimental systems

| n/a | Involved in the study |
|---|---|
| ☒ | ☐ Antibodies |
| ☒ | ☐ Eukaryotic cell lines |
| ☒ | ☐ Palaeontology and archaeology |
| ☐ | ☒ Animals and other organisms |
| ☒ | ☐ Clinical data |
| ☒ | ☐ Dual use research of concern |
| ☒ | ☐ Plants |

## Methods

| n/a | Involved in the study |
|---|---|
| ☒ | ☐ ChIP-seq |
| ☒ | ☐ Flow cytometry |
| ☒ | ☐ MRI-based neuroimaging |

## Animals and other research organisms

Policy information about studies involving animals; ARRIVE guidelines recommended for reporting animal research, and Sex and Gender in Research

| Laboratory animals | Caenorhabditis elegans (variant Bristol) hermaphrodites were used in this study. The developmental stages used are indicated in the manuscript ( mostly L1s and 1 day old adults).<br>The strains (involved in the study are listed in Supplementary Table 1, together with their exact genotypes. |
| --- | --- |
| Wild animals | No wild animals were used in this study |
| Reporting on sex | Hermaphrodite were used for all experiments.<br>Males were used only to generate new strains via crossing. |
| Field-collected samples | This study did not involve samples collected from the field |
| Ethics oversight | As a non-vertebrate, C. elegans does not fall under the Directive 2010/63/EU of the European Parliament and of the Council of 22 September 2010 on the protection of animals used for scientific purposes:<br>http://eur-lex.europa.eu/LexUriServ/LexUriServ.do?uri=OJ:L:2010:276:0033:0079:en:PDF<br>Thus, no ethical approval or guidance was required for this study |

Note that full information on the approval of the study protocol must also be provided in the manuscript.

## Plants

| Seed stocks | No seed stock or plant material was used in this study. |
| --- | --- |
| Novel plant genotypes | N/A |
| Authentication | N/A |

