## [Peer Review File · Nature Cell Biology]

Peer Review Information

Journal: Nature Cell Biology

Manuscript Title: Fasting shapes chromatin architecture through an mTOR/RNA Pol I axis

Corresponding author name(s): Dr Daphne Selvaggia Cabianca

Editorial Notes:

Reviewer Comments & Decisions:

Decision Letter, initial version:
--

*Please delete the link to your author homepage if you wish to forward this email to co-authors.

Dear Dr Cabianca,

Thank you for submitting your manuscript, "An mTOR/RNA Pol I axis shapes chromatin architecture in response to fasting", to Nature Cell Biology. Your manuscript has now been seen by 3 referees, who are experts in development, chromatin, *C. elegans* (Referee #1); chromatin, nucleolus (Referee #2); and TOR, *C. elegans* (Referee #3). As you will see from their comments (attached below) they found this work of potential interest but have raised substantial concerns that in our view would need to be addressed with considerable revisions before we can consider publication in Nature Cell Biology.

Nature Cell Biology editors discuss the referee reports in detail within the editorial team, including the chief editor, to identify key referee points that should be addressed with priority, and requests that are overruled as being beyond the scope of the current study. To guide the scope of the revisions, I have listed these points below. Our standard revision period is six months, and we are committed to providing a fair and constructive peer-review process, so please feel free to contact me if you would like to discuss any of the referee comments further or if you anticipate any issues or delays addressing the reviews.

I should stress that the reviewers' concerns indicate that the findings are relatively preliminary in that the changes in chromatin organization require further studies. These are significant concerns from experts in the areas covered by this work that should be addressed experimentally and thoroughly; reconsideration of the study for this journal and re-engagement of referees will depend on the strength of these revisions. In particular, it would be essential to address the following concerns:

A- The reviewers found the changes in chromatin architecture striking, but also felt that these analyses were limited without studies of the mechanisms underlying the changes, the cell types or tissues where this occurs, and the physiological role of the reorganization. We agree with the referees that additional tissues and cell types should be examined to better understand this phenomenon and we feel that it's essential to shed light on its physiological significance. Lastly, while we understand that providing mechanistic insight into this reorganization may be somewhat challenging, we encourage you to add data along these lines if available:

Rev#2 "However, the data remained mainly descriptive and correlative and there has been no attempt to determine (1) how this process occurs, (2) why it only happens in intestine cells, and (3) which is the biological significance of this chromatin alteration."

"Figure 3. The authors should measure whether chromatin rearrangements upon auxin-mediated depletion of Pol I occur in other cell types. Moreover, they should compare nucleolar size and rRNA transcription between different cell types in fed and fasting conditions. This might help to clarify whether and how chromatin in intestine cells is different or respond differently to nucleolus signal relative to other cell types and find out the differences."

"Figure 5. The authors need to measure rRNA transcription and nucleolar size upon mTOR inhibition and its constitutive activation in intestine and other cells. Moreover, they need to show whether other cell types have similar chromatin rearrangements under these conditions."

Rev#3: "Excitement of this finding would be stronger if the phenomenon could be linked to a physiological phenotype (e.g. fasting/refeeding adaptation), downstream transcriptional changes resulting from the chromatin reorganization event can be identified, and/or if the same type of genome reorganization could be seen in other model systems."

Rev#3 major comment #1

B- In addition, further analyses of the nucleolus and phenotypes are needed, as per:

Rev#1 E#1

Rev#2 "The analysis of the nucleolus should have been a central point of this study, however, measurements of nucleoli were only limited to their size and only in intestine cells. Does nucleolar size change in the other cells upon Pol I depletion and mTOR impairment and constitutive activation? Or it is only occurring in intestine cells? Is the size of nucleoli in intestine cells correlating with chromatin rearrangements? For example, is there still a chromatin ring in intestine cells which upon fasting still contain a large nucleolus? An important analysis which has not been performed is the quantification of rRNA levels in fed vs fasted cells and the other conditions and between the different cell types. All these analyses would have helped in dissecting the mechanisms related to nucleoli. Finally, which is the consequence of this chromatin arrangement? Is globally gene expression downregulated? And does it occur only in intestine cells?";

Rev#2 other major points, "Lane 109.." paragraph; "Fig 2g..." paragraph, "Lanes 169-170..." paragraph

Rev#3 major comments #2-3, minor comments #1, 2, 3, 4

C- The reviewers' other comments about the need to strengthen existing data, provide controls (including for depletion experiments), requests for text edits and additional information or discussion, should also be addressed in full.

D- Finally, please pay close attention to our guidelines on statistical and methodological reporting (listed below) as failure to do so may delay the reconsideration of the revised manuscript. In particular, please provide:

- a Supplementary Figure including unprocessed images of all gels/blots in the form of a multi-page pdf file. Please ensure that blots/gels are labeled and the sections presented in the figures are clearly indicated.
- a Supplementary Table including all numerical source data in Excel format, with data for different figures provided as different sheets within a single Excel file. The file should include source data giving rise to graphical representations and statistical descriptions in the paper and for all instances where the figures present representative experiments of multiple independent repeats, the source data of all repeats should be provided.

We would be happy to consider a revised manuscript that would satisfactorily address these points unless a similar paper is published elsewhere or is accepted for publication in Nature Cell Biology in the meantime.

- ensure that it conforms to our format instructions and publication policies (see below and <https://www.nature.com/nature/for-authors>).
- provide a point-by-point rebuttal to the full referee reports verbatim, as provided at the end of this letter.
- provide the completed Reporting Summary (found here <https://www.nature.com/documents/nr-reporting-summary.pdf>). This is essential for reconsideration of the manuscript will be available to editors and referees in the event of peer review. For more information see <http://www.nature.com/authors/policies/availability.html> or contact me.

When submitting the revised version of your manuscript, please pay close attention to our [href="https://www.nature.com/nature-portfolio/editorial-policies/image-integrity">Digital Image Integrity Guidelines](https://www.nature.com/nature-portfolio/editorial-policies/image-integrity). and to the following points below:

- that unprocessed scans are clearly labelled and match the gels and western blots presented in figures.
- that control panels for gels and western blots are appropriately described as loading on sample

processing controls

-- all images in the paper are checked for duplication of panels and for splicing of gel lanes.

Nature Cell Biology is committed to improving transparency in authorship. As part of our efforts in this direction, we are now requesting that all authors identified as 'corresponding author' on published papers create and link their Open Researcher and Contributor Identifier (ORCID) with their account on the Manuscript Tracking System (MTS), prior to acceptance. ORCID helps the scientific community achieve unambiguous attribution of all scholarly contributions. You can create and link your ORCID from the home page of the MTS by clicking on 'Modify my Springer Nature account'. For more information please visit www.springernature.com/orcid.

This journal strongly supports public availability of data. Please place the data used in your paper into a public data repository, or alternatively, present the data as Supplementary Information. If data can only be shared on request, please explain why in your Data Availability Statement, and also in the correspondence with your editor. Please note that for some data types, deposition in a public repository is mandatory - more information on our data deposition policies and available repositories appears below.

[Redacted]

We hope that you will find our referees' comments and editorial guidance helpful. Please do not hesitate to contact me if there is anything you would like to discuss. Thank you again for considering our journal for your work,

Best wishes,

Melina

Melina Casadio, PhD
Senior Editor, Nature Cell Biology
ORCID ID: <https://orcid.org/0000-0003-2389-2243>

Reviewers' Comments:

Reviewer #1:

Remarks to the Author:

A. Summary of the key results

The authors use fluorescently tagged DNA to look at chromatin localization with respect to the nucleolus, which is centrally located in intestine cells. In figure 1, they observe that upon starvation the chromatin is organized into two "rings. The inner one surrounds the nucleolus; outer one is juxtaposed to the edge of the nuclear membrane. In figure 2, they observe that expressed genes fall between the denser chromatin forming the rings and the expression correlates with location. In figure 3, the authors show that the rings recover after feeding, and this requires Pol I transcription, and not Pol II or III. Importantly, in figure 4 the authors show that if you deplete Pol I in presence of food, the rings form, thus this organization requires Pol I, meaning Pol I is downstream of the signaling from food. In figure 5, the authors use DAF15 depletion and a TORC constitutive mutant to show that the signaling is from TORC.

B. Originality and significance: if not novel, please include reference

This is an interesting observation and novel to my knowledge. Such large scale change to genome organization in intestine cells, induced by food could start a new line of research to understand the mechanism by which the organization occurs, and the cause and consequence of the organization in gene expression and worm survival/response to food.

C. Data & methodology: validity of approach, quality of data, quality of presentation

The data and figures are well done and well presented.

D. Appropriate use of statistics and treatment of uncertainties

The results appear clean. Uncertainties are not discussed but variation is represented in the plots.

E. Conclusions: robustness, validity, reliability

1. The ring organization in the adult intestine nuclei look quite different from that of the L1. The current text reads as if they are similar, and I encourage the authors to take a closer look at the data and compare quantitatively if possible.

2. Given there is no effect by RNA Pol II and RNA Pol III depletion on the concentric rings, it is important to show that the auxin depletion has indeed reduced/eliminated transcription of Pol II / Pol III genes. Another interpretation that the auxin depletion did not eliminate the activity of Pol II and Pol III very much, and that is why there is no effect on genome organization. At the minimum, the authors should do this for Pol II, using the reporters they already have. I understand this may be more difficult for pol III, and given it is the catalytic subunit that is being degraded and shown to decrease using the fluorescent tag, it is correct to assume transcription is reduced/eliminated.

3. There is no result that validates that auxin inducible depletion of Lin15 works, and to what extent.

4. It is not clear if one can observe the rings when the nucleolus is large in the raga-gf mutant. It is a bit of circular argument. Perhaps the authors want to comment on that. If they feel it interrupts the flow of the paper, it is OK. Because Pol I depletion in the mutant clearly demonstrates that raga is upstream of Pol I, supporting the model.

F. Suggested improvements: experiments, data for possible revision

1. Some of the figures are used for making statements but put in the supplemental figure. I encourage them to be placed in the main. For example the lack of rings in heat and cold shock, and the rings in adult intestine. Similarly, extended figure 2 can be easily put in figure 2, where the readers can see the lack of position change for the heterochromatin reporter and contrast that to that of the euchromatic. This is not too important if there is no room in a page or there are figure restrictions for the format of the article.

2. The authors should validate that rpb-2 depletion leads to reduced transcription and DAF15 protein depletion occurs in their experimental set up.

G. References: appropriate credit to previous work?

There are a limited number of references. Given that, I don't know if there was immediately relevant papers that were left out.

H. Clarity and context: lucidity of abstract/summary, appropriateness of abstract, introduction and conclusions.

This is a well-written and well-presented paper with intriguing observations and results that would be of interest to a wide range of researchers.

Reviewer #2:

Remarks to the Author:

Al-Refaie et al. described an intriguing rearrangement of chromatin organization in *C. elegans* intestine cells during fasting. This spatial reorganization consists in the formation of two "concentric rings" of chromatin, one at the nuclear periphery and the other around nucleoli. The data also showed that this genome reorganization is reversible upon refeeding and depends on the expression of Pol I whereas depletion of Pol II or III subunits has no effect. Moreover, the data indicated that the nucleolar volume in intestine cells decreases upon fasting. Finally, the authors suggested that mTOR is implicated in these alterations in genome architecture since mTOR inhibition (i.e., DAF-15 downregulation) causes similar alterations in chromatin organization under fed conditions whereas constitutive activation through the expression of RAGA-1GF mutant antagonized the reorganization of chromatin in intestine cells.

The changes in genome architecture in intestine cells upon fasting is a novel and beautiful phenotype that suggests that nutritional stimuli can have an impact on chromatin architecture. However, the data remained mainly descriptive and correlative and there has been no attempt to determine (1) how this process occurs, (2) why it only happens in intestine cells, and (3) which is the biological significance of this chromatin alteration. The analysis of the nucleolus should have been a central point of this study, however, measurements of nucleoli were only limited to their size and only in intestine cells. Does

nucleolar size change in the other cells upon Pol I depletion and mTOR impairment and constitutive activation? Or it is only occurring in intestine cells? Is the size of nucleoli in intestine cells correlating with chromatin rearrangements? For example, is there still a chromatin ring in intestine cells which upon fasting still contain a large nucleolus? An important analysis which has not been performed is the quantification of rRNA levels in fed vs fasted cells and the other conditions and between the different cell types. All these analyses would have helped in dissecting the mechanisms related to nucleoli. Finally, which is the consequence of this chromatin arrangement? Is globally gene expression downregulated? And does it occur only in intestine cells?

Other major points

Lane 109: "we find that the signal is low at the edge of the nucleolus." and Extended Data Fig. 1s. To better show the data on chromatin proximity to nucleoli, the x-axis should show the radius of nucleoli (i.e., nucleolus borders).

Fig. 2g. I think that gene expression levels of the euchromatin reporter with cherry-histone should be performed using RNA-FISH or other systems to detect mRNA and not by measurements of protein levels. Indeed, if fasting reduces rRNA synthesis, as mentioned by the authors, this should cause downregulation of ribosome production with a consequent reduction of protein synthesis. mRNA reporter measurements will also allow to directly determine whether loci contacting the nuclear periphery or nucleolus are active or repressed.

Lanes 169-170. "This argues that nucleolar size is regulated very early in the response to refeeding, which ultimately leads to the dispersion of the chromatin rings".

There is no data showing that nucleolar size "leads to the dispersion of the chromatin rings". The experiment has only showed that nucleolar size decreased upon fasting and increase upon refeeding.

Figure 3. The authors should measure whether chromatin rearrangements upon auxin-mediated depletion of Pol I occur in other cell types. Moreover, they should compare nucleolar size and rRNA transcription between different cell types in fed and fasting conditions. This might help to clarify whether and how chromatin in intestine cells is different or respond differently to nucleolus signal relative to other cell types and find out the differences.

Lanes 210-211. "we inhibited RNA pol I transcription by the addition of Actinomycin D". Depending on the concentration, ActD can also inhibit Pol II and Pol III. The authors should clarify whether the treatment they used is specific to Pol I.

Figure 5. The authors need to measure rRNA transcription and nucleolar size upon mTOR inhibition and its constitutive activation in intestine and other cells. Moreover, they need to show whether other cell types have similar chromatin rearrangements under these conditions.

Lanes 255-256. "expressing RAGA-1GF does not antagonize the formation of the concentric rings upon RNA Pol I inhibition (Fig. 5j-l)...". The authors should show the expression levels of RAGA-1GF upon Pol depletion since the decrease of ribosome production might have an impact of protein levels.

Minor points

The authors have often referred to quite old literature about regulation of rRNA transcription upon nutrient deprivation. It is not wrong, but maybe they can update it a bit.

Lanes 151-152 "these results indicate that the perinuclear and perinucleolar chromatin rings are

repressive compartments.”. Maybe here the authors should comment that their data are in line with previous works showing that both nuclear periphery and perinucleolar compartment are repressive.

Reviewer #3:

Remarks to the Author:

Reviewer assessment NCB-LE51990

In the manuscript “An mTOR/RNA Pol I axis shapes chromatin architecture in response to fasting” by Al-Refaie et al., the authors show that fasting induces dynamic chromatin reorganization in the intestine of *C. elegans*. The authors describe a fasting-induced double-spherical structure where chromatin redistributes to the nuclear lamina and nucleolar periphery. The genome reorganization is recapitulated by knockdown of RNA Pol I, but not Pol II nor Pol III. The authors propose an mTOR-Pol I axis for large-scale genome reorganization in response to nutrient deprivation.

The main significant finding from this study is the description of a stereotyped chromatin reorganization event in response to fasting. This is a visually captivating reorganization event that has gone unnoticed in the *C. elegans* field, and adds an interesting layer of regulation for the cellular response to fasting. The finding will likely spark future interest in large-scale chromatin reorganization events across model organisms and contexts. Excitement of this finding would be stronger if the phenomenon could be linked to a physiological phenotype (e.g. fasting/refeeding adaptation), downstream transcriptional changes resulting from the chromatin reorganization event can be identified, and/or if the same type of genome reorganization could be seen in other model systems. The findings regarding the upstream regulators, especially Pol I is intriguing as it links the chromatin reorganization event to rDNA transcription. The link with the mTOR pathway supports previous publications linking mTOR-Pol I-rDNA transcription (Tsang et al, 2003 EMBO; Mayer et al, 2004 Gene Dev). The study complements studies reporting links between mTOR/nutrient restriction with chromatin reorganization (Lu et al, 2021 eLife) and nucleolar reorganization (Tiku et al, 2017 Nat. Comm).

The data is of high quality and presented well. Some potential points for improvement are outlined below.

Major comments:

1. Is the double-sphere reorganization event really a tissue-specific fasting event?

- A later time point should be checked to see whether the intestine is the first responder, or the only responder. It could be possible that other cell types would reorganize after more prolonged periods in starvation. L1 arrested animals can survive for longer than a month, so a time point of 7 days could be a second time point to check whether other cell types also display a similar or possibly different type of genome reorganization. Alternatively/additionally, if nucleolar size is quantified in other tissues, it would indicate whether other cell-types are already responding to fasting but nonetheless is not showing the double-ring structure.
- Other cell-types besides the intestine should be shown in the Pol I knockdown, and whether this intervention results in a double-sphere structure.
- Testing the eat-2 mutant for its effect on the double-ring phenotype would indicate whether it is fasting that is required, or reduced feeding would also result in this chromatin reorganization.

2. Is the outer nucleolar area repressive?

- The statement that “The fasting-induced chromatin rings are repressive compartments,” (section title line 113) seems strong based on the experiments provided. Although it is reasonable to

hypothesize that the nucleolar periphery is in general repressive, a similar reporter construct using a different promoter (vs. *pha-4*), different gene/protein (vs. *HIS-24*), different genomic location (current location unknown) could show different results. The choice of *pha-4* promoter makes it difficult to generalize, as *pha-4* is required for dietary restriction-mediated longevity (Panowski et al., Nature 2007). The regulation and transcriptional output of the *pha-4* promoter may therefore be different from a general housekeeping gene such as actin, or a gene that may be strongly induced during fasting (e.g. *sod-3*).

- An orthogonal experiment of looking at repressive and active histone marks at the nucleolar periphery would help, although it seems experimentally difficult due to the labile nature of the double-ring structure. Something that could be possible would be to check in mutants of repressive chromatin H3K27me3, H3K9me3, whether knockdown/knockout of repressive chromatin marks would affect the double-ring reorganization event and/or the expression level change of *HIS-24::mCherry* that is correlated with the localization of the euchromatin reporter.
- Alternatively/additionally, co-localization of the ring structure could be checked with fluorescent reporter strains expressing enzymes or proteins that localize to heterochromatin/repressive chromatin (e.g. HPL, MES proteins) vs. euchromatic regions (potentially MRG-1 identified previously by the author(s)).

3. Other

- The authors nicely demonstrate that refeeding rapidly causes a reorganization of the heterochromatin and expansion of nucleolar size. What about the kinetics of fasting? Does it also cause a rapid reorganization of heterochromatin and nucleolus, or is it a slower process?

Minor comments:

1. In Fig. 3, the authors show that nucleolar size changes precede the concentric spheres to be dispersed. They argue that the nucleolar size change "leads to the dispersion of the chromatin rings (lines 168-170). The requirement for nucleolar shrinkage in chromatin reorganization could be tested in a *ncl-1* mutant, which retains a large nucleolus in response to dietary restriction.
2. *raga-1* loss of function (*ok386* or *ok701*) or *let-363* knockdown would add further support for the mTOR pathway, while using alleles/conditions that have been characterized in other contexts. Relatedly, *daf-15/raptor AID* causes only a partial reorganization of the heterochromatin. Perhaps other pathways contribute? In particular, the authors should consider testing MAPK signaling.
3. DIC insets would be helpful in visualizing the images (Fig 1 and others). For example, we would be able to see if fasting-induced morphological changes are occurring such as nucleolar shrinkage, and be able to compare among different tissues (whether nucleolar shrinkage and the double-ring occur together or not in the different tissues).
4. Is pol I expression or localization changed by fasting/refeeding? Also it would be helpful to explain RNA pol I somewhere in the discussion as a polymerase that mainly acts on rDNA, so the significance can be conveyed to the reader.
5. Worm strain information of the GW429 strain is listed to be 256x copies (Table S1), while in figure 2 it states it is reduced copy number. This should be clarified.
6. It would be helpful if the physiological difference in copy number could be explained regarding ~30 copies vs ~300 copies. The general threshold for heterochromatinization of repetitive DNA should be discussed/explained.
7. It would assist the reader if the observation that the heterochromatin reporter is unchanged from fasting is more carefully explained. A casual reader may expect to see a heterochromatin reporter at the nucleolar periphery, if as is proposed, the nucleolar periphery is repressive. However the

heterochromatin reporter used in this paper is not a reporter that labels all heterochromatin, but a couple of loci that are expected to be heterochromatic. A more careful explanation of the reporter and results could help in avoiding misinterpretations.

8. The same goes for the euchromatic reporter. The euchromatic reporter is from previous reports expected to be euchromatic in the fed condition. A reader may think that if the euchromatic reporter is moving to the nucleolar periphery, it may indicate that it is active there, whereas it is potentially heterochromatic/repressive at the nucleolar periphery.

9. Statistical tests could not be found for some figures (e.g. Ext. Fig 1 C, D). The profiles look different from fed to fasted in hypodermis and muscle, but whether these are significantly different should be tested/shown.

10. I am not sure if "concentric rings" is the right term to use for this phenomenon, as this view is based on a 2D slice. In reality they are concentric spheres.

11. The abstract should mention that the reorganization event is limited to the intestine.

12. I would suggest that the title would emphasize the chromatin architecture first, as this is the main point of emphasis of the paper. For example something like "Fasting-induced chromatin reorganization via mTOR/RNA Pol I axis"

13. In terms of the flow of the paper, I felt that the Pol I–double ring structure figure (Fig 4) and mTOR–double ring structure figure (Fig 5) might flow better if it came right after Fig 2, and preceded the refeeding figure (Fig 3). This is just a suggestion based on personal preference.

AUTHOR AFFILIATIONS – should be denoted with numerical superscripts (not symbols) preceding the

names. Full addresses should be included, with US states in full and providing zip/post codes. The corresponding author is denoted by: "Correspondence should be addressed to [initials]."

Methods should be written concisely, but should contain all elements necessary to allow interpretation and replication of the results. As a guideline, Methods sections typically do not exceed 3,000 words. The Methods should be divided into subsections listing reagents and techniques. When citing previous methods, accurate references should be provided and any alterations should be noted. Information must be provided about: antibody dilutions, company names, catalogue numbers and clone numbers for monoclonal antibodies; sequences of RNAi and cDNA probes/primers or company names and catalogue numbers if reagents are commercial; cell line names, sources and information on cell line identity and authentication. Animal studies and experiments involving human subjects must be reported in detail, identifying the committees approving the protocols. For studies involving human

subjects/samples, a statement must be included confirming that informed consent was obtained. Statistical analyses and information on the reproducibility of experimental results should be provided in a section titled "Statistics and Reproducibility".

All Nature Cell Biology manuscripts submitted on or after March 21 2016 must include a Data availability statement as a separate section after Methods but before references, under the heading "Data Availability". For Springer Nature policies on data availability see <http://www.nature.com/authors/policies/availability.html>; for more information on this particular policy see <http://www.nature.com/authors/policies/data/data-availability-statements-data-citations.pdf>. The Data availability statement should include:

- Accession codes for primary datasets (generated during the study under consideration and designated as "primary accessions") and secondary datasets (published datasets reanalysed during the study under consideration, designated as "referenced accessions"). For primary accessions data should be made public to coincide with publication of the manuscript. A list of data types for which submission to community-endorsed public repositories is mandated (including sequence, structure, microarray, deep sequencing data) can be found here <http://www.nature.com/authors/policies/availability.html#data>.
- Unique identifiers (accession codes, DOIs or other unique persistent identifier) and hyperlinks for datasets deposited in an approved repository, but for which data deposition is not mandated (see here for details <http://www.nature.com/sdata/data-policies/repositories>).
- At a minimum, please include a statement confirming that all relevant data are available from the authors, and/or are included with the manuscript (e.g. as source data or supplementary information), listing which data are included (e.g. by figure panels and data types) and mentioning any restrictions on availability.
- If a dataset has a Digital Object Identifier (DOI) as its unique identifier, we strongly encourage including this in the Reference list and citing the dataset in the Methods.

We recommend that you upload the step-by-step protocols used in this manuscript to the Protocol Exchange. More details can found at www.nature.com/protocolexchange/about.

All imaging data should be accompanied by scale bars, which should be defined in the legend. Cropped images of gels/blots are acceptable, but need to be accompanied by size markers, and to retain visible background signal within the linear range (i.e. should not be saturated). The boundaries of panels with low background have to be demarked with black lines. Splicing of panels should only be

considered if unavoidable, and must be clearly marked on the figure, and noted in the legend with a statement on whether the samples were obtained and processed simultaneously. Quantitative comparisons between samples on different gels/blots are discouraged; if this is unavoidable, it should only be performed for samples derived from the same experiment with gels/blots were processed in parallel, which needs to be stated in the legend.

The total number of Supplementary Figures (not including the “unprocessed scans” Supplementary Figure) should not exceed the number of main display items (figures and/or tables (see our Guide to Authors and March 2012 editorial <http://www.nature.com/ncb/authors/submit/index.html#suppinfo>; <http://www.nature.com/ncb/journal/v14/n3/index.html#ed>). No restrictions apply to Supplementary Tables or Videos, but we advise authors to be selective in including supplemental data.

GUIDELINES FOR EXPERIMENTAL AND STATISTICAL REPORTING

REPORTING REQUIREMENTS – We are trying to improve the quality of methods and statistics reporting in our papers. To that end, we are now asking authors to complete a reporting summary

that collects information on experimental design and reagents. The Reporting Summary can be found here <https://www.nature.com/documents/nr-reporting-summary.pdf>) If you would like to reference the guidance text as you complete the template, please access these flattened versions at <http://www.nature.com/authors/policies/availability.html>.

Author Rebuttal to Initial comments

Dear Melina,

Thank you for your patience and for providing us with the opportunity to improve our paper based on your and the reviewers' comments.

We have successfully conducted nearly all the experiments recommended by the reviewers and updated the manuscript with 25 new panels in the main figures and 41 in the extended data figures. The new sentences that describe and discuss them are shown underlined in the main text.

We now provide extensive information on changes in nucleolar size, rRNA transcription and 3D genome organization in three different tissues—intestine, hypoderm, and muscle—under three distinct conditions: fasting, mTOR inhibition, and RNA Pol I depletion. Additionally, we performed intestine-specific RNA-seq to gain insights into the physiological role of 3D genome reorganization. These new experiments, along with the suggested controls and other recommendations from the reviewers, have enabled us to greatly improve the paper.

We hope you now find it suitable for publication in Nature Cell Biology.

Best wishes,

Daphne Cabianca

Reviewer #1:

Remarks to the Author:

A. Summary of the key results

The authors use fluorescently tagged DNA to look at chromatin localization with respect to the nucleolus, which is centrally located in intestine cells. In figure 1, they observe that upon starvation the chromatin is organized into two “rings. The inner one surrounds the nucleolus; outer one is juxtaposed to the edge of the nuclear membrane. In figure 2, they observe that expressed genes fall between the denser chromatin forming the rings and the expression correlates with location. In figure 3, the authors show that the rings recover after feeding, and this requires Pol I transcription, and not Pol II or III. Importantly, in figure 4 the authors show that if you deplete Pol I in presence of food, the rings form, thus this organization requires Pol I, meaning Pol I is downstream of the signaling from food. In figure 5, the authors use DAF15 depletion and a TORC constitutive mutant to show that the signaling is from TORC.

B. Originality and significance: if not novel, please include reference

This is an interesting observation and novel to my knowledge. Such large scale change to genome organization in intestine cells, induced by food could start a new line of research to understand the

mechanism by which the organization occurs, and the cause and consequence of the organization in gene expression and worm survival/response to food.

We thank the reviewer for recognizing the novelty of our work and giving this positive feedback.

C. Data & methodology: validity of approach, quality of data, quality of presentation

The data and figures are well done and well presented.

D. Appropriate use of statistics and treatment of uncertainties

The results appear clean. Uncertainties are not discussed but variation is represented in the plots.

E. Conclusions: robustness, validity, reliability

1. The ring organization in the adult intestine nuclei look quite different from that of the L1. The current text reads as if they are similar, and I encourage the authors to take a closer look at the data and compare quantitatively if possible.

The reviewer is right. While the two rings in L1 and adult intestinal cells are qualitatively comparable, by quantifying the amplitude of the peaks in the radial chromatin intensity plots, we found that the HIS-72/H3.3 ratio of the outer/inner ring signal is lower in adults compared to L1s, which means that the outer ring is weaker in adults. To clarify this and to address the reviewer's concern we have now added this quantification in new Extended Data Fig. 1n and referred to it in the text, lines 86-87.

2. Given there is no effect by RNA Pol II and RNA Pol III depletion on the concentric rings, it is important to show that the auxin depletion has indeed reduced/eliminated transcription of Pol II / Pol III genes. Another interpretation that the auxin depletion did not eliminate the activity of Pol II and Pol III very much, and that is why there is no effect on genome organization. At the minimum, the authors should do this for Pol II, using the reporters they already have. I understand this may be more difficult for pol III, and given it is the catalytic subunit that is being degraded and shown to decrease using the fluorescent tag, it is correct to assume transcription is reduced/eliminated.

The reviewer is correct. For RNA Pol III, we tagged the catalytic subunit RPC-1 with both mNG and AID, to directly monitor its degradation upon addition of auxin (Extended Data Fig.5e, f). For RNA Pol II, we did not tag the catalytic subunit AMA-1, but RPB-2 the homolog of yeast Rpb2 and human POL2RB. To, as requested by the referee, confirm that RNA Pol II transcription is inhibited when RPB-2 is degraded:

1. We monitored the development of RPB-2-AID L1 larvae exposed to 1 mM auxin compared to TIR1 only expressing animals and found that in contrast to the negative control, 100% of the animals are arrested at the first larval stage, in agreement to what expected in absence of RNA Pol II activity (see Figure for the referee below).

2. We determined the expression of RNA Pol II targets upon RPB-2 degradation. In particular, we collected RNA from the same number of control (TIR1 only) and RPB-2-AID larvae exposed to 1 mM auxin for 3h, to induce RPB-2 degradation. Next, we randomly selected 9 genes for having one intron longer than 150 bp, to allow for optimal qPCR primer design, and quantified pre-mRNA expression by RT-qPCR. We found that the expression of 7 out of 9 pre-mRNAs is reduced when RPB-2 is degraded after 3 hours on auxin, suggesting that transcription by RNA Pol II is indeed inhibited but that some RNAs are more stable than others. In fact, when RBP-2-AID worms are exposed to auxin for 96 hours, also the two remaining genes are downregulated, together with a strong global reduction of the transcriptome. We have now added these control experiments in new Extended Data Fig. 6d, e.

Altogether these new results show clearly that RNA Pol II transcription is inhibited when RBP-2 is degraded, supporting previous structural work showing that RPB-2/Rpb2, together with AMA-1/Rpb1, is required to constitute the active center of the enzyme (Cramer et al., 2000, Cramer et al., 2001).

3. Lastly, we monitored 3D chromatin organization in wild type animals exposed to α amanitin (6 hours, 25 μ g/ml), which does not affect RNA Pol I transcription, but strongly inhibits RNA Pol II (Bensaude 2011) and found that no chromatin rings are induced (new Extended Data Fig. 6f-h). These new results confirm that RNA Pol II inhibition is not sufficient to induce the chromatin ring configuration.

3. There is no result that validates that auxin inducible depletion of Lin15 works, and to what extent.

We believe that the reviewer here refers to DAF-15, as Lin15 is not investigated in this work.

To acutely deplete DAF-15/RAPTOR, we used DAF-15 tagged by AID and fused to mNeonGreen at its endogenous site. We could not, however, robustly detect the protein fusion by spinning disc confocal live imaging (data not shown). We also tried 4 different antibodies (2 recognizing mNeonGreen epitopes and 2 targeting mammalian RAPTOR) but again none of the tested antibodies gave a signal specific for DAF-15 protein levels (neither by immuno-fluorescence nor western blotting). We conclude that DAF-15 is expressed at very low levels and the antibodies used are not specific enough to detect it. However, knowing that RAPTOR loss leads to growth defects in mammals (Kim et al., 2002) and *daf-15* mutant worms arrest their larval development as L3 larvae (Duong et al., 2020), to address the reviewer's concern, we monitored the development of DAF-15-AID worms upon auxin exposure. We found that they arrest at the L3 stage (new Extended Data Fig. 3k).

This phenotypic response recapitulating the null allele suggests that the depletion of DAF-15 through the AID system works. Importantly, we performed new experiments directly knocking down mTOR (LET-363 in worms) and found that this independent inhibition of mTOR signaling also induces a partial reorganization of chromatin into two rings in the intestine of fed animals (new Fig. 3a-c), similarly to what observed upon DAF-15 depletion. Both results confirm a role for mTOR in 3D genome organization in intestine.

4. It is not clear if one can observe the rings when the nucleolus is large in the *raga-gf* mutant. It is a bit of circular argument. Perhaps the authors want to comment on that. If they feel it interrupts the flow of the paper, it is OK. Because Pol I depletion in the mutant clearly demonstrates that *raga* is upstream of Pol I, supporting the model.

As shown in Fig. 4f, *raga-1^{GF}* intestinal cells have larger nucleoli than wild type. To address if larger nucleoli are responsible for impairing the 3D genome reorganization during fasting in these mutants (Fig. 3d-f), we measured radial chromatin distribution in wild type and *raga-1^{GF}* fasted intestinal cells selected for having i) the same large nucleolar size or ii) the same nucleolar AND nuclear area. Under both conditions the chromatin rings are detectable in wild type but not in *raga-1^{GF}* cells (new Fig. 4g-h and new Extended Data Fig. 4c-f). Thus, while a reduction in nucleolar size (induced by RNA Pol I inhibition) is likely necessary for the formation of the chromatin rings, we conclude that the size of the nucleolus *per se*, or the “space” available for chromatin inside the nucleus are not sufficient to explain the organization of the 3D genome, and other factors contribute to the altered chromatin organization in *raga-1^{GF}* mutants. We now describe these results in the main text, lines 232-240.

F. Suggested improvements: experiments, data for possible revision

5. Some of the figures are used for making statements but put in the supplemental figure. I encourage them to placed in the main. For example the lack of rings in heat and cold shock, and the rings in adult intestine. Similarly, extended figure 2 can be easily put in figure 2, where the readers can see the lack of position change for the heterochromatin reporter and contrast that to that of the euchromatic. This is not too important if there is no room in a page or there are figure restrictions for the format of the article.

As suggested, we included in main Fig. 1 the lack of chromatin rings in heat and cold shock (panels f, g), and the formation of rings in adult intestine during fasting (panels h, i). However, because we have generated new data on the effect of heterochromatin mutations on the positioning and expression of the euchromatic reporter in new Fig. 2, we did not move data from Extended Data Fig. 2 into main Fig. 2, as we think that the figure would lose focus and clarity.

6. The authors should validate that *rpb-2* depletion leads to reduced transcription and DAF15 protein depletion occurs in their experimental set up.

As outlined earlier in the response to point 2, we have now provided direct experimental evidence demonstrating that RBP-2 depletion causes RNA Pol II inhibition (by quantification of larval development and of transcription- new Extended Data Fig. 6d, e).

In the response to point 3, we described the work we performed to validate DAF-15 depletion in the DAF-15-AID strain. Briefly, while a direct quantification of DAF-15 protein levels was not possible, upon auxin exposure, we measured a developmental growth arrest of the DAF-15-AID strain (new Extended Data Fig. 3k) that mimics what previously described for *daf-15* null mutants (Duong et al., 2020). We now mention this in the revised manuscript, line 192.

Lastly, the use of *mTOR/let-363* RNAi to directly inhibit mTOR independently confirmed the role of this pathway in shaping the 3D genome of intestinal cells.

G. References: appropriate credit to previous work?

There are a limited number of references. Given that, I don't know if there was immediately relevant papers that were left out.

We now added 26 references to the Main Text and 11 to the Methods and Extended Data section, as appropriate to support the new analyses and interpretations.

H. Clarity and context: lucidity of abstract/summary, appropriateness of abstract, introduction and conclusions.

This is a well-written and well-presented paper with intriguing observations and results that would be of interest to a wide range of researchers.

We thank the reviewer for this supportive comment, and for the constructive critique.

Reviewer #2:

Remarks to the Author:

Al-Refaie et al. described an intriguing rearrangement of chromatin organization in *C. elegans* intestine cells during fasting. This spatial reorganization consists in the formation of two “concentric rings” of chromatin, one at the nuclear periphery and the other around nucleoli. The data also showed that this genome reorganization is reversible upon refeeding and depends on the expression of Pol I whereas depletion of Pol II or III subunits has not effect. Moreover, the data indicated that the nucleolar volume in intestine cells decreases upon fasting. Finally, the authors suggested that mTOR is implicated in these alterations in genome architecture since mTOR inhibition (i.e, DAF-15 downregulation) causes similar alterations in chromatin organization under fed conditions whereas constitutive activation through the expression RAGA-1GF mutant antagonized the reorganization of chromatin in intestine cells.

The changes in genome architecture in intestine cells upon fasting is a novel and beautiful phenotype that suggests that nutritional stimuli can have an impact in chromatin architecture.

We thank the reviewer for these positive remarks.

1. However, the data remained mainly descriptive and correlative and there has been no attempt to determine (1) how this process occurs, (2) why it only happens in intestine cells, and (3) which is the biological significance of this chromatin alteration.

We agree that these questions are very important, and we thank the reviewer for pointing them out. Many of the analyses suggested, and which we have performed, helped us gain fundamental new insights towards all three points and allowed us to conclude what follows:

For **point (1)** on how this 3D genome reorganization occurs, we have collected new data on how nucleolar size is regulated in all conditions tested (kinetics of fasting and refeeding, mTOR inhibition, RNA Pol I depletion) and in three different tissues. These demonstrate that a drastic reduction in nucleolar volume is critical for rings formation.

To address **point (2)** on why during fasting the reorganization is intestine specific, we performed new experiments to investigate chromatin organization and nucleolar volume in muscle and hypoderm, besides intestine, upon fasting, mTOR inhibition and RNA Pol I depletion. We have also performed FISH to quantify pre-rRNA levels in all these conditions. These experiments allowed us to uncover that cell types other than intestinal have the capacity to reorganize their genome into chromatin rings, and they (hypoderm) do so upon direct inhibition of RNA Pol I. However, during fasting this does not occur. Our new results argue that this is likely because the reduction in RNA Pol I activity induced by fasting in non-intestine tissues is not prominent enough to trigger a strong remodeling and relative drop in size of the nucleolus.

A detailed explanation of the experiments performed, and results obtained, to address points (1 and 2) can be found below, in response to the specific questions of the reviewer.

Given the known relationship between 3D genome architecture and transcription, to gain insights into the biological significance of this 3D chromatin reorganization (**point (3)**), we decided to perform RNA-seq in fed and fasted intestinal cells. Because chromatin rings form at nuclear compartments where heterochromatin accumulates (Solovei et al., 2016), we tested the role of known heterochromatin factors in rings formation. In case the 3D genome reorganization during fasting would be altered, we would compare gene expression in the affected chromatin mutant(s) to wild type. However, as shown in new Fig. 2c and new Extended Data Fig. 2b-c, the complete lack of H3K9 methylation and of H3K27me3, did not impair the reorganization of the genome into two rings during fasting. Similarly, rings persisted also upon loss of: the peripheral anchor CEC-4, the reader HPL-1, the linker histone H1.1, and the sirtuin SIR-2.1. Chromatin rings were also unaffected by the overexpression of CBP-1, homolog of the mammalian transactivator CBP/P300 (data not shown).

Since we did not observe any changes in the 3D genome organization induced by fasting when perturbing a chromatin factor, we searched for an alternative approach to identify potential genes sensitive to the chromatin ring configuration. In particular, we decided to compare gene expression changes when chromatin rings form in response to different inputs and determine whether specific sets of genes are coregulated. To this aim, we compared the gene expression changes occurring in the intestine of fasted animals (over fed), and of fed animals lacking RNA Pol I (over fed, control).

We found that, while genes downregulated when RNA Pol I is inhibited are overall not downregulated during fasting (new Fig. 7e, f), strikingly, the majority of genes that are upregulated in Pol I depleted animals, are also upregulated during fasting (new Fig. 7d, f). These co-upregulated genes are not involved in ribosome biogenesis (ribosomal, rRNA processing and nucleolar-related genes), and, thus, likely do not reflect a compensation to the reduction in RNA Pol I transcription, which occurs in both conditions. Rather, Gene Ontology enrichment shows that the co-upregulated genes are enriched in metabolic and stress response pathways.

These results suggest that the reorganization of the genome into chromatin rings might contribute to the transcriptional switch occurring in absence of nutrients, by favoring the expression of specific genes.

2. The analysis of the nucleolus should have been a central point of this study, however, measurements of nucleoli were only limited to their size and only in intestine cells. Does nucleolar size change in the other cells upon Pol I depletion and mTOR impairment and constitutive activation? Or it is only occurring in intestine cells?

We thank the reviewer for suggesting these experiments. We have now systematically quantified nucleolar volume in intestine, hypoderm and muscle in the following conditions:

- Fed and fasting
- mTOR inhibition by RNAi for *let-363* (and control)
- Pol I degradation (and control)

We found that:

1. During fasting the nucleolus shrinks before the rings are formed (new Fig. 4a-c), while during refeeding, the nucleolus enlarges before chromatin rings are fully dispersed (Fig. 5a-c).
2. By analyzing the volume of the nucleolus in three different tissues (intestine, muscle and hypoderm), and under three different conditions (fasting, mTOR inhibition and RNA Pol I depletion), we found that the nucleolus diminishes its size also in hypoderm and muscle under all of the tested conditions, albeit to different degrees. **Remarkably, a strong relative reduction in nucleolar size compared to the corresponding control is the best predictor for inducing rings formation.** This is true not only in intestine (new Fig. 4d-e and new Fig. 6g), but also in hypoderm, where upon Pol I inhibition rings are formed (new Fig. 6e, i), and the size of the nucleolus is reduced more compared to what we measured during fasting in the same tissue, and mirrors the relative change in size observed in intestinal cells (new Fig. 6g-h).
3. Muscle cells do not reorganize their genome into chromatin rings under any of the conditions tested (fasting, mTOR inhibition nor RNA Pol I depletion), and nucleolar size decreases less than in hypoderm and intestine.

In sum, these new data show that nucleoli are not affected exclusively in intestine. Muscle and hypoderm also respond to both fasting and mTOR signaling with a reduction in nucleolar size, albeit the effect is consistently weaker compared to what we measured in intestine under the same perturbations. Interestingly, the acute inhibition of RNA Pol I causes nucleoli in hypoderm to shrink, in relative terms, like intestinal cells, and this correlates with the formation of chromatin rings in hypoderm. The reduction in size of the nucleolus in muscle is less pronounced and there are no rings formed. This suggests that a strong relative reduction in nucleolar volume correlates with ring formation in multiple tissues.

We now included these data in a new paragraph in the revised manuscript dedicated to describing changes of the nucleolus, entitled “Remodeling of the nucleolus correlates with the reorganization of chromatin into two rings”, in lines 222-229. Additionally, we also discuss changes in the nucleolus upon RNA Pol I depletion in lines 333-338.

We have now determined that mTOR inhibition in fed animals induces a partial reorganization of chromatin into rings *only* in intestine and not in muscle or hypoderm (new Fig. 3a-c; Extended Data Fig. 3h). For this reason, we did not extend the analysis of nucleolar volume upon mTOR constitutive activation (in *raga-1^{GF}* mutants) beyond intestinal cells, where *raga-1^{GF}* mutants display larger nucleoli (Fig. 4f).

3. Is the size of nucleoli in intestine cells correlating with chromatin rearrangements? For example, is there still a chromatin ring in intestine cells which upon fasting still contain a large nucleolus?

From the experiment of Fig. 4f, we selected the 50 wild type fasted intestinal cells possessing the largest nucleolus (highlighted in red in the boxplot on the left) and measured their radial chromatin distribution. We found that they form chromatin rings (new Fig. 4h).

On the contrary, 50 *raga-1^{GF}* fasted intestinal cells selected for having the same of nucleolar size as in wild type (in red in the boxplot on the left and in new Fig. 4g) do not form rings. Thus, while during fasting and mTOR inhibition the volume of the nucleolus in intestine is drastically reduced (new Fig. 4d, e) and correlates with the formation of chromatin rings, nucleolar size *per se* is not sufficient to explain the 3D genome configuration and other factors are involved. We now describe these new data in the revised manuscript, lines 232-240.

4. An important analysis which has not been performed is the quantification of rRNA levels in fed vs fasted cells and the other conditions and between the different cell types. All these analyses would have helped in dissecting the mechanisms related to nucleoli.

We agree with the reviewer that this independent measurement of RNA Pol I activity is important. Thus, we designed fluorescent probes within the ITS1 (Internal Transcribed Spacer region 1, *its-1* in worms) of rRNA, thus being able to monitor unprocessed pre-rRNA, which provides a good readout for rDNA transcription, and quantified pre-rRNA levels in muscle, hypoderm and intestine in:

1. fed and fasted animals
2. control and RNA Pol I degraded animals
3. control and mTOR knocked down animals (*let-363*/mTOR RNAi).

We found that:

1. as expected, pre-rRNAs decrease in all tissues when RNA Pol I is inhibited by RPOA-2 AID-mediated degradation (new Extended data Fig. 7c).
2. in agreement with an overall decrease in RNA Pol I transcription, we found that pre-rRNA levels are reduced in all tissues tested upon fasting and mTOR inhibition (new Extended Data Fig. 6a, b).

3. while overall changes in pre-rRNAs amounts and nucleolar size are generally concordant (e.g. they are both reduced in fasting, upon Pol I depletion and mTOR inhibition), the relative changes of the two do not always match. For example, during fasting pre-rRNA levels in muscle decrease similarly to intestine and hypoderm (new Extended Data Fig. 6a), however the size of the nucleolus is drastically reduced in intestine and only modestly affected in muscle (new Fig. 4d).

In conclusion, RNA Pol I activity is diminished in all conditions where rings are formed, in agreement with a role of RNA Pol I in this large-scale genome reorganization. However, different tissues regulate their nucleoli differently in response to changes in pre-rRNA amounts and changes in nucleolar volume correlate better than changes in pre-rRNA levels with the formation of chromatin rings. We discuss these conclusions in a new paragraph of the discussion section of the revised manuscript, lines 403-415.

5. Finally, which is the consequence of this chromatin arrangement? Is globally gene expression downregulated? And does it occur only in intestine cells?

To be able to perform tissue-specific, global gene expression analysis, cells from individual tissues have to be isolated and counted, in order to add a spike-in for absolute quantification. Towards this aim, we tried to optimize larval dissociation into a single-cell suspension while preserving cell health, to then perform Fluorescence Activated Cell Sorting of intestinal and other cell types in L1 larvae. Unfortunately, the sorted intestinal cells were not of sufficient quality to yield reproducible RNA-seq results (data not shown).

We therefore switched to an alternative approach: the hand dissection of intestines from adults, which works very robustly (Hill et al., 2022) but is not suitable to isolate other somatic tissues or for absolute RNA quantification, as the number of cells extracted cannot be precisely estimated. Using this approach, we analyzed differentially expressed genes in 12 hours fasted adults, which form chromatin rings (Fig. 1h, i), compared to fed, and in fed adults where chromatin rings are induced by AID-mediated RNA Pol I inhibition (new Extended Data Fig. 8c-e) at different time points of auxin exposure, compared to TIR1 only, as control.

As explained earlier in the response to this reviewer, this allowed us to uncouple the configuration of the 3D genome from the nutritional status of the animal (scheme in new Fig. 7b) and determine whether specific genes are consistently altered in the two conditions. These would be genes potentially responsive to the reorganization of the intestinal genome into two rings.

We found that 182 genes are upregulated and 76 are downregulated in both conditions (new Extended Data Fig. 9a). Remarkably, the majority of genes overexpressed in response to Pol I inhibition in fed animals were also upregulated in intestinal cells during fasting (new Fig. 7d, f). In contrast, most genes that are downregulated by Pol I depletion after 24 hours on auxin are not repressed during fasting (new Fig. 7e, f). Interestingly, the reverse analysis yielded similar results: genes that are upregulated during fasting (new Fig. 7a) tended to be upregulated also in Pol I depleted animals (new Fig. 7g), while genes with reduced expression during fasting were overall not downregulated in absence of Pol I activity (new Fig. 7g). **These results indicate that the reorganization of the 3D genome into chromatin rings correlates with the upregulation of a specific subset of genes.** Importantly, these do not include ribosomal, rRNA processing, translation and nucleolar genes, (new Fig. 7h and Extended Data Fig. 9b). Instead, genes that are upregulated upon ring formation are enriched in GO categories related to amino acid metabolism and stress response (Fig. 7i), regardless of animals being fed or fasted.

These results provide evidence that the chromatin ring configuration may itself favor the expression of specific genes during fasting.

Other major points

6. Lane 109: “we find that the signal is low at the edge of the nucleolus.” and Extended Data Fig. 1s. To better show the data on chromatin proximity to nucleoli, the x-axis should show the radius of nucleoli (i.e., nucleolus borders).

We apologize if the figure was unclear. In the previous version, the red dotted line marking the 0 indicates the border of the nucleolus. In the revised version, we added the nucleolar center as an additional reference point (now Extended Data Fig. 2a). As the nucleolus shrinks in fasted worms, the center is closer to the border compared to the fed state.

7. Fig. 2g. I think that gene expression levels of the euchromatin reporter with cherry-histone should be performed using RNA-FISH or other systems to detect mRNA and not by measurements of protein levels. Indeed, if fasting reduces rRNA synthesis, as mentioned by the authors, this should cause downregulation of ribosome production with a consequent reduction of protein synthesis. mRNA reporter measurements will also allow to directly determine whether loci contacting the nuclear periphery or nucleolus are active or repressed.

We agree with the reviewer that RNA FISH, combined to DNA FISH would be the ideal method to determine both the position and the transcriptional output of a gene. However, this technique relies on cell fixation, and as we showed in Extended Data Fig. 2d, fixation disrupts the ring structure. Because the preservation of chromatin rings is essential to be able to determine the expression level of a gene with respect to its positioning at the inner, outer or in between the rings, we can only address this question with live imaging.

Although we monitor a protein and not an RNA output, our live imaging approach is valid to quantify allele expression (and we only used this term in the text, and avoid referring to transcription), and allows us to conclude that both the inner (perinucleolar) and outer (nuclear periphery) rings constitute repressive compartments, for the following reasons:

1. our new data (new Fig. 2g), show that upon depletion of H3K9 methylation (a well-known repressive histone mark), the levels of mCherry are largely comparable across intestinal nuclei of fasted larvae, regardless of where the alleles are located with respect to the chromatin rings. This indicates that the differences observed in wild type animals do not stem from an altered protein synthesis but are due to specific repressive features of the chromatin rings, which are impaired in absence of H3K9 methylation.
2. while we cannot exclude that 12 hour of fasting affects protein synthesis, this would affect ALL intestinal cells within the fasted worms equally, regardless of where the two euchromatic alleles are positioned within the nucleus.

Thus, the differences in mCherry levels that we measured across the different intestinal cells are unlikely to stem from a reduced protein synthesis and are rather due to the subnuclear location of the alleles.

8. Lines 169-170. “This argues that nucleolar size is regulated very early in the response to refeeding, which ultimately leads to the dispersion of the chromatin rings”.

There is no data showing that nucleolar size “leads to the dispersion of the chromatin rings”. The experiment has only showed that nucleolar size decreased upon fasting and increase upon refeeding.

We agree with the reviewer and apologize for having written this confusing sentence. We have now changed the sentence in the main text to make it clearer, lines 251-255:

“Upon refeeding, the nucleoli enlarge. Whereas 2 hours are required for the nucleolus to regain the size of fed animals, refeeding for only 5 minutes was sufficient to detect an increase in nucleolar volume (Fig. 5c). This argues that nucleolar size is regulated very early in the response to refeeding and its increase might be part of the changes that contribute to the dispersion of the chromatin rings.”

9. Figure 3. The authors should measure whether chromatin rearrangements upon auxin-mediated depletion of Pol I occur in other cell types.

Besides intestine, we have now measured chromatin organization in hypoderm and muscle upon depletion of RNA Pol I. Our new results show that when RNA Pol I is inhibited, hypoderm, but not muscle, cells reorganize their genome into two rings (new Fig. 6e, i and new Extended Data Fig. 7d). This indicates that **the capability to reorganize the genome into chromatin rings is not unique to intestinal cells**. Yet, during fasting only intestinal cells form chromatin rings, likely because in hypoderm the inhibition of RNA Pol I is not strong enough.

We present these important new data in the section of the revised manuscript entitled “Inhibition of RNA Pol I transcription is sufficient to induce a fasting-like chromatin architecture in the intestine and hypoderm of fed animals”, more precisely in lines 322-343.

10. Moreover, they should compare nucleolar size and rRNA transcription between different cell types in fed and fasting conditions. This might help to clarify whether and how chromatin in intestine cells is different or respond differently to nucleolus signal relative to other cell types and find out the differences.

We thank the reviewer for this suggestion. As explained in the answer to point 4 of this reviewer, as a readout for rRNA transcription, we quantified pre-rRNA levels by FISH in intestine, hypoderm and muscle during fasting, mTOR inhibition and RNA Pol I depletion. In parallel, we measured nucleolar volume.

We found that:

1. changes in pre-rRNAs amounts and nucleolar size are overall concordant across tissues. For example, they are both reduced in fasting, upon Pol I depletion and mTOR inhibition, supporting the notion that RNA Pol I transcription decreases in these conditions.
2. however, the relative changes of in pre-rRNAs amounts and nucleolar size in the different tissues vary. For example, during fasting pre-rRNA levels in muscle decrease similarly to intestine and hypoderm (new Extended Data Fig. 6a), but the size of the nucleolus is only weakly reduced in muscle compared to intestine, where it is drastically reduced (new Fig. 4d).

From these results, we conclude that while RNA Pol I activity is diminished in all conditions where rings are formed, in agreement with a key role of RNA Pol I, nucleoli from different tissues respond differently to changes in RNA Pol I transcription. Importantly, our data suggest that changes in nucleolar volume correlate better than pre-rRNA levels with the formation of chromatin rings. We present these new data in the revised manuscript:

Lines :224-229

Lines: 283-287

Lines: 330-343

and extensively discuss their interpretation in the discussion, lines 403-415.

11. Lanes 210-211. “we inhibited RNA pol I transcription by the addition of Actinomycin D”. Depending on the concentration, ActD can also inhibit Pol II and Pol III. The authors should clarify whether the treatment they used is specific to Pol I.

We supplemented plates with a concentration of ActD (100 µg/ml), which is not considered to be Pol I specific (Hori and Watanabe 2008; Perry and Kelley 1970). While it is unfortunately not possible to determine the exact concentration of ActD absorbed by intestinal cells, it is unlikely that our treatment is specific to RNA Pol I. To clarify this, we changed the text accordingly in lines 311-313:

“exposure to Actinomycin D, a broad transcriptional inhibitor with a higher affinity for blocking rRNA synthesis by RNA Pol I^{60, 61}, induced a fasting-like reorganization of chromatin in intestine of fed animals (Extended Data Fig. 6i-k).”

12. Figure 5. The authors need to measure rRNA transcription and nucleolar size upon mTOR inhibition and its constitutive activation in intestine and other cells. Moreover, they need to show whether other cell types have similar chromatin rearrangements under these conditions.

What about nucleolar volume in other tissues of ragaGF fasted?

As outlined in the response to previous points, we have now quantified pre-rRNA levels, nucleolar volume and chromatin organization upon mTOR inhibition, by RNAi-mediated knock down of *let-363*/mTOR.

1. Upon mTOR inhibition, we observed partial chromatin rings in intestine (new Fig. 3a-c and Extended Data Fig. 3h), phenocopying what we previously showed for depletion of DAF-15/RAPTOR (now in Extended Data Fig. 3l-o). However, the same reorganization did not occur in hypoderm and muscle (new Fig. 3a-c and Extended Data Fig. 3h).
2. We quantified pre-rRNA levels and found that they are decreased in all three tissues tested, albeit the effect is less pronounced in muscle (Extended Data Fig. 6b).
3. By measuring the volume of the nucleolus in fed animals where mTOR was inhibited, we observed that both intestine and hypoderm strongly reduce nucleolar size, while in muscle the effect is minimal (new Fig. 4e). Similarly to fasting, the biggest relative reduction in nucleolar volume was detected in intestine.

These results suggest that the three tissues tested respond differently to mTOR inhibition. Muscle cells are very weakly affected as their decrease in pre-rRNAs amounts is modest, the size of the nucleolus is almost unaffected and the chromatin organization is like that of control worms. In contrast, both

intestine and hypoderm respond with a more pronounced inhibition of rRNA transcription and reduction of nucleolar size. However, partial rings are detected only in intestine, the tissue where the relative nucleolar volume reduction is the strongest.

We now discuss the effect of mTOR inhibition on chromatin organization, nucleolar volume and pre-rRNA amounts in intestine, hypoderm and muscle in the following paragraphs of the revised manuscript, respectively:

Lines 186-197

Lines 224-229

Lines 283-287

The inhibition of mTOR in fed animals triggers a partial chromatin rings architecture only in intestinal cells (new Fig. 3a-c). Additionally, fasting induces chromatin rings only in intestinal cells (Fig. 1a-c). While expression of RAGA-1^{GF} counteracts rings formation in intestine (Fig. 3d-f and Extended Data Fig. 3p), together with the presence of larger nucleoli (Fig. 4f), investigating the effects of mTOR constitutive activation by RAGA-1^{GF} in the other tissues would not be very informative with respect to understanding the formation of chromatin rings. Hence, chromatin organization and nucleolar volume in muscle and hypoderm were not analyzed in *raga-1^{GF}* mutants.

13. Lanes 255-256. “expressing RAGA-1GF does not antagonize the formation of the concentric rings upon RNA Pol I inhibition (Fig. 5j-l)...”. The authors should show the expression levels of RAGA-1GF upon Pol depletion since the decrease of ribosome production might have an impact of protein levels.

To address this point, we now measured the effects of RNA Pol I inhibition on the expression of the RAGA-1^{GF} protein by taking advantage of animals expressing a GFP-RAGA-1^{GF} fusion. Briefly, we exposed GFP-RAGA-1^{GF}-expressing worms to 100 µg/ml ActD for 6 hours and quantified the GFP-RAGA-1^{GF} intensity by live imaging. As shown in new Extended Data Fig. 6l, we found that the levels of the RAGA-1^{GF} protein are stable within this time frame of transcription inhibition. This result indicates that the inability of RAGA-1^{GF} to counteract the formation of chromatin rings upon 5 hours of auxin exposure in RNA Pol I-AID worms (Fig. 6d and Extended Data Fig. 6m-n.) does not stem from a reduction in the protein levels of RAGA-1^{GF}, but rather argues that RNA Pol I acts downstream of mTOR, (consistent with other studies (Mayer et al, 2004; James et al., 2004). These new control data are described in the revised manuscript in lines 318-321.

Minor points

The authors have often referred to quite old literature about regulation of rRNA transcription upon nutrient deprivation. It is not wrong, but maybe they can update it a bit.

We thank the reviewer for this suggestion. We have now added 3 more references on the regulation of rRNA transcription upon nutrient deprivation: 1 recent book chapter (Tanaka and Tsuneoka, 2018) and 2 original research articles (Marguerat et al. 2012 and Tanaka et al 2010).

Lanes 151-152 “these results indicate that the perinuclear and perinucleolar chromatin rings are

repressive compartments.”. Maybe here the authors should comment that their data are in line with previous works showing that both nuclear periphery and perinucleolar compartment are repressive.

We agree with the reviewer and added these new sentences to the main text:

- Lines 105-107: “The 3D organization of chromatin reflects the functional compartmentalization of the genome¹⁰, with the nuclear and nucleolar periphery being sites where heterochromatin, the silenced portion of the genome, accumulates¹¹.”

And about H3K9 methylation in 3D genome organization:

- Lines 141-142: “in agreement with the role of this mark in perinuclear positioning of genes in worms^{12, 19} and mammals²⁰⁻²³”.
- Lines 420-422: “This is not unexpected given that rings form at the nuclear and nucleolar periphery, compartments known for being enriched in silenced chromatin domains marked also by H3K9me¹¹.”

Reviewer #3:

Remarks to the Author:

Reviewer assessment NCB-LE51990

In the manuscript “An mTOR/RNA Pol I axis shapes chromatin architecture in response to fasting” by Al-Refaie et al., the authors show that fasting induces dynamic chromatin reorganization in the intestine of *C. elegans*. The authors describe a fasting-induced double-spherical structure where chromatin redistributes to the nuclear lamina and nucleolar periphery. The genome reorganization is recapitulated by knockdown of RNA Pol I, but not Pol II nor Pol III. The authors propose an mTOR-Pol I axis for large-scale genome reorganization in response to nutrient deprivation.

The main significant finding from this study is the description of a stereotyped chromatin reorganization event in response to fasting. This is a visually captivating reorganization event that has gone unnoticed in the *C. elegans* field, and adds an interesting layer of regulation for the cellular response to fasting. The finding will likely spark future interest in large-scale chromatin reorganization events across model organisms and contexts.

We thank the reviewer for very positive evaluation of our work.

Excitement of this finding would be stronger if the phenomenon could be linked to a physiological phenotype (e.g. fasting/refeeding adaptation), downstream transcriptional changes resulting from the chromatin reorganization event can be identified, and/or if the same type of genome reorganization could be seen in other model systems.

The findings regarding the upstream regulators, especially Pol I is intriguing as it links the chromatin reorganization event to rDNA transcription. The link with the mTOR pathway supports previous publications linking mTOR-Pol I-rDNA transcription (Tsang et al, 2003 EMBO; Mayer et al, 2004 Gene Dev). The study complements studies reporting links between mTOR/nutrient restriction with chromatin reorganization (Lu et al, 2021 eLife) and nucleolar reorganization (Tiku et al, 2017 Nat. Comm).

The data is of high quality and presented well. Some potential points for improvement are outlined below.

Major comments:

1. Is the double-sphere reorganization event really a tissue-specific fasting event?

- A later time point should be checked to see whether the intestine is the first responder, or the only responder. It could be possible that other cell types would reorganize after more prolonged periods in starvation. L1 arrested animals can survive for longer than a month, so a time point of 7 days could be a second time point to check whether other cell types also display a similar or possibly different type of genome reorganization. Alternatively/additionally, if nucleolar size is quantified in other tissues, it would indicate whether other cell-types are already responding to fasting but nonetheless is not showing the double-ring structure.

We thank the reviewer for these insightful suggestions. To address this point, we quantified nucleolar size and pre-rRNA levels using FISH, as a readout for rDNA transcription, in different tissues:

1. Intestine
2. Hypoderm
3. Muscle

We found that:

1. all tissues analyzed respond to 12 hours fasting. In fact, the levels of pre-rRNA are reduced (new Extended Data Fig. 6a) and the nucleolus shrinks (new Fig. 4d).
2. despite a similar relative reduction in pre-rRNAs in the three tissues, the relative change in nucleolar volume is strongest in the intestine, the only tissue forming chromatin rings (new Fig. 4d).

These data suggest that different tissues regulate their nucleoli differently in response to similar changes in pre-rRNA amounts and that the strongest decrease in nucleolar volume correlates with formation of chromatin rings. We now discuss these points in the results section of the revised paper.

Lines: 222-229

Lines: 285-286

and in the discussion, lines 403-415

- Other cell-types besides the intestine should be shown in the Pol I knockdown, and whether this intervention results in a double-sphere structure.

We thank the reviewer for suggesting this important experiment, which proved to be very informative.

Besides intestine, we have now measured chromatin organization in hypoderm and muscle upon depletion of RNA Pol I. Our new results show that when RNA Pol I is inhibited, hypoderm cells, but not muscle, reorganize their genome into two rings (new Fig. 6e, f, i and Extended Data Fig. 7d).

This indicates that the capability to reorganize the genome into chromatin rings is not unique to intestinal cells. Yet, during 12 hours of fasting only intestinal cells form chromatin rings, likely because in hypoderm the inhibition of RNA Pol I is not strong enough.

These new data are part of the revised manuscript and are discussed in the section entitled “Inhibition of RNA Pol I transcription is sufficient to induce a fasting-like chromatin architecture in the intestine and hypoderm of fed animals”, in lines 322-343.

- Testing the *eat-2* mutant for its effect on the double-ring phenotype would indicate whether it is fasting that is required, or reduced feeding would also result in this chromatin reorganization.

We thank the reviewer for this suggestion and tested *eat-2* mutants. We did not observe chromatin rings in intestinal cells, as shown below:

We conclude that reduced feeding is not sufficient to induce a chromatin rings configuration in the intestine, at least at the time point monitored (40 hours after hatching).

2. Is the outer nucleolar area repressive?

- The statement that “The fasting-induced chromatin rings are repressive compartments,” (section title line 113) seems strong based on the experiments provided. Although it is reasonable to hypothesize that the nucleolar periphery is in general repressive, a similar reporter construct using a different promoter (vs. *pha-4*), different gene/protein (vs. *HIS-24*), different genomic location (current location unknown) could show different results. The choice of *pha-4* promoter makes it difficult to generalize, as *pha-4* is required for dietary restriction-mediated longevity (Panowski et al., Nature 2007). The regulation and transcriptional output of the *pha-4* promoter may therefore be different from a general housekeeping gene such as *actin*, or a gene that may be strongly induced during fasting (e.g. *sod-3*). An orthogonal experiment of looking at repressive and active histone marks at the nucleolar periphery would help, although it seems experimentally difficult due to the labile nature of the double-ring structure. Something that could be possible would be to check in mutants of repressive chromatin *H3K27me3*, *H3K9me3*, whether knockdown/knockout of repressive chromatin marks would affect the double-ring reorganization event and/or the expression level change of *HIS-24:mCherry* that is correlated with the localization of the euchromatin reporter.

We agree with the reviewer that the use of just one reporter has limitations. However, the generation, validation and quantification of 2, 3 or 4 new reporter worm lines would require several months of work, and would still provide just a partial understanding of the repressive (or not) nature of these chromatin structures. Thus, we have decided to follow the orthogonal approach suggested by the reviewer and further characterized the repressive features of the rings using knockdown/knockouts of repressive chromatin marks.

- In particular we focused on:

- the depletion of H3K9me1, me2 and me3 (referred to as H3K9me) by deleting the histone methyltransferase MET-2/SETDB1 together with a deletion/knock down of SET-25/SUV39H1 (Towbin et al., 2012)
- the loss of H3K27me3 in *mes-2/EZH2* mutants (Holdeman and Strome 1998) or upon *mes-6/EED* RNAi (Korf et al., 1998)

We found that both perturbations did not alter the formation of the chromatin rings during fasting (new Fig. 2c and Extended Data Fig. 2b, c), showing that H3K9me and H3K27me3 are dispensable for the reorganization of the genome in response to lack of nutrients. We now discuss these findings in the revised main text, lines 107-112.

Next, we quantified the effects of depleting these histone marks for positioning and expression of the euchromatic reporter and found that:

1. impairing H3K9me drastically decreased the frequency at which the active reporter interacts with the nuclear periphery both in fed and fasted conditions (new Fig. 2e), in agreement with the fundamental role of this mark in tethering chromatin to the nuclear lamina in worms (Towbin et al., 2012; Gonzalez-Sandoval et al., 2015) and mammals (Bian et al., 2013, Kind et al., 2013, Harr et al., 2015, Poleshko et al., 2019). In contrast, the frequency of localization at the inner ring remained unchanged.
2. when H3K9me is reduced, allele position-dependent differences in mCherry expression are almost completely abolished, except in nuclei where both alleles are at the inner ring, which continue to show reduced expression (new Fig. 2g).
3. animals carrying low levels of H3K27me3 did not alter the repression of the reporter by chromatin rings (new Fig. 2g) but revealed a role for positioning the active allele to the outer ring during fasting (new Fig. 2e).

In sum, these new results suggest that both H3K9me and H3K27me3 contribute to positioning the euchromatic allele at the outer ring during fasting but only H3K9me mediates repression at this location. In contrast, positioning and repression of the euchromatic reporter at the inner ring remain largely unaffected, suggesting that inner and outer ring are regulated differently. We now discuss these new findings in lines 139-166.

- Alternatively/additionally, co-localization of the ring structure could be checked with fluorescent reporter strains expressing enzymes or proteins that localize to heterochromatin/repressive chromatin (e.g. HPL, MES proteins) vs. euchromatic regions (potentially MRG-1 identified previously by the author(s)).

To address this point, we took advantage of worms expressing fluorescently tagged MRG-1 (H3K36me2/3 reader, homolog of mammalian MRG15), HPL-1 and HPL-2 (homologs of mammalian heterochromatin protein 1 (HP1), readers of H3K9me) and monitor their subnuclear distribution in fed and fasted intestinal nuclei.

During fasting, MRG-1 reorganizes to generate ring-like structures, while HPL-1 and HPL-2 do not; they remain diffuse in the nucleoplasm (see left for representative images).

This might suggest that H3K9me is not enriched at chromatin rings while H3K36me is. However, the nuclear distribution of chromatin readers is not only influenced by the abundance and location of their respective histone targets, but also by their relative affinity for histone modification and dynamics of binding. Thus, also considering our new results showing that H3K9me is required for the repression of the euchromatic reporter when this is positioned at the outer ring (new Fig. 2g), we cannot draw conclusions on the enrichment or not of H3K9me at chromatin rings. Nonetheless, it is interesting to note that chromatin rings are not devoid of MRG-1, which typically labels H3K36me2/3 marked euchromatin (Cabianca et al., 2019).

3. Other

- The authors nicely demonstrate that refeeding rapidly causes a reorganization of the heterochromatin and expansion of nucleolar size. What about the kinetics of fasting? Does it also cause a rapid reorganization of heterochromatin and nucleolus, or is it a slower process?

We thank the reviewer for suggesting this experiment. We now monitored both nucleolar volume and chromatin organization at earlier time points during fasting to monitor the kinetics of the reorganization. We found that:

1. after 2 hours of fasting the radial distribution of chromatin in intestinal cells still shows a single peak, yet partial chromatin rings can be observed after 6 hours. After 10 hours in absence of food, chromatin rings are fully formed (new Fig. 4a, b and new Extended Data Fig. 4a).
2. at the same time points, we quantified nucleolar volume and interestingly observed a significant reduction in size already at 2 hours of fasting, with the size decreasing further at later time points (new Fig. 4c).

These experiments show that the dynamics of ring formation is much slower compared to their dispersal upon refeeding (which occurs in 30 minutes, Fig. 5a, b). Interestingly, a reduction in nucleolar size precedes the appearance of the chromatin rings.

These new results are discussed in a new paragraph of the revised manuscript entitled “Remodeling of the nucleolus correlates with the reorganization of chromatin into two rings”, lines 209-219.

Minor comments:

1. In Fig. 3, the authors show that nucleolar size changes precede the concentric spheres to be dispersed. They argue that the nucleolar size change “leads to the dispersion of the chromatin rings (lines 168-170). The requirement for nucleolar shrinkage in chromatin reorganization could be tested in a *ncl-1* mutant, which retains a large nucleolus in response to dietary restriction.

We apologize for this confusing sentence. What we meant to say is that the refeeding ultimately leads to the dispersion of the chromatin rings and not that this is caused by the nucleolar size change. Nonetheless, our new data suggest that a regulation of nucleolar size is critical for 3D genome architecture regulation in response to nutrients:

1. during fasting the nucleolus shrinks before the rings are formed (new Fig. 4a, c), while during refeeding, the nucleolus enlarges before chromatin rings are fully dispersed (Fig. 5a, c).
2. by analyzing the volume of the nucleolus in three different tissues (intestine, muscle and hypoderm), in three different conditions (fasting, mTOR inhibition and RNA Pol I depletion), we find that a strong relative reduction in nucleolar size compared to the corresponding control is the best predictor for inducing rings formation. This is true not only in intestine (new Fig. 4d-e and new Fig. 6f, g), but also in hypoderm, where upon Pol I inhibition we now find that rings are formed (new Fig. 6e, i), and the size of the nucleolus is reduced more compared to what we measured during fasting in the same tissue, and mirrors the relative change in size observed in intestinal cells (new Fig. 6g, h).

The reviewer is right that the role of the nucleolus could be tested in a *ncl-1* mutant. However, we already presented data showing that the constitutive activation of mTOR (in *raga-1^{GF}* mutants) increases nucleolar size in intestine (Fig. 4f), and that RAGA-1^{GF} expression antagonizes the formation of chromatin rings in the intestine cells upon fasting (Fig. 3d-f). Thus, to determine the role of nucleolar size *per se* in regulating 3D genome reorganization in absence of nutrients, we decided to:

1. select the 50 wild type fasted intestinal cells possessing the largest nucleolus (highlighted in red in the boxplot below) and measure their radial chromatin distribution. We found that they form chromatin rings (new Fig. 4h).

2. select 50 *raga-1^{GF}* fasted intestinal cells with the same nucleolar size as in wild type (in red in the boxplot on the left and in new Fig. 4g) and monitor 3D chromatin organization. We found that these *raga-1^{GF}* intestinal cells do not form rings.

Altogether, this shows that while during fasting the volume of the nucleolus in intestine is drastically reduced (new Fig. 4d) and this correlates with the formation of chromatin rings, nucleolar size *per se* is not sufficient to explain the 3D genome configuration, and other factors are involved.

We extensively discuss these new results, together with the others mentioned above on nucleolar volume changes in all the tissues and conditions tested, in a new paragraph of the revised manuscript entitled "Remodeling of the nucleolus correlates with the reorganization of chromatin into two rings", lines 222-240.

2. *raga-1* loss of function(ok386 or ok701) or *let-363* knockdown would add further support for the mTOR pathway, while using alleles/conditions that have been characterized in other contexts. Relatedly, *daf-15/raptor* AID causes only a partial reorganization of the heterochromatin. Perhaps other pathways contribute? In particular, the authors should consider testing MAPK signaling.

We have measured chromatin distribution in all conditions suggested by the reviewer:

- *raga-1* loss of function (*ok701*)
- *let-363* RNAi, compared to control
- downregulation of the MAPK signaling (*mpk-1* RNAi), compared to control

We found that:

1. Mutating *raga-1* is not sufficient to induce the formation of chromatin rings in the intestine of fed animals, at least at the time point monitored (30 hours after hatching), as the radial chromatin distribution shows a single peak (see below).

2. Upon *let-363*/mTOR knockdown, we observed partial chromatin rings in intestine (new Fig. 3a-c and Extended Data Fig. 3h), phenocopying what we previously showed for depletion of DAF-15/RAPTOR (Extended Data Fig. 3l-o).
3. In worms MPK-1 is the ortholog of mammalian ERK1/2, components of the 'classical' Mitogen-Activated Protein Kinase (MAPK) pathway. MPK-1 is activated in absence of nutrients (You et al., 2006). Thus, we knocked down *mpk-1* and tested whether this could affect the chromatin reorganization in intestine during starvation. As shown in new Extended Data Fig. 3d-f we found no alteration.

In summary, mutation in *raga-1* alone does not induce rings or partial rings, likely because of redundancy from other mTOR activators (Blackwell 2019). In fact, while DAF-15/RAPTOR and LET-363/mTOR depleted larvae completely arrest development (new Extended Data Fig. 3g, k), *raga-1* mutants reach adulthood. Remarkably, the results obtained by knocking down *let-363*/mTOR further confirmed the role of mTOR in mediating the 3D genome organization in response to nutrients.

The effect of inhibiting mTOR is partial compared both to fasting (Fig. 1a-e) and the direct inhibition of RNA Pol I (Fig. 6a-c and Extended Data Fig. 6c). Thus, we agree with the reviewer that other pathways acting on RNA Pol I activity in response to nutrients are likely to be implicated. With our work we could exclude AMPK (Extended Data Fig. 3a-c) and MAPK, at least via MPK-1 (new Extended Data Fig. 3d-f).

We added to the revised manuscript the data on LET-363/mTOR and MAPK, which are discussed in the paragraph entitled “mTORC1 signaling is necessary and sufficient to regulate 3D genome architecture in response to nutrients”, lines 176-189 and 195-197.

3. DIC insets would be helpful in visualizing the images (Fig 1 and others). For example, we would be able

to see if fasting-induced morphological changes are occurring such as nucleolar shrinkage, and be able to compare among different tissues (whether nucleolar shrinkage and the double-ring occur together or not in the different tissues).

While we did not acquire our images with a DIC setting and therefore cannot use it to do the analysis suggested by the reviewer, our new quantification of nucleolar volume in other cell types using live imaging of FIB-1/Fibrillarin fused to mCherry nicely serves the same scope.

We have now systematically quantified nucleolar volume in intestine, hypoderm and muscle in the following conditions:

- Fed and fasting
- mTOR inhibition by RNAi for *let-363* (and control)
- Pol I degradation (and control)

As explained above, we found that:

1. the nucleolus diminishes its size during fasting also in hypoderm and muscle, but less than in intestinal cells (new Fig. 4d) and the same occurs upon mTOR inhibition (new Fig. 4e). Importantly, rings are formed only in intestinal cells in both conditions (rings are partial for mTOR inhibition).
2. upon Pol I inhibition in hypoderm and muscle, the size of the nucleolus is reduced more compared to what we measured during fasting in the same tissue (new Fig. 6g, h). Yet, only in hypoderm the relative change in nucleolar size now matches what observed in intestinal cells both during fasting and upon Pol I inhibition, while in muscle the effect remains weaker (new Fig. 6g, h). Remarkably, chromatin rings are formed in hypoderm but not in muscle (new Fig. 6e, i)

In sum, these new data show that nucleoli are not affected exclusively in intestine. Muscle and hypoderm also respond to both fasting and mTOR signaling with a reduction in nucleolar size, albeit the effect is consistently weaker compared to what we measured in intestine under the same perturbations. Interestingly, the acute inhibition of RNA Pol I causes nucleoli in hypoderm to shrink, in relative terms, like intestinal cells, and this correlates with the formation of chromatin rings in hypoderm, suggesting that a strong relative reduction in nucleolar volume correlates with rings formation across tissues.

As outlined in the response to previous points, we now extensively discuss changes in the nucleolus in the different conditions in the results section of the main text:

Lines: 209-232

lines: 251-255

lines: 322-343

and in the discussion, lines 403-415.

4. Is pol I expression or localization changed by fasting/refeeding? Also it would be helpful to explain RNA pol I somewhere in the discussion as a polymerase that mainly acts on rDNA, so the significance can be conveyed to the reader.

To determine the localization and quantify the expression of RNA Pol I, we monitored RPOA-2, a core subunit of RNA Pol I, fused to GFP, in fed, fasted and refed intestinal cells. We found that RPOA-2 remains

localized inside the nucleolus (labeled by FIB-1/Fibrillarin-mCherry) and its expression is moderately diminished during fasting (see left). While it is possible that this contributes to reducing RNA Pol I activity during fasting, other mechanisms are likely to be involved.

As suggested by the reviewer, to better convey the significance of our findings

on RNA Pol I to the reader, in the discussion we now wrote: “Based on our results, we propose that the 3D genome architecture of intestinal cells is modulated by mTOR signaling in response to nutrients acting through the regulation of RNA Pol I, which transcribes the rDNA in the nucleolus”. Lines 395-397.

5. Worm strain information of the GW429 strain is listed to be 256x copies (Table S1), while in figure 2 it states it is reduced copy number. This should be clarified.

We apologize for the confusion. The 256x refers to the copies of the LacO repeats *per plasmid*, not the array of the integrated *pha-4-mCherry* plasmid. Briefly, the GW429 strain (Meister et al., 2010) was generated by ballistic transformation, which generates a rare integration event that results in the stable propagation of the exogenous DNA, with typically 1 to 50 copies of the plasmid (these are called small arrays, see answer to point below). In Meister et al., 2010, the copy number of the *pha-4::mCherry* plasmid in the GW429 strain was quantified to be around 30. To enable its visualization using GFP-LacI, the *pha-4::mCherry* allele was cointegrated with a plasmid carrying 256X of LacO repeats.

To help the readers with this information, we have now adjusted the text in the methods, lines 621-624: “The GW429 strain¹⁸ was created by ballistic transformation, generating a rare integration event of about 30 copies of the *pha-4::mCherry* plasmid. The 256x copies in the genotype refers to the copies of LacO repeats carried by the cointegrated plasmid to enable visualization of the allele by GFP-LacI.”

6. It would be helpful if the physiological difference in copy number could be explained regarding ~30 copies vs ~300 copies. The general threshold for heterochromatinization of repetitive DNA should be discussed/explained.

In *C. elegans*, integrated transgene arrays acquire different chromatin marks and subnuclear locations, based on their size. Especially in embryos, large arrays composed of 300-500 plasmid copies are “heterochromatinized” by the deposition of H3K9me3 and H3K27me3, are peripherally located and silenced, while the same sequence, if present in less than 50 copies, is not (Towbin et al., 2011). As cells differentiate, arrays carrying tissue-specific promoters shift inwards from the periphery exclusively in the tissue in which they are active regardless of their copy number (Meister et al., 2010).

To examine how chromatin rings affect the spatial positioning of genes, we used these reporters (one heterochromatic - Extended Data Fig. 2f-h - and one euchromatic - Fig. 2d-g). The copy numbers are indicated (~300 copies for the heterochromatic and ~30 for the euchromatic allele). The relevant difference between the two is not the copy number, but the fact that one is silent in intestine and the other one is active, thanks to the presence of an intestine-specific *pha-4* promoter.

To further clarify this, we now provide information on copy number, position and “heterochromatinization” of the arrays, citing all the relevant literature in the methods, under “Information on the hetero- and euchromatic reporters used in Fig. 2 and Extended Data Fig. 2”, lines 610-624. Additionally, we now mention in the revised main text that: “we chose to monitor a transcriptionally repressed, high-copy number transgene (heterochromatic) and a low-copy number reporter, actively transcribed in intestine (euchromatic), both integrated at a single site in the worm genome, using a LacO/LacI-GFP-based visualization strategy in living cells.” Lines 117-120.

7. It would assist the reader if the observation that the heterochromatin reporter is unchanged from fasting is more carefully explained. A casual reader may expect to see a heterochromatin reporter at the nucleolar periphery, if as is proposed, the nucleolar periphery is repressive. However, the heterochromatin reporter used in this paper is not a reporter that labels all heterochromatin, but a couple of loci that are expected to be heterochromatic. A more careful explanation of the reporter and results could help in avoiding misinterpretations.

The reviewer is correct. The heterochromatic reporter in Extended Data Fig. 2f-h does not label all heterochromatin, but represents a well-characterized, integrated reporter that bears H3K9 methylation and represses transcription (Towbin et al., 2012; Gonzalez-Sandoval et al., 2015). Note that we refrain from drawing conclusions about the general positioning of heterochromatin, and we now state clearly in the text that this reporter does not represent all heterochromatin, but rather a H3K9me marked, perinuclear heterochromatin array, in lines 127-128:

“As previously reported¹⁵, this repressed, repetitive and H3K9me-marked heterochromatic reporter is strongly enriched at the nuclear periphery...”

Additionally, as written above, we have now carefully explained the reporters used in the study in the methods section.

Typically, heterochromatin accumulates at the nuclear periphery and around the nucleolus (Solovei et al., 2016). However, the frequency with which specific heterochromatic regions localize at the two compartments is variable (Bersaglieri et al., 2022, Kind et al., 2013) and the molecular mechanism is not clear. The reporter we use localizes about 80% at the nuclear periphery in fed intestinal cells (Cabianca et al., 2019 and Extended Data Fig. 2h), and this does not change during fasting (Extended Data Fig. 2h). We did not check whether the remaining 20% are near the nucleolus, because this reporter does not respond to fasting.

8. The same goes for the euchromatic reporter. The euchromatic reporter is from previous reports expected to be euchromatic in the fed condition. A reader may think that if the euchromatic reporter is moving to the nucleolar periphery, it may indicate that it is active there, whereas it is potentially heterochromatic/repressive at the nucleolar periphery.

We apologize if we failed to explain these data more clearly in the text. Indeed, the euchromatic reporter is actively transcribed and internally positioned in fed intestinal cells (Meister et al., 2010 and Fig. 2e). We found that during fasting, this reporter can localize both at the outer (peripheral) and inner ring (perinucleolar) (Fig. 2e). The goal of our experiment was to see whether positioning at these novel chromatin structures would affect the reporter's expression. We found that the expression of this allele diminishes when it is found either at the outer or inner ring (Fig. 2g), leading us to conclude that chromatin rings function as repressive compartments.

Our data do not clarify whether the allele itself gains heterochromatic marks or is rather repressed because of the proximity to repressive environments (the chromatin rings). Nonetheless, considering that *pha-4* remains active in absence of nutrients (Panowski et al., Nature 2007; Chen and Riddle 2008) we propose that the reporter remains "euchromatic" in the sense that it is active even when positioned at the chromatin rings but that its expression is reduced compared to when it is located between the rings.

To facilitate the understanding of this, we have now added this sentence to the main text, lines 165-166: "particularly for the outer ring, H3K9 methylation, but not H3K27me3, contributes to reducing the expression of the tested euchromatic reporter allele."

9. Statistical tests could not be found for some figures (e.g. Ext. Fig 1 C, D). The profiles look different from fed to fasted in hypodermis and muscle, but whether these are significantly different should be tested/shown.

In each chromatin profile line plot, the shaded area represents the 95% confidence interval of the mean, which allows to visually determine the variability of the data and differences between profiles. In the revised version, we have now additionally performed permutation tests to estimate the significance of the difference between the mean at each distance percentage 5% bin when comparing profiles both having a single peak.

To calculate a final p-value we adjusted the multiple p-values for multiple tests using Bonferroni correction and combined the adjusted p-values at three different regions of the plot (0%-30%, 35%-65% and 70%-100%) using Pearson's combined probability test.

We added all these p-values to the Supplementary Table 2 (statistics table) and referred to them in the Figure legends.

The profiles indicated by the reviewer are fed and fasted hypoderm and muscle (now Fig. 1c). By performing the statistical analysis explained above, we find that the change in 3D genome architecture in fasted compared to fed hypoderm is significant only in one of the three regions of the chromatin profile plot. In contrast, the 3D genome profile in fasted muscle is different in all three regions compared to fed (see results below, which were taken from Supplementary Table 2):

		p value over the same tissue fed region 1 (0%-30%)	p value over the same tissue fed region 2 (35%-65%)	p value over the same tissue fed region 3 (70%-100%)	n nuclei	N replica
Figure	nutritional status, tissue					
1c	fed, hypoderm	NA	NA	NA	70	3
1c	fasted, hypoderm	2.63E-13	1	1	70	3
1c	fed, muscle	NA	NA	NA	70	3
1c	fasted, muscle	2.63E-13	0.01313	0.002125	70	3

When two profiles with a single peak are significantly different in at least 1 of the 3 regions, like for example in the case of fasted hypoderm and muscle compared to fed, in the text we wrote “does not form rings” or sentences alike, and avoided writing that the profiles are not different, or alike.

10. I am not sure if “concentric rings” is the right term to use for this phenomenon, as this view is based on a 2D slice. In reality, they are concentric spheres.

The reviewer is correct, in the 3D space rings correspond to concentric spheres.

We now mention this in the main text, lines 54-57:

“Live confocal imaging of fed and fasted worms showed that histones underwent a drastic spatial reorganization forming two “concentric rings”, corresponding to concentric spheres in the 3D space, in all intestinal cell nuclei during fasting (Fig. 1a).”

However, because throughout the paper we show 2D images of single focal planes, we think that calling them concentric rings facilitates understanding.

11. The abstract should mention that the reorganization event is limited to the intestine.

We agree and changed “tissue-specific” to “intestine-specific”.

12. I would suggest that the title would emphasize the chromatin architecture first, as this is the main point of emphasis of the paper. For example something like “Fasting-induced chromatin reorganization via mTOR/RNA Pol I axis”.

We thank the reviewer for this suggestion, which we fully agree with. In fact, we have now changed the title to “Fasting shapes chromatin architecture via an mTOR/RNA Pol I axis”.

13. In terms of the flow of the paper, I felt that the Pol I–double ring structure figure (Fig 4) and mTOR–double ring structure figure (Fig 5) might flow better if it came right after Fig 2, and preceded the refeeding figure (Fig 3). This is just a suggestion based on personal preference.

In this revised version of the paper, we have moved the figure about mTOR (now Fig. 3) right after Fig. 2, as suggested. However, we left the refeeding data (now Fig. 5) before the figure showing that depleting RNA Pol I is sufficient to induce rings formation in fed animals (now Fig. 6), as we think that this flow helps the reader understand the rationale for the RNA-seq experiments where we compare fasted to fed, Pol I depleted intestinal cells (new Fig. 7).

Decision Letter, first revision:

Our ref: NCB-LE51990A

2nd July 2024

Dear Dr. Cabianca,

Thank you for submitting your revised manuscript "Fasting shapes chromatin architecture via an mTOR/RNA Pol I axis" (NCB-LE51990A). It has now been seen by the original Referees #2-3 and their comments are below. Rev#1 was not able to re-review, but Rev#2 kindly evaluated the responses to their points, which they found appropriate. The reviewers overall found that the paper has improved in revision, however, they both had remaining and overlapping issues consistent with their previous points that require further clarifications and test edits. Based on their feedback, we will be happy in principle to publish the manuscript in Nature Cell Biology, pending minor revisions to satisfy the referees' final requests and to comply with our editorial and formatting guidelines.

Please note that the current version of your manuscript is in a PDF format; could you please email us a copy of the file in an editable format (Microsoft Word or LaTeX) as we can not proceed with PDFs at this stage? Many thanks for your attention to this point.

Once we have the word file, we will be performing detailed checks on your paper and will send you a checklist detailing our editorial and formatting requirements in about 2 weeks. Please do not upload the final materials and make any revisions until you receive this additional information from us.

Thank you again for your interest in Nature Cell Biology. Please do not hesitate to contact me if you have any questions.

Sincerely,

Melina

Melina Casadio, PhD
Senior Editor, Nature Cell Biology
ORCID ID: <https://orcid.org/0000-0003-2389-2243>

Reviewer #2 (Remarks to the Author):

The authors have made significant improvements to the manuscript. I am pleased with the additional experiments addressing various nucleolar aspects, such as size and pre-rRNA measurements, and the analyses of other cell types. I support the publication of this work but have two points that should be addressed.

1. I am not satisfied with how the important point 7, concerning the measurements of gene expression levels of the euchromatin reporter using Cherry-Histone protein levels (Fig. 2g,h), has been addressed. These data should be removed from the manuscript. This revision will not affect the overall work, which describes a compelling 3D-genome reorganization during fasting and its regulation by Pol I and mTOR signaling

While I understand that direct measurements of mRNA levels for the histone-Cherry reporter are technically unfeasible, it is wrong to measure reporter expression at the protein level in cells where ribosome biogenesis and translational activity are known to be compromised. The use of H3K9me depletion to prove that the measurements are correct is unconvincing. Additionally, as noted by Reviewer 3 (point 2), the choice of the *pha-4* promoter, which controls the expression of *pha-4* - a gene essential for dietary restriction-mediated longevity - may not support the conclusion that active genes near nucleoli become transcriptionally repressed, as this gene might be regulated by other nutrient signaling pathways. The appropriate choice should have been a housekeeping gene. Therefore, there is no data demonstrating that the inner ring near nucleoli is a compartment that establishes gene repression. The presence of repressive chromatin around nucleoli in many cell types has been well documented. However, there is currently no evidence showing that an active gene becomes repressed due to its proximity to nucleoli. The results presented in this manuscript cannot demonstrate this for the reasons mentioned above. Additionally, the new RNA-seq data suggest an upregulation of gene expression upon fasting. Most genes upregulated by Pol I inhibition in fed animals were also upregulated in intestinal cells during fasting, whereas no overlap was observed for downregulated genes. Thus, the formation of inner and outer chromatin rings during fasting and Pol I depletion is more correlated with gene activation than gene repression.

2. The new RNAseq data are very interesting, however, they have been overstated for the correlation to the 3D-genome reorganization. The data have clearly demonstrated that during fasting there is large fraction of genes that are upregulated and this upregulation depends on Pol I transcription since they are also upregulated upon depletion of the Pol I factor RPOA-2. However, there is no correlation with 3D-genome reorganization since the authors cannot perform DNA-FISH analyses to determine where regulated genes are located during fasting or upon Pol I depletion. Thus, the whole section must be toned down since these data have no power to "determine whether specific genes are sensitive to the reorganization of the intestinal genome" (lane 353).

Lane 373. "These results indicate that the conditions that result in reorganization of the 3D genome into chromatin rings correlate with the upregulation of a specific subset of genes". This is an overstatement that is not corroborated by the data, which have only indicated that Pol I activity is implicated in promoting the expression of genes during fasting. This is a very nice result!

Lanes 384-5 "Intriguingly, we found that, instead, genes that are upregulated when rings are formed regardless of animals being fed or fasted are enriched in GO categories related to metabolism and stress response (Fig. 7i)." This sentence is very unclear. I have thought that the rings only form during fasting. How these genes were identified?

Minor point

Lines 141-142: "in agreement with the role of this mark in perinuclear positioning of genes in worms^{12, 19} and mammals²⁰⁻²³".

The authors have discussed and incorporated references regarding the perinucleolar compartment as a repressive compartment, following my suggestion. However, they primarily cited studies on the nuclear periphery in general and overlooked works specifically focused on the nucleolus in genome organization. I believe this could be improved further also because their study is mainly on the

nucleolus.

Reviewer #3 (Remarks to the Author):

The revised manuscript adds extensive additional experiments and has been revised appropriately to improve the manuscript. Overall the authors have done an excellent job in adding substantial experimental data to strengthen their findings.

My main concerns for the original manuscript were:

1. Is the double-sphere reorganization really a tissue-specific event?
2. Is the outer nucleolar area repressive?

Regarding the first point, the authors have now characterized the chromatin changes in intestine, hypodermis, and muscle, and show that the double ring reorganization only occurs in the intestine at 12 hours, while pre-rRNA is reduced to comparable extents in all tissues. Interestingly, pol I knockdown can induce the double ring structure also in hypodermis. I would guess non-intestinal cells may show the double ring structure at later fasting time-points, but that would be outside the scope of this study. The authors have also checked in an eat-2 mutant and shows that chronic reduced dietary intake cannot induce the double ring structure, suggesting acute starvation is necessary to induce this response.

Regarding the second point, the authors have characterized mutants of H3K9me and H3K27me for the double-ring phenotype and the effect on the euchromatic reporter. Interestingly, these repressive marks are not required for the formation of the double-ring structure. Therefore, chromatin reorganization can occur without these epigenetic marks. However, H3K9me affects positioning of the euchromatic reporter and the allele-position dependent expression changes, suggesting H3K9me plays a role in reducing expression of the euchromatic reporter allele at the outer ring. These are interesting findings that add to the mechanistic understanding of the repressive nature of the outer ring. I would still suggest that the interpretation could be a bit more balanced in terms of stating whether "the chromatin rings are repressive." (e.g. lines 91-92; lines 162-166; lines 418-420)

The authors have provided additional data that could be interpreted in a different way:

1. The authors see a strong skew towards upregulation in the intestine-specific RNAseq (2982 up vs 351 down) at a time point when ring formation has already occurred; suggesting that many genes are being activated despite forming the double ring structure.
2. The authors see MRG-1 in the ring but not HPL-1 and HPL-2. Although I agree with the authors that we cannot draw strong conclusions from these localization experiments, I think this is a very interesting finding and leaves the door open for an alternative interpretation. The MRG-1 expression image provided looks punctate around the nuclear lamina (compared to other images provided with double ring formation) potentially representing active compartments.

I am fine with the authors highlighting their interpretation, which is very reasonable and fits with the general dogma but hope that an alternative interpretation could also be included in 1 or 2 sentences discussing the potential of these rings promoting expression of starvation responsive genes or the role of euchromatic/active compartments within these generally heterochromatic/repressive regions.

One point of criticism is the discussion is a bit too centered on their own data and could be more relational to observations out there. For example, they should discuss their observations in relation to the described nucleolar vacuole (PMID: 37537842) or how histones are degraded in response to starvation (PMID: 37641865).

Decision Letter, Final Checks:

Our ref: NCB-LE51990A

10th July 2024

Dear Dr. Cabianca,

Thank you for your patience as we've prepared the guidelines for final submission of your Nature Cell Biology manuscript, "Fasting shapes chromatin architecture via an mTOR/RNA Pol I axis" (NCB-LE51990A). Please carefully follow the step-by-step instructions provided in the attached file, and add a response in each row of the table to indicate the changes that you have made. Ensuring that each point is addressed will help to ensure that your revised manuscript can be swiftly handed over to our production team.

In recognition of the time and expertise our reviewers provide to Nature Cell Biology's editorial process, we would like to formally acknowledge their contribution to the external peer review of your manuscript entitled "Fasting shapes chromatin architecture via an mTOR/RNA Pol I axis". For those reviewers who give their assent, we will be publishing their names alongside the published article.

Nature Cell Biology offers a Transparent Peer Review option for new original research manuscripts submitted after December 1st, 2019. As part of this initiative, we encourage our authors to support increased transparency into the peer review process by agreeing to have the reviewer comments, author rebuttal letters, and editorial decision letters published as a Supplementary item. When you submit your final files please clearly state in your cover letter whether or not you would like to participate in this initiative. Please note that failure to state your preference will result in delays in accepting your manuscript for publication.

Cover suggestions

COVER ARTWORK: We welcome submissions of artwork for consideration for our cover. For more information, please see our guide for cover artwork.

Nature Cell Biology has now transitioned to a unified Rights Collection system which will allow our Author Services team to quickly and easily collect the rights and permissions required to publish your work. Approximately 10 days after your paper is formally accepted, you will receive an email in providing you with a link to complete the grant of rights. If your paper is eligible for Open Access, our Author Services team will also be in touch regarding any additional information that may be required to arrange payment for your article.

Please note that *Nature Cell Biology* is a Transformative Journal (TJ). Authors may publish their research with us through the traditional subscription access route or make their paper immediately open access through payment of an article-processing charge (APC). Authors will not be required to make a final decision about access to their article until it has been accepted. Find out more about Transformative Journals

[Redacted]

Best regards,

Kendra Donahue
Staff
Nature Cell Biology

On behalf of

Melina Casadio, PhD
Senior Editor, Nature Cell Biology
ORCID ID: <https://orcid.org/0000-0003-2389-2243>

Reviewer #2:

Remarks to the Author:

The authors have made significant improvements to the manuscript. I am pleased with the additional experiments addressing various nucleolar aspects, such as size and pre-rRNA measurements, and the analyses of other cell types. I support the publication of this work but have two points that should be addressed.

1. I am not satisfied with how the important point 7, concerning the measurements of gene expression levels of the euchromatin reporter using Cherry-Histone protein levels (Fig. 2g,h), has been addressed. These data should be removed from the manuscript. This revision will not affect the overall work, which describes a compelling 3D-genome reorganization during fasting and its regulation by Pol I and mTOR signaling

While I understand that direct measurements of mRNA levels for the histone-Cherry reporter are technically unfeasible, it is wrong to measure reporter expression at the protein level in cells where ribosome biogenesis and translational activity are known to be compromised. The use of H3K9me depletion to prove that the measurements are correct is unconvincing. Additionally, as noted by Reviewer 3 (point 2), the choice of the pha-4 promoter, which controls the expression of pha-4 - a gene essential for dietary restriction-mediated longevity - may not support the conclusion that active genes near nucleoli become transcriptionally repressed, as this gene might be regulated by other nutrient signaling pathways. The appropriate choice should have been a housekeeping gene. Therefore, there is no data demonstrating that the inner ring near nucleoli is a compartment that establishes gene

repression. The presence of repressive chromatin around nucleoli in many cell types has been well documented. However, there is currently no evidence showing that an active gene becomes repressed due to its proximity to nucleoli. The results presented in this manuscript cannot demonstrate this for the reasons mentioned above. Additionally, the new RNA-seq data suggest an upregulation of gene expression upon fasting. Most genes upregulated by Pol I inhibition in fed animals were also upregulated in intestinal cells during fasting, whereas no overlap was observed for downregulated genes. Thus, the formation of inner and outer chromatin rings during fasting and Pol I depletion is more correlated with gene activation than gene repression.

2. The new RNAseq data are very interesting, however, they have been overstated for the correlation to the 3D-genome reorganization. The data have clearly demonstrated that during fasting there is large fraction of genes that are upregulated and this upregulation depends on Pol I transcription since they are also upregulated upon depletion of the Pol I factor RPOA-2. However, there is no correlation with 3D-genome reorganization since the authors cannot perform DNA-FISH analyses to determine where regulated genes are located during fasting or upon Pol I depletion. Thus, the whole section must be toned down since these data have no power to “determine whether specific genes are sensitive to the reorganization of the intestinal genome” (lane 353).

Lane 373. “These results indicate that the conditions that result in reorganization of the 3D genome into chromatin rings correlate with the upregulation of a specific subset of genes”. This is an overstatement that is not corroborated by the data, which have only indicated that Pol I activity is implicated in promoting the expression of genes during fasting. This is a very nice result!

Lanes 384-5 “Intriguingly, we found that, instead, genes that are upregulated when rings are formed regardless of animals being fed or fasted are enriched in GO categories related to metabolism and stress response (Fig. 7i).” This sentence is very unclear. I have thought that the rings only form during fasting. How these genes were identified?

Minor point

Lines 141-142: “in agreement with the role of this mark in perinuclear positioning of genes in worms^{12, 19} and mammals²⁰⁻²³”.

The authors have discussed and incorporated references regarding the perinucleolar compartment as a repressive compartment, following my suggestion. However, they primarily cited studies on the nuclear periphery in general and overlooked works specifically focused on the nucleolus in genome organization. I believe this could be improved further also because their study is mainly on the nucleolus.

Reviewer #3:

Remarks to the Author:

The revised manuscript adds extensive additional experiments and has been revised appropriately to

improve the manuscript. Overall the authors have done an excellent job in adding substantial experimental data to strengthen their findings.

My main concerns for the original manuscript were:

1. Is the double-sphere reorganization really a tissue-specific event?
2. Is the outer nucleolar area repressive?

Regarding the first point, the authors have now characterized the chromatin changes in intestine, hypodermis, and muscle, and show that the double ring reorganization only occurs in the intestine at 12 hours, while pre-rRNA is reduced to comparable extents in all tissues. Interestingly, pol I knockdown can induce the double ring structure also in hypodermis. I would guess non-intestinal cells may show the double ring structure at later fasting time-points, but that would be outside the scope of this study. The authors have also checked in an eat-2 mutant and shows that chronic reduced dietary intake cannot induce the double ring structure, suggesting acute starvation is necessary to induce this response.

Regarding the second point, the authors have characterized mutants of H3K9me and H3K27me for the double-ring phenotype and the effect on the euchromatic reporter. Interestingly, these repressive marks are not required for the formation of the double-ring structure. Therefore, chromatin reorganization can occur without these epigenetic marks. However, H3K9me affects positioning of the euchromatic reporter and the allele-position dependent expression changes, suggesting H3K9me plays a role in reducing expression of the euchromatic reporter allele at the outer ring. These are interesting findings that add to the mechanistic understanding of the repressive nature of the outer ring. I would still suggest that the interpretation could be a bit more balanced in terms of stating whether “the chromatin rings are repressive.” (e.g. lines 91-92; lines 162-166; lines 418-420)

The authors have provided additional data that could be interpreted in a different way:

1. The authors see a strong skew towards upregulation in the intestine-specific RNAseq (2982 up vs 351 down) at a time point when ring formation has already occurred; suggesting that many genes are being activated despite forming the double ring structure.
2. The authors see MRG-1 in the ring but not HPL-1 and HPL-2. Although I agree with the authors that we cannot draw strong conclusions from these localization experiments, I think this is a very interesting finding and leaves the door open for an alternative interpretation. The MRG-1 expression image provided looks punctate around the nuclear lamina (compared to other images provided with double ring formation) potentially representing active compartments.

I am fine with the authors highlighting their interpretation, which is very reasonable and fits with the general dogma but hope that an alternative interpretation could also be included in 1 or 2 sentences discussing the potential of these rings promoting expression of starvation responsive genes or the role of euchromatic/active compartments within these generally heterochromatic/repressive regions. One point of criticism is the discussion is a bit too centered on their own data and could be more relational to observations out there. For example, they should discuss their observations in relation to the described nucleolar vacuole (PMID: 37537842) or how histones are degraded in response to starvation (PMID: 37641865).

Author Rebuttal, first revision:

Dear Melina Casadio, dear reviewers,

We thank you for your comments, which we found very useful and insightful.

As you will read below, we addressed all the reviewers' remaining concerns, including the removal of panel g in Figure 2, which, as explained in more detail below, we agree might lead to misinterpretation by the reader.

Wherever we mention line numbers, we refer to the manuscript file where all changes were accepted. The new sentences are easily recognizable because they are underlined in the main text. Nonetheless, to facilitate checking the corrections we made to directly address the referees' points, the sentences are also directly pasted in this point-by-point response.

We believe the paper has greatly improved since its first submission. We hope you will find it suitable for publication in Nature Cell Biology.

Best wishes,

Daphne Cabianca

Reviewer #2 (Remarks to the Author):

The authors have made significant improvements to the manuscript. I am pleased with the additional experiments addressing various nucleolar aspects, such as size and pre-rRNA measurements, and the analyses of other cell types. I support the publication of this work but have two points that should be addressed.

We thank the reviewer for this positive recognition of our revision work.

1. I am not satisfied with how the important point 7, concerning the measurements of gene expression levels of the euchromatin reporter using Cherry-Histone protein levels (Fig. 2g,h), has been addressed. These data should be removed from the manuscript. This revision will not affect the overall work, which describes a compelling 3D-genome reorganization during fasting and its regulation by Pol I and mTOR signaling

While I understand that direct measurements of mRNA levels for the histone-Cherry reporter are technically unfeasible, it is wrong to measure reporter expression at the protein level in cells where ribosome biogenesis and translational activity are known to be compromised. The use of H3K9me depletion to prove that the measurements are correct is unconvincing. Additionally, as noted by Reviewer 3 (point 2), the choice of the pha-4 promoter, which controls the expression of pha-4 - a gene essential for dietary restriction-mediated longevity - may not support the conclusion that active genes near nucleoli become transcriptionally repressed, as this gene might be regulated by other nutrient signaling pathways. The appropriate choice should have been a housekeeping gene. Therefore, there is no data demonstrating that the inner ring near nucleoli is a compartment that establishes gene repression. The presence of repressive chromatin around nucleoli in many cell types has been well documented. However, there is currently no evidence showing that an active gene becomes repressed due to its proximity to nucleoli. The results presented in this manuscript cannot demonstrate this for the

reasons mentioned above. Additionally, the new RNA-seq data suggest an upregulation of gene expression upon fasting. Most genes upregulated by Pol I inhibition in fed animals were also upregulated in intestinal cells during fasting, whereas no overlap was observed for downregulated genes. Thus, the formation of inner and outer chromatin rings during fasting and Pol I depletion is more correlated with gene activation than gene repression.

We fully understand the reviewer's concerns. While we did not propose that gene repression is established at chromatin rings, we see how the data in Fig. 2g might mislead the readers towards this interpretation. We agree with the reviewer that the repressive (or not) nature of the chromatin rings should be tested more thoroughly and with different experimental tools in another study. Hence, we removed Fig. 2g from the revised manuscript, as suggested by the reviewer. We believe that the referment to Figure 2h is a typo, as Figure 2h did not exist.

2. The new RNAseq data are very interesting, however, they have been overstated for the correlation to the 3D-genome reorganization. The data have clearly demonstrated that during fasting there is large fraction of genes that are upregulated and this upregulation depends on Pol I transcription since they are also upregulated upon depletion of the Pol I factor RPOA-2. However, there is no correlation with 3D-genome reorganization since the authors cannot perform DNA-FISH analyses to determine where regulated genes are located during fasting or upon Pol I depletion. Thus, the whole section must be toned down since these data have no power to “determine whether specific genes are sensitive to the reorganization of the intestinal genome” (lane 353).

Lane 373. “These results indicate that the conditions that result in reorganization of the 3D genome into chromatin rings correlate with the upregulation of a specific subset of genes“. This is an overstatement that is not corroborated by the data, which have only indicated that Pol I activity is implicated in promoting the expression of genes during fasting. This is a very nice result!

Lanes 384-5 “Intriguingly, we found that, instead, genes that are upregulated when rings are formed regardless of animals being fed or fasted are enriched in GO categories related to metabolism and stress response (Fig. 7i).” This sentence is very unclear. I have thought that the rings only form during fasting. How these genes were identified?

We apologize if the RNA-seq data were presented in an unclear way. The rings form i) during fasting (Fig 1a-e) and ii) upon RNA Pol I depletion in fed animals (Fig 6a-c and Ext. Data Fig 8c-e). Therefore, genes that are upregulated in both conditions are not strictly dependent on the nutritional status of the animal. However, to avoid generating confusion into the readers, we changed the old sentence in lines 384-5 to the NEW sentence, lines 367-369 “Intriguingly, we found that, instead, genes that are upregulated when rings are formed either in fed animals lacking RNA Pol I or during fasting are enriched in GO categories related to metabolism and stress response (Fig. 8i).”

We do agree with the reviewer that our data do not allow us to draw conclusions on the role of the 3D genome reorganization *per se* in regulating gene expression, as the co-regulated genes might solely depend on RNA Pol I activity. For this reason, we removed the terms correlate and correlations to describe the RNA-seq results throughout the text, including the abstract and results subheading. Next, we changed the sentences as follows:

- 1) Old line 353, now lines 332-337 “Thus, a first step to identify genes that might be sensitive to the reorganization of the intestinal chromatin into two rings, is to uncouple the configuration of the 3D genome from the nutritional status of the animal and quantify gene expression changes (Fig. 8b). To this aim, we performed RNA-seq in intestinal cells of fed adults where chromatin rings are induced by AID-mediated RNA Pol I inhibition (Extended Data Fig. 8c-e).
- 2) Old line 373, now lines 353-356 “Thus, in the two different conditions where the intestinal genome is reorganized into chromatin rings, we observed the upregulation of an overlapping set of genes. Whether their expression is regulated by RNA Pol I activity alone or through the reorganization of the 3D genome remains to be determined.
- 3) And in the discussion we added the following sentence, new lines 421-424 “Whether the observed changes in gene expression are driven by RNA Pol I inhibition, the 3D genome reconfiguration, or both, remains to be determined. Nonetheless, our data provide a basis to further explore the regulation of RNA Pol II targets by a 3D chromatin configuration that is modulated by RNA Pol I activity.

Minor point

Lines 141-142: “in agreement with the role of this mark in perinuclear positioning of genes in worms^{12, 19} and mammals²⁰⁻²³”.

The authors have discussed and incorporated references regarding the perinucleolar compartment as a repressive compartment, following my suggestion. However, they primarily cited studies on the nuclear periphery in general and overlooked works specifically focused on the nucleolus in genome organization. I believe this could be improved further also because their study is mainly on the nucleolus.

We thank the reviewer for pointing out this gap in our citations. To improve this, we have now added original research articles from the Lamond (van Koningsbruggen et al., 2010), Längst (Nemeth et al., 2010), Groudine (Ragoczy et al., 2014) and Santoro (Bersaglieri et al., 2022) laboratories, which focus on the nucleolus in genome organization, as suggested by the reviewer, and refer to them in the text: new line 199, references 39-42.

Reviewer #3 (Remarks to the Author):

The revised manuscript adds extensive additional experiments and has been revised appropriately to improve the manuscript. Overall the authors have done an excellent job in adding substantial experimental data to strengthen their findings.

We thank the reviewer for this very positive evaluation of our revision work.

My main concerns for the original manuscript were:

1. Is the double-sphere reorganization really a tissue-specific event?
2. Is the outer nucleolar area repressive?

Regarding the first point, the authors have now characterized the chromatin changes in intestine,

hypodermis, and muscle, and show that the double ring reorganization only occurs in the intestine at 12 hours, while pre-rRNA is reduced to comparable extents in all tissues. Interestingly, pol I knockdown can induce the double ring structure also in hypodermis. I would guess non-intestinal cells may show the double ring structure at later fasting time-points, but that would be outside the scope of this study. The authors have also checked in an eat-2 mutant and shows that chronic reduced dietary intake cannot induce the double ring structure, suggesting acute starvation is necessary to induce this response.

Regarding the second point, the authors have characterized mutants of H3K9me and H3K27me for the double-ring phenotype and the effect on the euchromatic reporter. Interestingly, these repressive marks are not required for the formation of the double-ring structure. Therefore, chromatin reorganization can occur without these epigenetic marks. However, H3K9me affects positioning of the euchromatic reporter and the allele-position dependent expression changes, suggesting H3K9me plays a role in reducing expression of the euchromatic reporter allele at the outer ring. These are interesting findings that add to the mechanistic understanding of the repressive nature of the outer ring.

We are happy that the reviewer found our new results interesting.

I would still suggest that the interpretation could be a bit more balanced in terms of stating whether “the chromatin rings are repressive.” (e.g. lines 91-92; lines 162-166; lines 418-420). The authors have provided additional data that could be interpreted in a different way:

1. The authors see a strong skew towards upregulation in the intestine-specific RNAseq (2982 up vs 351 down) at a time point when ring formation has already occurred; suggesting that many genes are being activated despite forming the double ring structure.
2. The authors see MRG-1 in the ring but not HPL-1 and HPL-2. Although I agree with the authors that we cannot draw strong conclusions from these localization experiments, I think this is a very interesting finding and leaves the door open for an alternative interpretation. The MRG-1 expression image provided looks punctate around the nuclear lamina (compared to other images provided with double ring formation) potentially representing active compartments.

I am fine with the authors highlighting their interpretation, which is very reasonable and fits with the general dogma but hope that an alternative interpretation could also be included in 1 or 2 sentences discussing the potential of these rings promoting expression of starvation responsive genes or the role of euchromatic/active compartments within these generally heterochromatic/repressive regions.

We thank the reviewer for these insightful comments. Considering the new results obtained and the concerns raised by both reviewers, we agree that our results do not allow us to unambiguously conclude that the chromatin rings are repressive. Furthermore, reviewer 2 raised substantial concerns on the same point (the chromatin rings being repressive) and asked for the data on the histone-mCherry expression (Fig. 2g) to be removed from the revised manuscript. To avoid misinterpretations of the data, we decided to follow his/her advice. Therefore, in the current revised text there is no description of the chromatin rings being repressive.

One point of criticism is the discussion is a bit too centered on their own data and could be more relational to observations out there. For example, they should discuss their observations in relation to

the described nucleolar vacuole (PMID: 37537842) or how histones are degraded in response to starvation (PMID: 37641865).

We thank the reviewer for pointing this out. To improve the discussion of the revised manuscript, we now cite the papers he/she mentions and relate them to our own observations, as suggested by the reviewer.

- 1) New lines 403-407 “Further confirming that nucleolar structure is regulated tissue-specifically, vacuole-containing nucleoli are prominent in intestine but not in hypodermal cells at the L3-L4 stage in *C. elegans* and their formation is promoted by an alternative rRNA processing pathway⁷¹. This suggests that, like nucleolar size, the processing of rRNA varies across cell types and might be implicated in 3D genome organization in response to nutrients.” (Ref 71 is PMID: 37537842).
- 2) New lines 413-614 “in *C. elegans*, prolonged starvation leads to a global degradation of histone H2Bs⁷³. The tissue-specific dynamics of histone abundance in the early stages of fasting is not known, and its investigation might help to shed light on the mechanism of chromatin ring formation.” (Ref 73 is PMID: 37641865).

Additionally, to further help putting our work in the context of what previously done, we now discuss two more citations, one on mTOR inhibition affecting histone levels in flies (PMID: 33988501) and one showing that heterochromatin silencing is compromised when the number of rDNA repeats is reduced and the nucleolus is remodeled (PMID: 19822756):

- 1) New lines 412-413 “Rapamycin-induced inhibition of the mTOR pathway increases core histone expression in the intestine of *Drosophila melanogaster*⁷².” (Ref 72 is PMID: 33988501)
- 2) New lines 395-397 “Accordingly, deleting rDNA repeats in *Drosophila melanogaster* leaves the steady-state concentration of rRNA unaltered but remodels the nucleolus and compromises heterochromatic silencing in other sites of the genome⁷⁰.” (Ref 70 is PMID: 19822756)

Final Decision Letter:

Dear Dr Cabisanica,

I am pleased to inform you that your manuscript, "Fasting shapes chromatin architecture through an mTOR/RNA Pol I axis", has now been accepted for publication in Nature Cell Biology.

You may wish to make your media relations office aware of your accepted publication, in case they consider it appropriate to organize some internal or external publicity. Once your paper has been scheduled you will receive an email confirming the publication details. This is normally 3-4 working days in advance of publication. If you need additional notice of the date and time of publication, please let the production team know when you receive the proof of your article to ensure there is

sufficient time to coordinate. Further information on our embargo policies can be found here: <https://www.nature.com/authors/policies/embargo.html>

Please note that *Nature Cell Biology* is a Transformative Journal (TJ). Authors may publish their research with us through the traditional subscription access route or make their paper immediately open access through payment of an article-processing charge (APC). Authors will not be required to make a final decision about access to their article until it has been accepted. Find out more about Transformative Journals

If you have not already done so, we strongly recommend that you upload the step-by-step protocols used in this manuscript to protocols.io (<https://protocols.io>), an open online resource that allows researchers to share their detailed experimental know-how. All uploaded protocols are made freely available and are assigned DOIs for ease of citation. Protocols and Nature Portfolio journal papers in which they are used can be linked to one another, and this link is clearly and prominently visible in the online versions of both. Authors who performed the specific experiments can act as primary authors for the Protocol as they will be best placed to share the methodology details, but the Corresponding Author of the present research paper should be included as one of the authors. By uploading your Protocols onto protocols.io, you are enabling researchers to more readily reproduce or adapt the methodology you use, as well as increasing the visibility of your protocols and papers. You can also establish a dedicated workspace to collect your lab Protocols. Further information can be found at <https://www.protocols.io/help/publish-articles>.

With kind regards,

Melina Casadio, PhD
Senior Editor, Nature Cell Biology
ORCID ID: <https://orcid.org/0000-0003-2389-2243>

** Visit the Springer Nature Editorial and Publishing website at www.springernature.com/editorial-and-publishing-jobs for more information about our career opportunities. If you have any questions please click here.**